# Towards a theory of how the structure of language is acquired by deep neural networks

**Francesco Cagnetta**
Institute of Physics
École Polytechnique Fédérale de Lausanne
`francesco.cagnetta@epfl.ch`

**Matthieu Wyart**
Institute of Physics
École Polytechnique Fédérale de Lausanne
`matthieu.wyart@epfl.ch`

## Abstract

How much data is required to learn the structure of a language via next-token prediction? We study this question for synthetic datasets generated via a Probabilistic Context-Free Grammar (PCFG)—a tree-like generative model that captures many of the hierarchical structures found in natural languages. We determine token-token correlations analytically in our model and show that they can be used to build a representation of the grammar's hidden variables, the longer the range the deeper the variable. In addition, a finite training set limits the resolution of correlations to an effective range, whose size grows with that of the training set. As a result, a Language Model trained with increasingly many examples can build a deeper representation of the grammar's structure, thus reaching good performance despite the high dimensionality of the problem. We conjecture that the relationship between training set size and effective range of correlations holds beyond our synthetic datasets. In particular, our conjecture predicts how the scaling law for the test loss behaviour with training set size depends on the length of the context window, which we confirm empirically in Shakespeare's plays and Wikipedia articles.

## 1 Introduction

Two central foci of linguistics are the language structure and how humans acquire it. Formal language theory, for instance, describes languages with hierarchical generative models of grammar, classified in different levels of complexity [1, 2]. In this context, the 'poverty of the stimulus' argument [3]—stating that the data children receive is insufficient to uniquely determine the grammatical structure of their language—led to the hypothesis that linguistic faculties are largely innate. By contrast, statistical learning theory [4, 5] posits that the statistics of the input data can be used to deduce the language structure. This assumption is supported by empirical evidence concerning a broad range of tasks, including word segmentation [6] and reconstruction of the hierarchical phrase structure [7].

Large Language Models (LLMs) offer an interesting perspective on the subject. For instance, the success of LLMs trained for next-token prediction [8, 9] establishes that a language can be acquired from examples alone—albeit with a training set much larger than what humans are exposed to. Furthermore, empirical studies of LLMs' representations showed that they learn a hierarchy of contextual information, including notions of linguistics such as word classes and syntactic structure [10, 11, 12]. Recent studies have begun revealing the inner workings of LLMs by using synthetic data generated via context-free grammars [13, 14], determining, in particular, the algorithm that these models follow when predicting the next token. However, there is no consensus on the mechanisms behind language *acquisition* by LLMs [15, 16]. As a result, empirical phenomena such as the *scaling* of the test loss with dataset size and number of parameters [17] and the *emergence* of specific skills at certain scales [18, 19] remain unexplained. In this work, we use hierarchical generative models of data to describe how the structure of a language is learnt as the training set grows.

38th Conference on Neural Information Processing Systems (NeurIPS 2024).

## 1.1 Our contributions

We consider synthetic datasets generated via the Random Hierarchy Model (RHM) [20], an ensemble of probabilistic context-free grammars (PCFGs). The RHM generates sequences of tokens by applying randomly chosen production rules to a hierarchy of hidden variables that live on the nodes of a tree with fixed geometry.

- We characterise analytically the power-law decay of the correlations between tokens with their distance. We then show that, because of this decay, a finite training set size $P$ limits the resolution of correlations to an effective context window, whose size $t^*$ increases with $P$.

- Building on previous works on classification, we argue that deep learning models trained on next-token prediction can use measurable correlations to represent the hidden variables of the PCFG, with larger $P$ allowing the representation of deeper hidden variables.

- Combining these results, we predict a sequence of sample complexities where the emergence of a deeper data structure representation leads to a jump in test loss. We empirically validate this for deep transformers and CNNs. Notably, the sample complexities are polynomial in the effective context size $t^*$, avoiding the curse of dimensionality.

- We conjecture that the relationship between training set size, correlations and effective context window holds beyond our data model, and we test it by training deep transformers on collections of Shakespeare's lines and Wikipedia articles. In particular, we find that the test loss decay levels off at a characteristic training set size that depends on the length of the context window and can be measured from token-token correlations.

## 1.2 Additional related works

Fixed-tree hierarchical generative models have been introduced to study phylogeny [21], then used in supervised learning [22, 23, 20, 24] and score-based diffusion [25, 26]. In particular, [22] introduced a sequential clustering algorithm that reveals the importance of correlations between the input features and the labels for supervised learning. The RHM of [20] provides a framework to show how features-label correlations emerge from the generative model and can be used by deep networks to represent the hidden hierarchical structure of the data. Here we extend this result to self-supervised learning, where the relevant correlations are those between the different input features.

PCFGs can in principle generate sequences with token correlations that decay as a power of their distance [27]. When the production rule probabilities are random [28, 29], these probabilities must follow a broad distribution for the data to retain information about the generative process. Learning a PCFG from examples is a longstanding problem of theoretical linguistics [30]. While some PCFG classes are learnable using distributional information [31], the sample complexity is unknown. In the context of deep learning, PCFGs have been used to study how trained transformers encode the grammar's structure [13, 14]. [14], in particular, showed that the operations performed by BERT-like transformers resemble well-known algorithms for grammatical inference, and proved that, for PCFG data, these algorithms are optimal solutions of the masked language modelling objective. However, when the training data is compatible with both a PCFG and a non-hierarchical generative model, neither recurrent language models [32] nor transformer [33] consistently prefer the hierarchical explanation. In addition, none of these works study the learning process.

Empirical work on the learning dynamics of Long Short-Term Memories showed that short-range dependencies are learnt first, then used as a foundation for forming longer-range dependencies [34]. Our work introduces a theoretical framework to explain this hierarchical inductive bias, focusing on the learning curves of deep learning architectures. Shortly after our submission, [35] unveiled another form of hierarchical inductive bias in the training dynamics of transformers, whereby many-body interactions among tokens are learnt in the order of the interaction's degree.

## 2 Notation and setup

This work focuses on the pretraining phase of language models, aimed at building an approximation of the data distribution via unlabelled examples [8, 9]. Let us define a text datum, or sentence, as a sequence $\boldsymbol{x} = (x_1, \ldots, x_d)$ of $d$ tokens belonging to a finite vocabulary $\mathcal{V}$. Denoting with $v$ the

vocabulary size, each token $x_i$ is represented as a $v$-dimensional one-hot vector $(x_{i,\mu})_{\mu=1,\dots,v}$ [1]:

$$x_{i,\mu} = \begin{cases} 1, & \text{if } x_i \equiv \mu\text{-th element of } \mathcal{V}, \\ 0, & \text{otherwise.} \end{cases} \tag{1}$$

A dataset, or *corpus*, consists of a probability distribution over sequences, which measures the frequency at which a given combination of tokens appears within the text. Assuming that all sequences have length $d$, the data distribution is a joint probability over $d$-dimensional sequences with elements in $\mathcal{V}$, $P_{\boldsymbol{X}}(\boldsymbol{x}) := \mathbb{P}\{X_1 = x_1, \dots, X_d = x_d\}$. The specifics of the approximation of $P_{\boldsymbol{X}}$ depend on the training objective. In Masked Language Modelling, for instance, a random fraction of tokens is masked, i.e. replaced with a fixed token $x_{\text{mask}}$, and the model is tasked with predicting their value [8]. Autoregressive language models, instead, are trained to predict the $i$-th token of a sequence based on all the previous ones [9]. Here we consider a simplified setup where the last token of the sequence is masked and the model is trained to predict it. In other words, the model takes the *context window* $(x_1, \dots, x_{d-1})$ as input and outputs a parametric approximation $p_\theta$ of the conditional probability of the last token,

$$p_\theta(x_d|x_1, \dots, x_{d-1}) \approx \mathbb{P}\{X_d = x_d | X_1 = x_1, \dots, X_{d-1} = x_{d-1}\}, \tag{2}$$

obtained by updating the parameters $\theta$ via gradient descent on the empirical cross-entropy,

$$\mathcal{L}(\mathcal{X}_P) = -\frac{1}{P} \sum_{\boldsymbol{x} \in \mathcal{X}_P} \log\left(p_\theta(x_d|x_1, \dots, x_{d-1})\right), \tag{3}$$

where $\mathcal{X}_P$ is a set of $P$ training examples drawn from $P_{\boldsymbol{X}}$. Numerical experiments are performed in PyTorch [36], with the code available at `https://github.com/fracagnetta/random-hierarchy-model`. Details of the machine learning models, training hyperparameters and computer resources are presented in App. A.

## 2.1 Hierarchical generative models

To model the hierarchical structure of sentences, we consider synthetic datasets generated via a probabilistic context-free grammar (PCFG) [37]. PCFGs are collections of symbols and rules that prescribe how to generate sequences. In particular, the PCFGs we consider consist of

- $L$ finite vocabularies of hidden (nonterminal) symbols $(\mathcal{V}_\ell)_{\ell=1,\dots,L}$;
- A finite vocabulary of observable (terminal) symbols $\mathcal{V} \equiv \mathcal{V}_0$;
- $L$ sets of *production rules* describing how one symbols of $\mathcal{V}_\ell$ generates a tuple of symbols of $\mathcal{V}_{\ell-1}$, for $\ell = 1, \dots, L$.

Production rules take the form

$$\mu^{(\ell)} \to \mu_1^{(\ell-1)}, \dots, \mu_{s_\ell}^{(\ell-1)}, \quad \text{for } \mu^{(\ell)} \in \mathcal{V}_\ell, \mu_i^{(\ell-1)} \in \mathcal{V}_{\ell-1}, \tag{4}$$

for some integer size $s_\ell \geq 1$. The left panel of Fig. 1 shows an example of the generative process, represented as a tree: pick (uniformly at random) a level-3 symbol (root) and one of the production rule having that symbol on the left-hand side (also uniformly at random), replace the symbol with the right-hand side of the production rules (first generation), then repeat the process until left with only terminal symbols (leaves). The resulting datum is a sequence in $(\mathcal{V}_0)^d$, with $d = \prod_\ell s_\ell$. Assuming a finite number of production rules emanating from each nonterminal symbol, this model generates a finite number of $d$-dimensional sequences. Since the probabilities of the level-$L$ symbol and the production rules are uniform, the data distribution $P_{\boldsymbol{X}}$ is uniform over the generated sequences.

The Random Hierarchy Model (RHM) of [20] is an ensemble of such generative models, obtained by prescribing a probability distribution over production rules. In particular, the $\ell$-th set of production rules is chosen uniformly at random between all the *unambiguous* sets of rules in the form of Eq. 4. Unambiguity means that each $s_\ell$-tuple of level-$(\ell-1)$ symbols can be generated by one level-$\ell$ symbol at most. The uniform probability and unambiguity assumptions are not satisfied in a generic natural language, but they allow us to characterise quantitatively the effects of the hierarchical structure. We will further assume, to ease notation, that all the vocabularies $\mathcal{V}_\ell$ have the same size $v$ and that the size of the production rules is homogeneous, i.e. $s_\ell = s$ for all $\ell$. We further assume that each nonterminal appears as the left-hand side of exactly $m$ production rules, i.e. the hidden symbols have $m$ equivalent low-level representations. Since there are $v^s$ distinct low-level representations and each of the $v$ high-level symbols is assigned $m$, unambiguity requires $m \leq v^{s-1}$.

---

[1] throughout the paper, Latin indices indicate the token position and Greek indices the vocabulary entry.

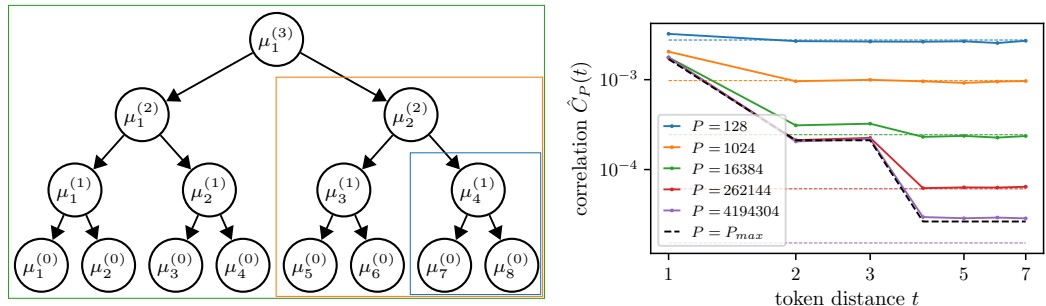

Figure 1: **Left:** Example of data generation according to the RHM, with depth $L=3$ and branching factor $s=2$. Starting from the root with $\ell=3$ and following the arrows, each level-$\ell$ symbol is replaced with a pair of lower-level symbols, down to the leaves with $\ell=0$. **Right:** Empirical (coloured) and analytical (black dashed) correlation functions of RHM data, with $L=3$, $s=2$, $v=32$ and $m=8$. The stepwise decay mirrors the tree structure of the generative model. Empirical estimates obtained from $P$ examples initially follow the true correlation function, but then saturate due to the sampling noise (coloured dashed). As a result, a finite training set only allows for measuring correlations with the tokens up to a certain distance $t^*(P)$. Graphically, $t^*(P)$ corresponds to the highest value of $t$ where the empirical estimate matches the true correlation (e.g. 1 for the orange and green curves, 3 for the red curve).

## 3 Correlations, training set size and effective context window

Given a dataset of $d$-dimensional sequences of tokens in $\mathcal{V}$, we measure correlations via the token co-occurrences matrix, [2]

$$C_{i,j}(\mu,\nu) := \mathbb{P}\{X_i=\mu, X_j=\nu\} - \mathbb{P}\{X_i=\mu\}\,\mathbb{P}\{X_j=\nu\}, \tag{5}$$

where $\mu$ and $\nu$ are arbitrary elements of the vocabulary $\mathcal{V}$ and $\mathbb{P}$ refers to the data distribution $P_{\mathbf{X}}$. Since the masked token is always the last in our setup, it is convenient to set $j=d$ and write $C_{i,d}$ as a function of the distance $t=|i-d|$ between the $i$-th and the masked token. Taking the root mean square over the vocabulary yields the *correlation function*,

$$\tilde{C}(t) := \left(v^{-2} \sum_{\mu,\nu\in\mathcal{V}} (C_{d-t,d}(\mu,\nu))^2\right)^{1/2}, \tag{6}$$

which measures the typical dependency between tokens as a function of their distance $t$. For RHM data with $m=v^{s-1}$, $P_{\mathbf{X}}$ is uniform over all the possible sequences of tokens in $\mathcal{V}$ and there are no correlations. If, instead, $m<v^{s-1}$, the correlations strength depends on the distance. Fig. 1 shows an example for RHM data with $L=4$, $s=2$, $v=32$ and $m=8$.

**Correlations decay with distance.** The stepwise decay of $\tilde{C}(t)$ mirrors the tree structure of the generative model. The masked token has the highest correlations with those belonging to the same $s$-tuple, as they were all generated by the same level-1 symbol (as in the blue box of Fig. 1, left). The second highest is with the tokens generated by the same level-2 symbol (orange box in the figure), and so on until the root. Formally, with $\ell=1,\ldots,L$ denoting the height of the lowest common ancestor (LCA) of the $d$-th and $(d-t)$-th tokens,

$$\tilde{C}(t) = \tilde{C}^{(\ell)} \quad \forall \quad t=s^{\ell-1},\ldots,s^\ell-1; \quad \tilde{C}^{(1)} > \tilde{C}^{(2)} > \cdots > \tilde{C}^{(L)}. \tag{7}$$

These $L$ plateau values can be determined analytically in the large $v$ limit by approximating the variance over the vocabulary entries $\mu$ and $\nu$ on the right-hand side of Eq. 7 with the variance over realisations of the RHM. Denoting the average over such realisations with $\langle.\rangle$,

$$\tilde{C}^{(\ell)} = \left(\left\langle\left(C^{(\ell)}(\mu,\nu)\right)^2\right\rangle\right)^{1/2} \simeq \sqrt{\frac{(1-m/v^{s-1})}{v^3 m^{2\ell-1}}}, \tag{8}$$

---

[2]$C_{i,j}(\mu,\nu)$ is also equivalent to the covariance matrix of the one-hot representation, $\mathbb{E}\left[(X_{i,\mu}-\mathbb{E}\left[X_{i,\mu}\right])(X_{j,\nu}-\mathbb{E}\left[X_{j,\nu}\right])\right]$

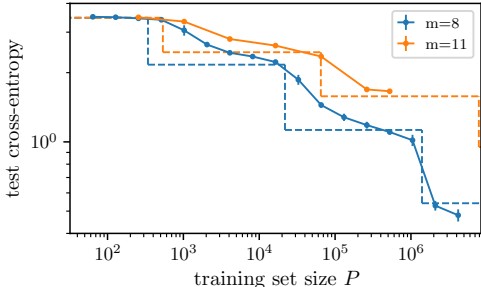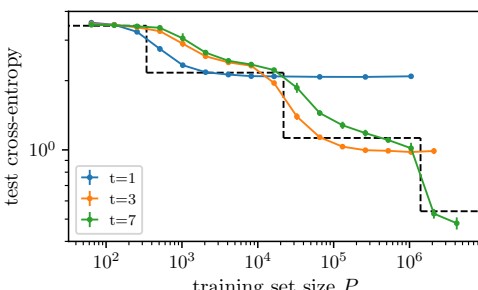

Figure 2: **Left:** Learning curves of depth-3 transformers trained on RHM data with $L = 3$, $s = 2$, $v = 32$ and $m = 8$ (blue) or 11 (orange, both are averaged over 8 independent realisations of the dataset and initialisations of the network), displaying a stepwise behaviour analogous to the correlation function. The vertical dashed lines mark the characteristic training set sizes $P_k$ at which the correlation with tokens at distances up to $t = s^k - 1$ emerge from the sampling noise. Horizontal dashed lines represent (upper bounds on) the cross-entropy of the probability of the last token conditioned on the previous $s^k - 1$, suggesting that the steps correspond to the model learning a progressively larger sub-tree of the data structure. **Right:** Learning curves of transformers for $m = 8$ and different sizes $t$ of the context window. The saturation of the loss decay due to the finite context window highlights that the decay is entirely due to the ability to leverage a larger portion of the context window.

where the rightmost equality is exact asymptotically in $v$ and $m$. Eq. 8 is derived in detail in App. D, confirmed empirically in the right panel of Fig. 1 and can be given a simple interpretation in terms of the sample size required for the empirical measurement of correlations, as discussed in the following paragraph. In addition, notice that, upon replacing $s^\ell$ with $t$, the $m^{-\ell}$ dependence on $\ell$ is approximated by a power-law decay $\tilde{C}(t) \sim t^{-\beta}$, with $\beta = \log m / \log s$.

**Saturation due to finite training set.** When measuring the correlation function from a finite sample $\mathcal{X}_P$ of $P$ data, there is an additional contribution due to the sampling noise. The scenario is illustrated in Fig. 1, right: the empirical estimates $\hat{C}_P(t)$, shown as coloured lines for different values of $P$, begin by following the descent of the true correlation function $\tilde{C}(t)$, shown as a black dashed line. However, empirical estimates saturate when approaching the sampling noise size $(v^2 P)^{-1/2}$, as proved in App. E and shown as dashed coloured lines in Fig. 1, right. Combining the saturation with the values of the steps, we deduce that a finite training set allows for the resolution of correlations up to distance $t^* = s^{\ell^*} - 1$ such that

$$\tilde{C}^{(\ell^*)} > (v^2 P)^{-1/2} > \tilde{C}^{(\ell^*+1)}. \tag{9}$$

Eq. 9 suggests that a language model trained with $P$ examples can only extract information from the tokens within distance $t^*(P)$ from the last. In other words, a finite training set is equivalent to an *effective context window* of size $t^*(P)$. If $\tilde{C} \sim t^{-\beta}$, then $t^*(P) \sim P^{1/2\beta}$. Alternatively, setting $\tilde{C}^{(\ell)} = (v^2 P)^{-1/2}$ yields a sequence of thresholds $P_\ell$ for the resolution of correlations of increasing range. From Eq. 8, $P_\ell \propto vm^{2\ell-1}$, which has a simple interpretation as the number of choices in the generative process to determine two tokens at a distance $t \in [s^{\ell-1}, s^\ell)$: $v$ choices for the level-$\ell$ LCA, $m$ for the first production rule and $m^2$ ($m$ per branch) for each of the remaining $\ell - 1$ generations.

## 4 Self-supervised learning of the Random Hierarchy Model

We now show how the correlations can be translated into a prediction of sample complexities that allow for a sequence of increasingly accurate approximations of the masked token probability, based on reconstructing the hidden variables of the generative tree. We then test these predictions in numerical experiments with deep networks.

### 4.1 Prediction of the sequence of performance steps and sample complexities

**Loss steps.** Due to the structure of the data, there is a natural sequence of $L$ increasingly accurate approximations of the last token probability in Eq. 2. For all $\ell = 1, \ldots, L$, these approximations are

realised by conditioning the probability of the last token on the previous $s^\ell - 1$. These approximations amount to using an effective context window of size $t_\ell = s^\ell - 1$. The effective context windows consist of the leaves of the subtree generated by the level-$\ell$ hidden symbol above the last token, as illustrated by the coloured boxes of Fig. 1, left. The resulting cross-entropy loss is given by

$$
\begin{aligned}
\mathcal{L}_\ell &= \mathbb{E}_{\mathbf{x} \sim P_{\mathbf{X}}} \left[ - \log \mathbb{P} \left\{ X_d | X_{d-s^\ell+1} = x_{d-s+1}, \ldots, X_{d-1} = x_{d-1} \right\} \right] \\
&= \mathbb{E}_{\mathbf{x} \sim P_{\mathbf{X}}} \left[ \log N(x_{d-s^\ell+1}, \ldots, x_{d-1}) \right],
\end{aligned}
\tag{10}
$$

where $N(x_{d-s^\ell+1}, \ldots, x_{d-1})$ denotes the number of possible values of the masked token depending on the effective context window. For $\ell = 0$, there is no restriction on the masked token value and this number equals $v$—the vocabulary size. For $\ell = 1$, we can determine the average $\bar{N}_1 := \mathbb{E}\left[ N(x_{d-s+1}, \ldots, x_{d-1}) \right]$ as follows. For each $s$-tuple $(x_{d-s+1}, \ldots, x_d)$ there is at least one value of the mask compatible with the other $s - 1$ symbols, i.e. $x_d$ itself. In addition, each of the remaining $v - 1$ values $\mu_d \neq x_d$ has a probability $f$ of being compatible with the context, coinciding with the probability that the $s$-tuple $(x_{d-s+1}, \ldots, \mu_d)$ is compatible with the production rules. This probability is given by $(mv - 1)$, i.e. the number of $s$-tuples compatible with the production rules except $(x_{d-s+1}, \ldots, x_d)$, over $(v^s - 1)$, i.e. the total number of $s$-tuples except $(x_{d-s+1}, \ldots, x_{\underline{d}})$. Therefore, $\bar{N}_1 = 1 + (v - 1)f = 1 + (v - 1)(mv - 1)/(v^s - 1)$. For $\ell > 1$, the average number $\bar{N}_\ell$ of symbols compatible with the context can be determined iteratively. The level-$\ell$ symbol generating the whole $s^\ell$-tuple can take any of the $v$ values, but the level-$(\ell - 1)$ symbol below it is now restricted to $\bar{N}_1$ values. By the previous argument, $\bar{N}_\ell = 1 + (v - 1)(m\bar{N}_{\ell-1} - 1)/(v^s - 1)$. Due to the concavity of the logarithm, we can bound the test loss of Eq. 10 with $\bar{\mathcal{L}}_\ell = \log \bar{N}_\ell$, i.e., after solving the recurrence relation and introducing the fraction of compatible $s$-tuples $f = m/v^{s-1}$.

$$
\begin{aligned}
\bar{\mathcal{L}}_\ell &= \log \left( \frac{v^s - v}{v^s - 1 - m(v - 1)} + \frac{(v^s - mv)(v - 1)}{v^s - 1 - m(v - 1)} \left( \frac{m(v - 1)}{v^s - 1} \right)^\ell \right) \\
&\xrightarrow{v, m \gg 1} \log \left( \frac{1}{1 - f} + v f^\ell \right),
\end{aligned}
\tag{11}
$$

**Naive strategy.** The simplest strategy to estimate $\mathbb{P}\left\{ X_d | X_{d-s^\ell+1} = x_{d-s^\ell+1}, \ldots, X_{d-1} = x_{d-1} \right\}$ is to count the empirical occurences of the $s^\ell$-dimensional subsequences of the input in the training set—the so-called $n$-gram language model with $n = s^\ell$. This estimation requires the training set to contain all the distinct subsequences of size $s^\ell$. Following the generative process, each of these subsequences occurs with probability given by that of the corresponding LCA symbol (the root of the subtree encased by the corresponding coloured box in Fig. 1, left), times the probability of the production rules that generate the subsequence from the LCA, $m^{-1} \times m^{-s} \times \cdots \times m^{-(s^\ell-1)} = m^{-(s^\ell-1)/(s-1)}$. The latter is exponentially small in the effective context length $t_\ell = s^\ell - 1$, hence the required sample size is exponentially large in $t_\ell$.

**Efficient strategy leveraging the hidden variables.** Using the hidden variables results in a much lower sample complexity. Indeed, due to the tree structure of PCFGs, the value of the last token is conditionally independent of most of the observable tokens when the hidden variables are given. For instance, looking at the tree in Fig. 1, left, the probability of the last token is independent of the pair $(\mu_5^{(0)}, \mu_6^{(0)})$ if the parent level-1 variable $\mu_3^{(1)}$ is given. In general, fixing a hidden symbol splits the tree into an *inside* (the subtree rooted at the hidden symbol) and an *outside* (the rest of the tree) that are conditionally independent. As a result, the minimal set of variables that the $s^\ell$-gram probability depends on consist of $s - 1$ observable tokens (those in the same patch as the last token) and $(s - 1)(\ell - 1)$ hidden variables ($(s - 1)$ for each level below the LCA of the context window). The probability of any such set of variables is given by the LCA probability times $m^{-\ell}$. The resulting sample complexity grows exponentially with $\ell$, or as a power of the effective context length $t_\ell$.

**Reconstruction of the hidden variables.** We now argue that, as shown [20] in the context of classification, the hidden variables can be represented via the correlations between tokens. Consider, for instance, the pair $(\mu_5^{(0)}, \mu_6^{(0)})$ in Fig. 1, left. Because of the aforementioned conditional independence, the correlation between any such pair and the last token depends only on the level-1 hidden variable $\mu_3^{(1)}$. Thus, pairs displaying the same correlations can be grouped as descendants of the same hidden variable. This strategy requires enough training data to resolve correlations between the masked token and the adjacent $s$-tuples of observable tokens. As shown in App. F, replacing an observable token

with a whole $s$-tuple reduces correlation plateaus and sampling noise by the same factor. Therefore, the condition for the resolution of correlations with the nearest $s$-tuples is given by Eq. 9 with $\ell = 2$, implying $P > P_2 = vm^3$. By iterating this argument we get a sequence of sample complexities $P_\ell$ that allow for resolving correlations between the masked token and $s$-tuples up to distance $t = s^\ell - 1$,

$$P_\ell = (v^2 \tilde{C}^{(\ell)})^{-1} = vm^{2\ell-1} \left(1 - \frac{m}{v^{s-1}}\right)^{-1}. \tag{12}$$

For instance, in the case illustrated in Fig. 1, left, the correlations of the pairs $(\mu_1^{(0)}, \mu_2^{(0)})$ and $(\mu_3^{(0)}, \mu_4^{(0)})$ with the masked token can be used to reconstruct the pair of hidden symbols $(\mu_1^{(1)}, \mu_2^{(1)})$. The hidden symbols have a higher correlation with the masked token than their children. Hence, as in the case of classification [20], a training set large enough to resolve correlations between observable and masked tokens also allows for resolving correlations of the masked token with the hidden symbols. These correlations yield a representation of higher-level hidden symbols (e.g. $\mu_1^{(2)}$ for $(\mu_1^{(1)}, \mu_2^{(1)})$ in the figure), which, in turn, enables the reconstruction of $\mathbb{P}\{X_d | X_{d-s^\ell+1} = x_{d-s+1}, \ldots, X_{d-1} = x_{d-1}\}$ via the efficient strategy. As $\ell$ increases, the sample complexity of Eq. 12 grow faster than $m^\ell$, but still polynomially in the effective context length $t_\ell$.

**Scaling law of the RHM.** After solving Eq. 12 for $\ell$ as a function of $P$, we can use Eq. 11 to derive the *scaling law* for the behaviour of the loss steps as a function of the training set size $P$. Neglecting all the factors that do not depend on $\ell$, Eq. 12 implies $\ell \approx \log P/(2 \log m)$. Thus, from Eq. 11,

$$\bar{\mathcal{L}}(P) + \log(1 - f) \approx \log\left(1 + v(1-f)e^{\frac{\log f \log P}{2 \log m}}\right). \tag{13}$$

Notice that $f < 1$, thus $\log f < 0$. Therefore, Eq. 11 implies an early logarithmic decay as long as $|\log f| \log P \ll 2 \log m \log(v(1-f))$. For larger $P$, the expansion $\log(1+x) \simeq x$ recovers the ubiquitous power-law decay $P^{-\alpha}$ [17], with exponent $\alpha = \log f/(2 \log m)$. Notice that the power-law scaling is caused by the sequence of steps associated with the emergence of the hidden variables representation. Therefore, this picture unifies the emergence and scaling paradigms.

## 4.2 Comparison with empirical learning curves

Fig. 2, left, compares the learning curves of deep transformers (details of the architectures in subsection A.2) with the sample complexities $P_\ell$ of Eq. 12 (vertical dashed lines in the figure) and the test loss upper bounds $\bar{\mathcal{L}}_\ell$ of Eq. 11 (horizontal dashed lines), showing good qualitative agreement. Additional experiments that support the quantitative scaling of the sample complexities $P_1$ and $P_2$ with $m$ are shown in App. G. Fig. 2, right, shows the learning curves of models trained on a reduced context window. In this setting, our description correctly predicts the saturation of the loss due to the finite context window size $t$: with $t = s^\ell - 1$, the model can only learn the level-$\ell$ hidden variable above the masked token, thus follow only the first $\ell$ of the $L$ steps of Eq. 11.

Let us remark that, as shown in App. G, the learning curves are qualitatively similar for CNNs, despite a noticeable quantitative dependence on architecture and context size $t$. These differences are not captured by the analysis of subsection 4.1, although, in some cases, they can be rationalised using results from the theory of shallow neural networks. We discuss these aspects in detail in App. G.

## 4.3 Emergence of hierarchical representations of the data structure

We now study the hidden representations of models trained on RHM data to show that, as the training set size increases, these representations encode for deeper hidden variables. More specifically, we show that certain representations depend only on specific, high-level hidden variables of a datum's tree structure, thus becoming insensitive to the entire subtree emanating from this hidden variable. For the sake of interpretability, we consider deep convolutional networks (CNNs) with architecture matched to the data structure, represented schematically in the graphs on the right of Fig. 3 (further details in subsection A.1). To probe representations we introduce two sets of transformations. Given a datum and the associated tree ( Fig. 1, left), consider the $i$-th level-$\ell$ symbol $\mu_i^{(\ell)}$: $\mathcal{S}_{\ell,i}$ replaces it with another one randomly chosen from the vocabulary, whereas $\mathcal{R}_{\ell,i}$ resets the choice of the production rule emanating from $\mu_i^{(\ell)}$. Both transformations alter the subtree originating from $\mu_i^{(\ell)}$ (e.g. the subtree within the orange box of Fig. 2, left for $\ell = 2$ and $i = 2$), affecting $s^\ell$ observable tokens. However, $\mathcal{R}_{\ell,i}$ preserves the hidden symbols that generated the subtree. Therefore, a hidden representation that encodes only the $i$-th level-$\ell$ hidden symbol will be invariant to $\mathcal{R}_{\ell,i}$ but not to $\mathcal{S}_{\ell,i}$.

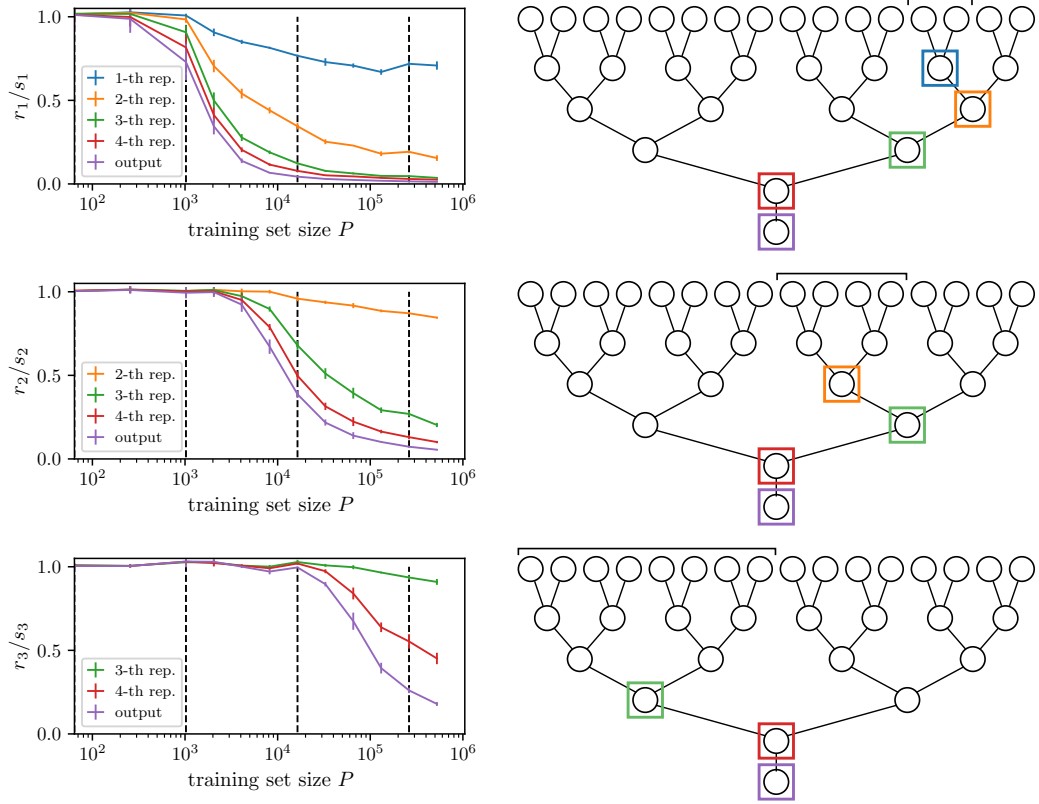

Figure 3: Relative sensitivity $r_\ell/s_\ell$ of the representation of trained depth-4 CNNs (sketched on the right panels) for input transformations (the affected tokens are indicated by the black horizontal segments on the right panels) corresponding to resetting the production rule emanating from a given level-$\ell$ variable ($\ell = 1, 2, 3$ for top, centre and bottom), as a function of training set size $P$. Colours represent the layer of the representation, as indicated in the key and by the squares on the right panels. The CNNs are trained on RHM data with $L = 4$, $s = 2$, $v = 16$, $m = 4$. Vertical dashed lines mark the sample complexities $P_\ell$ of Eq. 12. The drop of the curves from $\simeq 1$ to $\simeq 0$ around $P_\ell$ signals that the trained representations only encode for the relevant level-$\ell$ symbol when $P > P_\ell$.

We define hidden representations $h_\ell(x)$ (hidden nodes of the network's graphs in Fig. 3) as the sequence of pre-activations in a given layer $\ell$ (depth of the node in the tree), standardised over the dataset (i.e. centred around the mean and scaled by the standard deviation). For CNNs, representations carry a spatial index $j = 1, \ldots, s^{L-\ell}$ (horizontal position of the node within the layer) and a channel index. We measure the sensitivity to $\mathcal{R}$ or $\mathcal{S}$ via the cosine similarity between original and transformed representations, i.e.

$$r_{\ell,i}(h) = \mathbb{E}_{x \sim P_\mathbf{X}}\left[h_{\ell',j}(x) \cdot h_{\ell',j}(\mathcal{R}_{\ell,i}x)\right], \, s_{\ell,i}(h) = \mathbb{E}_{x \sim P_\mathbf{X}}\left[h_{\ell',j}(x) \cdot h_{\ell',j}(\mathcal{S}_{\ell,i}x)\right], \qquad (14)$$

where the $\cdot$ symbol denotes the scalar product over the channels. In order to leave the masked token unaltered, we always apply the transformations to the penultimate hidden symbol of the level, i.e. $i = s^{L-\ell} - 1$. Hence, from now on, we omit the spatial index $i$. The left column of Fig. 3 reports the ratio $r_\ell/s_\ell$ for the hidden representations of a deep CNN trained on RHM data. Each row refers to the level of the data transformations. The group of observable tokens affected by the transformation is highlighted by horizontal square brackets in the right panels. The drop of $r_\ell/s_\ell$ from $\approx 1$ to $\approx 0$ signals that a representation depends on the corresponding level-$\ell$ hidden variable, but not on the other variables in the associated subtree. [3] These drops occur at the same training set sizes $P_\ell$ as the

---

[3]Notice that only the representations with $\ell' > \ell$ can become invariant, which is due to the fact the production rules are not linearly separable. Let us focus on the first level: the corresponding $s$-dimensional patch of the input can take $mv$ distinct values—$m$ for each of the $v$ level-2 features. Invariance of a linear transformation is equivalent to the following set of constraints: for each level-2 features $\mu$, and $\boldsymbol{x}_{1,i}$ encoding for one of the $m$ level-1 representations generated by $\mu$, $\boldsymbol{w} \cdot \boldsymbol{x}_{1,i} = c_\mu$. Since $c_\mu$ is an arbitrary constant, there are $v \times (m-1)$ constraints for the $v \times s$ components of $\boldsymbol{w}$, which cannot be satisfied in general unless $m \leq (s+1)$.

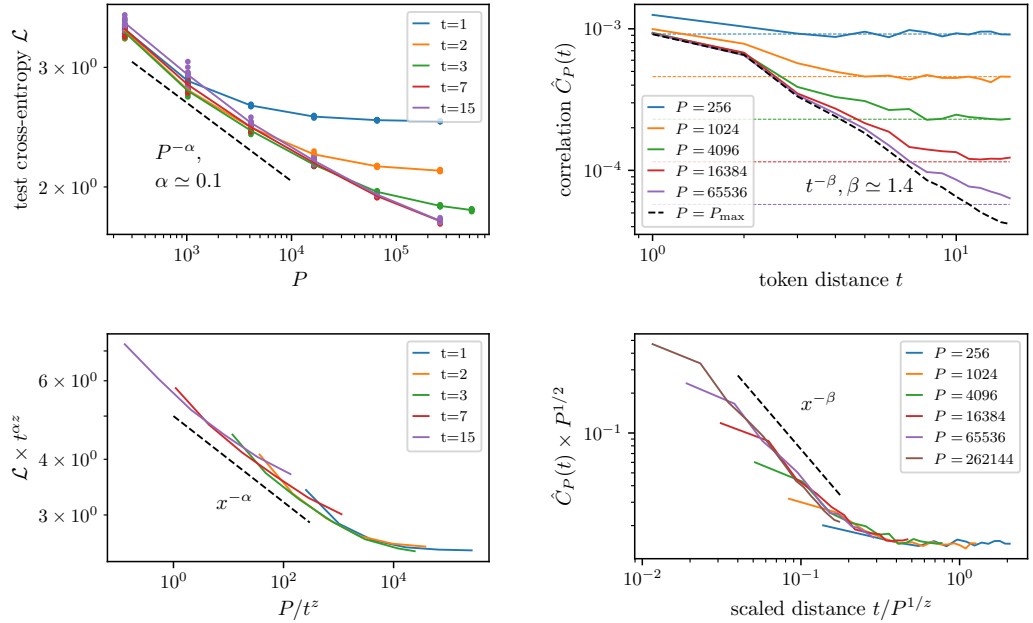

Figure 4: **Top, Left:** Test losses of 3-layers transformers trained on $(t+1)$-characters blocks of the tiny-Shakespeare dataset [38] ($t$ as in the key). The saturation of the loss to some $t$-dependent value indicates that performance improves with $P$ because the model can use information from a larger context window. **Top, Right:** Empirical estimates $\hat{C}_P(t)$ for different training set sizes $P$ as in the key. The curves initially follow the true correlation $\tilde{C}(t)$ (black dashed), but then saturate due to the sampling noise (coloured dashed). **Bottom, Right:** The empirical curves $\hat{C}_P(t)$ collapse when rescaling correlations by the sampling noise size $P^{-1/2}$ and $t$ by the characteristic distance $t^*(P) \sim P^{1/z}$, with $z \simeq 2.8$. **Bottom, Left:** As predicted by our conjecture, the losses collapse when rescaled according to Eq. 16 with the same $z$ as the correlation functions.

test loss steps, highlighted in the figures with vertical dashed lines. This result confirms that, as $P$ increases, trained models learn a deeper representation of the tree structure of the data.

## 5   Conjecture and test on real language data

We conjecture that the relationship between training set size, correlations and effective context window holds beyond our synthetic dataset.

**Conjecture:** *"If the token correlation function decays with the token distance, then a language model trained to predict the next token from a training set of $P$ examples can only extract relevant information from an effective context window of $P$-dependent size $t^*(P)$."*

We test this conjecture in two datasets: a selection of lines from Shakespeare's plays [38] and a collection of articles from English Wikipedia [39]. For both datasets we adopt a character-level tokenisation, resulting in over $10^6$ tokens. We then extract sequences of $t$ consecutive tokens and train BERT-like deep transformers in the setup of section 2—further details of architecture and training are in subsection A.3. The results of our test are reported in Fig. 4 for Shakespeare and Fig. 5 of App. B for Wikipedia. First, with a large context window, the test loss follows the empirical scaling law $\mathcal{L} \sim P^{-\alpha}$ (top left panel). However, the learning curve levels off at some characteristic scale $\bar{P}$ that grows with the size $t$ of the context window. This phenomenon can be explained via the correlation function, which decays as a power of the distance $\tilde{C}(t) \sim t^{-\beta}$, with $\beta \simeq 1.4$ [4] (top right panel). Empirical estimates $\hat{C}_P(t)$ saturate when reaching the sampling noise scale $\sim P^{-1/2}$: following the analysis of section 3, this behaviour results in an effective context window size $t^*(P)$, given by the

---

[4]Let us remark that, while the exponent depends on the corpus and the choice of tokenisation, power-law decays are observed empirically also for syllables [40], words [41] and part-of-speech tags [42].

value of $t$ where the correlation function $\tilde{C}(t) \sim t^{-\beta}$ intersects the sampling noise scale $\sim P^{-1/2}$,

$$(t^*)^{-\beta} \sim P^{-1/2} \Rightarrow t^*(P) \sim P^{1/z}, \quad \text{with } z = 2\beta \simeq 2.8. \tag{15}$$

As a result, the empirical correlation functions with varying $P$ collapse when rescaling $\hat{C}$ by the sampling noise and the distance by $t^*(P)$ (bottom right panel).

By inverting $t^*(P)$ we get a characteristic training set size $P^*(t)$ where the training set allows for resolving correlations at all distances $t' < t$, $P^*(t) \sim t^z$. Paired with the empirical power-law scaling with $P$, this result leads to the following context-dependent scaling hypothesis for the test loss:

$$\mathcal{L}(P, t) = t^{-\alpha z} f\left(\frac{P}{t^z}\right), \tag{16}$$

with $f(x) \sim x^{-\alpha}$ for $x \ll 1$ and constant for $x \gg 1$. In particular, Eq. 16 implies that the behaviour of the empirical correlation functions predicts the saturation of the loss decay. The collapse reported in the bottom left panels of Fig. 4 and Fig. 5 quantitatively confirms Eq. 16 and our previous conjecture.

## 6 Conclusions

We proposed a conceptual framework for understanding the performance-vs.-data scaling laws of language models trained for next-token prediction. In our picture, increasing the number of data allows for the resolution of a longer range of correlations. These correlations, in turn, can be exploited to build a hierarchical representation of the data structure, the longer the range the deeper the representation. For our synthetic hierarchical data, the emergence of deeper representation results in a series of steps in the next-token prediction performance. These steps conspire to determine the scaling law, whose exponent depends on the dataset structure. This scenario is consistent with the empirical phenomenology of language models, including both the emergence of skills at specific training set sizes [18, 43, 44, 45] and the steady improvement of overall performance [17]. To the best of our knowledge, this is the first theoretical description of scaling laws in a setting where learning data features is crucial, whereas previous works focused on kernel limits [46, 47, 48, 49, 50].

Furthermore, our analysis predicts a fundamental relationship between the effective context window captured by a language model trained with a finite training set and the decay of token-token correlations, which we confirmed empirically on two examples of text data. This finding suggests that the exponents entering scaling laws are influenced by the intrinsic properties of the data. On the one hand, our predictions can be tested on state-of-the-art LLMs trained on larger datasets. On the other hand, our framework can be extended to shed light on other aspects of scaling laws of high practical relevance, such as the role of the number of parameters and the behaviour of performance when the model size and the number of data are optimised under a fixed compute budget.

**Limitations.** Our hierarchical model of data is limited by the context-free structure of the rules, which describes most, but not all, of the syntactic forms observed in natural languages [51]. Understanding the role of context-sensitive structures in language acquisition is a promising avenue for future research. In addition, the RHM assumes a fixed geometry of the data tree and the uniform probability and unambiguity of the production rules. These assumptions are not satisfied by real text data and are responsible for the stepwise behaviour of correlations in our model. Relaxing these constraints while keeping the large-scale, power-law decay of correlations with the distance, which is indeed observed in real data, could broaden the scope of our conceptual framework. On the technical side, there is no proof of the connection between the strategy illustrated in subsection 4.1 and the sample complexity of deep neural networks trained with gradient descent and its variants. Such a proof would require a formal description of the dynamics of deep networks trained on hierarchical data, which is beyond the scope of the present paper. This description would also capture the discrepancies between different architectures presented in App. G, making it a valuable direction for future work.

## Acknowledgments and Disclosure of Funding

We thank Kai Nakaishi for pointing references [52, 27] to us; and Allan Raventós for feedback on an earlier version of the manuscript. We also thank Antonio Sclocchi, Alessandro Favero and Umberto Tomasini for helpful discussions and feedback on the manuscript. This work was supported by a grant from the Simons Foundation (# 454953 Matthieu Wyart).

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

# A  Details of the experiments

Our experiments on RHM data consider both Deep CNNs tailored to the RHM structure and simple transformers made by stacking standard Multi-Head Attention layers. Our experiments on the tiny-Shakespeare and WikiText-103 datasets consider deep, encoder-only transformers, where Multi-Head Attention layers are interspersed with residual connections, layer normalization and two-layer perceptrons. All our experiments were performed on a cluster of NVIDIA V100 PCIe 32 GB GPUs (2×7TFLOPS). Single experiments require up to 20 GPU hours for the largest models ($\approx 10 \times 10^6$) with the largest training set sizes ($\approx 4 \times 10^6$), with an estimated total (including hyperparameter tuning) of $6,000$ GPU hours. We provide architecture and training details below.

## A.1  Deep CNNs (RHM)

The deep CNNs we consider are made by stacking standard convolutional layers. To tailor the network to the structure of the data generative model, we fix both the stride and filter size of these layers to $s$. Since each layer reduces the spatial dimensionality by a factor $s$, the input size $d$ must be an integer power of $s$ and the CNNs depth equals $\log d / \log s$.

We use the Rectified Linear Unit (ReLU) $\sigma(x) = \max(0, x)$ as activation function, set the number of channels to $H$ for each layer, and consider the maximal update parametrization [53], where the weights are initialised as random gaussian variables with zero mean and unit variance, all the hidden layers but the last are rescaled by a factor $H^{-1/2}$, whereas the last is rescaled by $H^{-1}$. This factor causes the output at initialization to vanish as $H$ grows, which induces representation learning even in the $H \to \infty$ limit. In practice, $H$ is set to 256 for Fig. 3, 512 for Fig. 6, left and Fig. 9, 1024 for Fig. 6, right, 512 for Fig. 7 and Fig. 8. Increasing the number of channels does not affect any of the results presented in the paper.

Deep CNNs are trained with SGD, with the learning rate set to $H$ to compensate for the factor of $H^{-1}$. A cosine annealing scheduler reduces the learning rate by 10 within the first 100 training epochs. The batch size is set to the minimal size allowing convergence, where we define convergence as the training cross-entropy loss reaching a threshold value of $10^{-3}$. We use a validation set of size $2^{15}$ to select the model with the best validation loss over the training trajectory.

## A.2  Multi-layer self-attention (RHM)

The deep Transformers that we train on RHM data are made by stacking standard Multi-Head Attention layers [54], without any residuals, layer normalization and multi-layer perceptron in between. We found that the removed components do not affect the model's performance on data generated from the RHM. Each layer has the same number of heads $n_h$ and embedding dimension $d_{\mathrm{emb}} = n_h \times v$, with $v$ the vocabulary size. The input dimension is adapted to the embedding dimension via a learnable linear projection, to which we add learnable positional encodings. The choice of $n_h$ follows two principles: the model should be large enough for the training loss to reach a threshold value of $10^{-3}$ and changing $n_h$ should not affect performance beyond the fluctuations due to the model initialisations. Specifically, we set $n_h = 16$ and notice no significant change in performance up to $n_h = 64$. Also scaling $d_{\mathrm{emb}}$ up to $4n_h \times v$ does not impact performance.

Multi-layer self-attention networks are trained with the Adam optimizer, with a warmup scheduler bringing the learning rate to $10^{-2}$ within the first 10 training epochs. As for CNNs, the batch size is set to the lowest value allowing for convergence.

## A.3  Encoder-only Transformer (tiny-Shakespeare and WikiText-103)

The architectures trained on real text data have the same structure as BERT [8], that is they include additional token-wise two-layer perceptions (MLPs) after each self-attention layer, together with layer normalization operations before the attention layer and the MLP and residual connections. The training procedure is the same as above.

For tiny-Shakespeare, we set the number of heads to $n_h = 8$, the embedding dimension to $d_e = 256$, the size of the MLP hidden layer to $4d_e$, and the number of layers to 3. For WikiText-103, we set $n_h = 8$, $d_e = 512$, and the number of layers to 6. Increasing the number of layers or the number of heads does not affect the results presented in the paper.

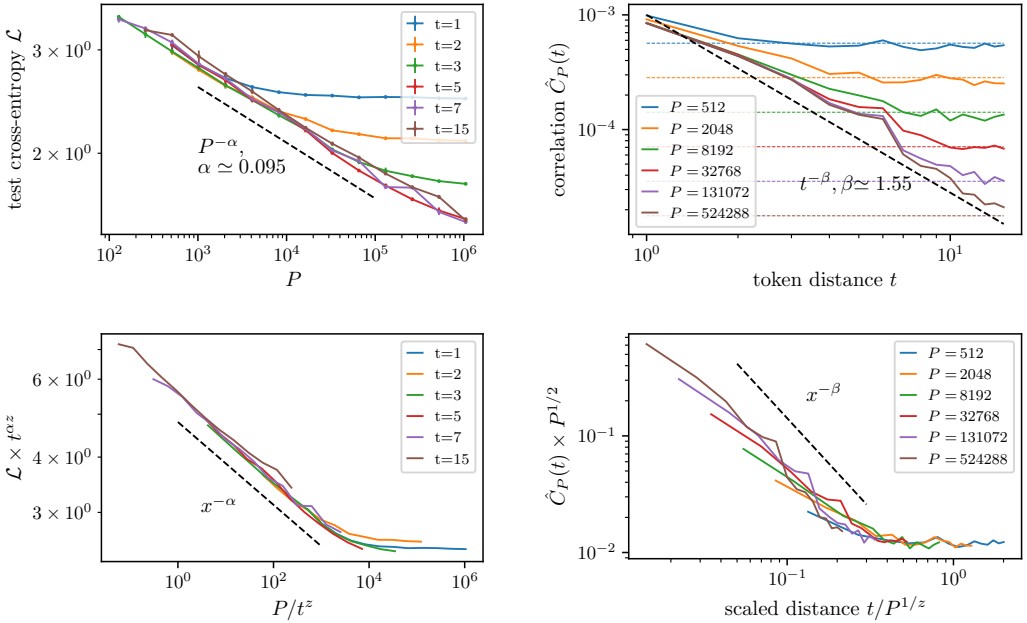

Figure 5: **Top, Left:** Test losses of 6-layers transformers trained on $(t+1)$-characters blocks of the WikiText-103 [39] ($t$ as in the key). As in Fig. 4, the loss saturates to some $t$-dependent value after reaching a characteristic training set size. **Top, Right:** Empirical correlation functions $\hat{C}_P(t)$ with $P$ as in the key, showing saturation for large $t$ due to the sampling noise (coloured dashed). **Bottom, Right:** Collapse of the empirical curves $\hat{C}_P(t)$ is achieved when rescaling the correlations by the sampling noise size $P^{-1/2}$ and $t$ by the characteristic distance $t^*(P) \sim P^{1/z}$, with $z = 2\beta \simeq 3.1$. **Bottom, Left:** As predicted by our conjecture, the losses collapse when rescaled according to Eq. 16 with the same $z$ as the correlation functions and $\alpha \simeq 0.095$.

## B  Loss saturation and correlations for WikiText-103

In this section, we report the results of the test of our conjecture for the WikiText-103 dataset of [39]. The original dataset was preprocessed to remove the article's headers and subheaders. The results are summarised in Fig. 5, which displays the same measures as Fig. 4 and, as Fig. 4, confirms our conjecture.

## C  Statistics of the RHM data

For each token $i = 0, \ldots, d-1$, single-token probabilities can be written as products of probabilities over the single production rules,

$$\mathbb{P}\{X_i = \mu\} = \sum_{\mu_1, \ldots, \mu_L = 1}^{v} p_{i_1}^{(1)}(\mu|\mu_1) \ldots p_{i_L}^{(L)}(\mu_{L-1}|\mu_L) p^{(L+1)}(\mu_L), \tag{17}$$

where

  (i) the indices $i_L, \ldots, i_L$ are such that $i_L \ldots i_1$ equals the $s$-ary representation of $i$, with $i_\ell = 0, \ldots, s-1$, and 0's added to ensure that the representation always consists of $L$ indices. In other words, the multi-index $i_L, \ldots, i_L$ uniquely identifies the path linking the root of the tree to the $i$-th leaf.

 (ii) $p_{i_\ell}^{(\ell)}(\mu_{\ell-1}|\mu_\ell)$ denotes the probability of choosing, among the available production rules starting from $\mu_\ell$, one that has the symbol $\mu_{\ell-1}$ on the $i_\ell$-th position of the right-hand size.

(iii) $p^{(L)}(\mu_L)$ denotes the probability of selecting the symbol $\mu_L$ as the root ($1/v$ for our model).

These decompositions arise naturally due to the connection between probabilistic context-free grammars and Markov processes. For the joint probability of two tokens $i$ and $j$ at distance $t = |j-i|$,

with $s^{\ell-1} < t < s^\ell - 1$ such that the lowest common ancestor (LCA) is a level-$\ell$ hidden symbol, and $i_{\ell+1}$ denoting the position of the LCA within its level,

$$\mathbb{P}\{X_i = \mu, X_j = \nu\} = \sum_{\substack{\mu_1,\dots,\mu_\ell=1 \\ \nu_1,\dots,\nu_{\ell-1}=1}}^{v} \sum_{\mu_\ell=1}^{v} p_{i_1}^{(1)}(\mu|\mu_1) p_{j_1}^{(1)}(\nu|\nu_1) \dots p_{i_\ell,j_\ell}^{(\ell)}(\mu_{\ell-1},\nu_{\ell-1}|\mu_\ell) p_{i_{\ell+1}}^{(\ell+1)}(\mu_\ell).$$

(18)

Both in Eq. 17 and Eq. 38 simplify when replacing $\mu$ with a whole $s$-tuple of observable symbols $\boldsymbol{\mu}_j = (\mu_{1+(j-1)s}, \dots, \mu_{js})$ for some $j = 1, \dots, s^{L-1}$. The simplification arises because the level-1 rule probability $p^{(1)}(\boldsymbol{\mu}_j|\mu_1)$, is uniform and equal to $1/m$ if the production rule $\mu_1 \to \boldsymbol{\mu}_j$ exists, 0 otherwise. Then, the sum over $\mu_1$ selects the only level-1 symbol that generates the tuple $\boldsymbol{\mu}_j$. As a result, one is left with a probability represented by a smaller tree, where the $s$ leaves representing $\boldsymbol{\mu}_j$ are pruned, and an additional factor of $1/m$.

## C.1 Statistics of production rules

For each set of production rules, we call $N_i^{(\ell)}(\mu_{\ell-1}; \mu_\ell)$ the number of occurrences of the level-$\ell-1$ feature $\mu_{\ell-1}$ in the $i$-th position of the right-hand side for all the production rules emanating from the level-$\ell$ feature $\mu_\ell$. In our generative model, there are $m$ production rules emanating from a given symbol. The rule to follow when generating a datum is chosen uniformly at random among these $m$. Hence,

$$p_i^{(\ell)}(\mu_{\ell-1}|\mu_\ell) = \frac{1}{m} N_i^{(\ell)}(\mu_{\ell-1}; \mu_\ell).$$

(19)

For the sake of clarity, let us omit the level index in the following paragraph. The probability of $N_i(\mu; \nu)$ over different realisations of the set of production rules is that of the number of successes when drawing $m$ times (number of $s$-tuples associated with the high-level feature $\nu$) without replacement from a pool of $v^s$ (total number of $s$-tuples with vocabulary size $v$) objects where only $v^{s-1}$ (number of $s$-tuples displaying feature $\mu$ in position $i$) leads to success:

$$\mathbb{P}\{N_i(\mu_0; \mu_1) = k\}_{RHM} = \binom{v^{s-1}}{k}\binom{v^s - v^{s-1}}{m-k} \Big/ \binom{v^s}{m} = \mathrm{Hg}_{m,v^{s-1},v^s}(k),$$

(20)

where $\mathrm{Hg}_{n,K,N}$ denotes a Hypergeometric distribution with population size $N$, $K$ success states, and $n$ draws. The mean and variance with respect to realisations of the RHM (denoted with $\langle . \rangle$ to avoid confusion with data averages $\mathbb{E}[.]$) are

$$\langle N \rangle = m\frac{v^{s-1}}{v^s} = \frac{m}{v}, \quad \sigma_N^2 := \langle (N - \langle N \rangle)^2 \rangle = m\frac{v^{s-1}}{v^s}\frac{v^s - v^{s-1}}{v^s}\frac{v^s - m}{v^s - 1} = \frac{m}{v}\frac{v-1}{v}\frac{v^s - m}{v^s - 1}.$$

(21)

For $v \gg 1$, the variance converges to $m/v$ ($m \le v^{s-1}$ with $s$ fixed, thus $v^s - m \to v^s$).

Equations (19) to (21) easily generalise to the case where $\mu_0$ represents a group of $s' \le s$ low-level features (instead of a single low-level feature). With $\boldsymbol{\mu}_0$ denoting a $s'$-tuple of features and $\boldsymbol{i}$ the $s'$-tuple of associated spatial indices,

$$\mathbb{P}\{N_{\boldsymbol{i}}(\boldsymbol{\mu}_0; \mu_1) = k\}_{RHM} = \binom{v^{s-s'}}{k}\binom{v^s - v^{s-s'}}{m-k} \Big/ \binom{v^s}{m},$$

(22)

resulting in

$$\langle N_{s'} \rangle = m\frac{v^{s-s'}}{v^s} = \frac{m}{v^{s'}}, \quad \sigma_{N_{s'}}^2 := \langle (N_{s'} - \langle N_{s'} \rangle)^2 \rangle = \frac{m}{v^{s'}}\frac{v^{s'} - 1}{v^{s'}}\frac{v^s - m}{v^s - 1} \xrightarrow{v \gg 1} \frac{m}{v^{s'}}.$$

(23)

## C.2 Statistics via splitting

An alternative way to obtain all statistics is by writing the level-$\ell$ probabilities as sums over the production rules,

$$p_i^{(\ell)}(\mu_{\ell-1}|\mu_\ell) = \frac{1}{m} \sum_{\psi=1}^{m} \delta(r_{\psi,i}(\mu_\ell), \mu_{\ell-1}),$$

(24)

where $r_{\psi,i}(\mu_1)$ denotes the $i$-th element of the right-hand side of the $\psi$-th production rule emanating from $\mu_1$. Eq. 24 generalises immediately to the case where $i$ and $\mu_{\ell-1}$ are $s'$-tuples with $s' \leq s$. Using

$$\langle \delta(r_{\psi,i}(\mu_1), \mu) \rangle = \mathbb{P}\left\{ \mu_1 \xrightarrow{\psi,i} \mu \right\}_{RHM} = \mathrm{Hg}_{1,v^{s-s'},v^s}(1) = \frac{1}{v^{s'}}, \tag{25}$$

where $\mu_1 \xrightarrow{\psi,i} \mu$ denotes the event that the $i$-th element of the right-hand side of the $\psi$-th production rule emanating from $\mu_1$ coincides with $\mu$, we can compute all one-point averages. In addition, for $(\nu_1, \phi, j, \nu) \neq (\mu_1, \psi, i, \mu)$,

$$\langle \delta(r_{\psi,i}(\mu_1), \mu)\delta(r_{\phi,j}(\nu_1), \nu) \rangle = \mathbb{P}\left\{ \mu_1 \xrightarrow{\psi,i} \mu \right\}_{RHM} \mathbb{P}\left\{ \nu_1 \xrightarrow{\phi,j} \nu \,\middle|\, \mu_1 \xrightarrow{\psi,i} \mu \right\}_{RHM}, \tag{26}$$

where

$$\mathbb{P}\left\{ \nu_1 \xrightarrow{\phi,j} \nu \,\middle|\, \mu_1 \xrightarrow{\psi,i} \mu \right\}_{RHM} = \begin{cases} \mathbb{P}\left\{ \nu_1 \xrightarrow{\phi,j} \nu \right\}_{RHM} = v^{-1}, \text{if } i \neq j \\[2mm] \qquad\qquad 0, \text{if } i = j, \nu_1 = \mu_1, \phi = \psi, \mu \neq \nu, \\[2mm] \mathrm{Hg}_{1,v^{s-1}-1,v^s-1}(1) = \dfrac{v^{s-1}-1}{v^s-1}, \text{if } i = j, \nu_1 = \mu_1, \mu = \nu, \phi \neq \psi, \\[2mm] \mathrm{Hg}_{1,v^{s-1},v^s-1}(1) = \dfrac{v^{s-1}}{v^s-1}, \text{if } i = j, \nu_1 = \mu_1, \mu \neq \nu, \phi \neq \psi, \\[2mm] \mathrm{Hg}_{1,v^{s-1}-1,v^s-1}(1) = \dfrac{v^{s-1}-1}{v^s-1}, \text{if } i = j, \nu_1 \neq \mu_1, \mu = \nu, \\[2mm] \mathrm{Hg}_{1,v^{s-1},v^s-1}(1) = \dfrac{v^{s-1}}{v^s-1}, \text{if } i = j, \nu_1 \neq \mu_1, \mu \neq \nu. \end{cases} \tag{27}$$

Notice that, once the right-hand side of the rules ($\mu$ and $\nu$) are fixed, the conditional probability can only attain two distinct values: one for $\mu_1 = \nu_1$ and $\psi = \phi$, and one for the other cases. Then, it is convenient to define a 'rule index' $\psi$ that comprises both the starting high-level feature and the chosen production rule. This index runs in $(1, \ldots, mv)$. With these formulas, one can get all the joint statistics of the rules. For instance (omitting the $RHM$ subscript on $\mathbb{P}$ to ease notation),

$$\langle p_i(\mu_0|\mu_1)p_i(\mu_0|\mu_1) \rangle = \frac{1}{m^2} \sum_{\psi_1,\psi_2=1}^{m} \mathbb{P}\left\{ \mu_1 \xrightarrow{\psi_1,i} \mu_0; \mu_1 \xrightarrow{\psi_2,i} \mu_0 \right\}$$

$$= \frac{1}{m^2} \sum_{\psi_1=\psi_2} \mathbb{P}\left\{ \mu_1 \xrightarrow{\psi_1,i} \mu_0 \right\}$$

$$+ \frac{1}{m^2} \sum_{\psi_1,\psi_2 \neq \psi_1} \mathbb{P}\left\{ \mu_1 \xrightarrow{\psi_1,i} \mu_0 \right\} \mathbb{P}\left\{ \mu_1 \xrightarrow{\psi_1,i} \mu_0 \,\middle|\, \mu_1 \xrightarrow{\psi_1,i} \mu_0 \right\}$$

$$= \frac{1}{m^2}\left[ \frac{m}{v} + \frac{m(m-1)}{v}\frac{v^{s-1}-1}{v^s-1} \right], \tag{28}$$

hence

$$\sigma_p^2 = \langle (p_i(\mu_0|\mu_1) - \langle p \rangle)(p_i(\mu_0|\mu_1) - \langle p \rangle) \rangle = \langle p_i(\mu_0|\mu_1)p_i(\mu_0|\mu_1) \rangle - \left(\frac{1}{v}\right)^2 = \frac{1}{mv}\frac{v-1}{v}\frac{v^s-m}{v^s-1}, \tag{29}$$

equivalent to dividing $\sigma_N^2$ from Eq. 21 by $m^2$. Analogously, with $\nu_0 \neq \mu_0$,

$$\langle p_i(\mu_0|\mu_1)p_i(\nu_0|\mu_1) \rangle = \frac{1}{m^2} \sum_{\psi_1,\psi_2=1}^{m} \mathbb{P}\left\{ \mu_1 \xrightarrow{\psi_1,i} \mu_0; \mu_1 \xrightarrow{\psi_2,i} \nu_0 \right\}$$

$$= \frac{1}{m^2} \sum_{\psi_1,\psi_2 \neq \psi_1} \mathbb{P}\left\{ \mu_1 \xrightarrow{\psi_1,i} \mu_0 \right\} \mathbb{P}\left\{ \mu_1 \xrightarrow{\psi_1,i} \nu_0 \,\middle|\, \mu_1 \xrightarrow{\psi_1,i} \mu_0 \right\}$$

$$= \frac{1}{m^2}\left[ \frac{m(m-1)}{v}\frac{v^{s-1}}{v^s-1} \right], \tag{30}$$

thus

$$c_p = \langle (p_i(\mu_0|\mu_1) - \langle p \rangle)(p_i(\nu_0|\mu_1) - \langle p \rangle) \rangle = \langle p_i(\mu_0|\mu_1)p_i(\nu_0|\mu_1) \rangle - \left(\frac{1}{v}\right)^2 = -\frac{1}{mv^2}\frac{v^s - m}{v^s - 1}.$$
(31)

Notice that $c_p = \sigma_p^2/(v-1)$ in agreement with the constraint $\sum_{\mu_0} p_i(\mu_0|\mu_1) = 1$. Indeed, for any finite sequence of identically distributed random variables $X_\mu$ with a constraint on the sum, $\sum_\mu X_\mu = C$ for some constant $C$,

$$\sum_{\mu=1}^{v} X_\mu = C \Rightarrow \sum_{\mu=1}^{v}(X_\mu - \langle X_\mu \rangle) = 0 \Rightarrow$$

$$(X_\nu - \langle X_\nu \rangle)\sum_{\mu=1}^{v}(X_\mu - \langle X_\mu \rangle) = 0 \Rightarrow$$

$$\sum_{\mu=1}^{v}\langle (X_\nu - \langle X_\nu \rangle)(X_\mu - \langle X_\mu \rangle) \rangle = 0 \Rightarrow$$

$$\langle (X_\mu - \langle X_\mu \rangle)^2 \rangle + (v-1)\langle (X_\mu - \langle X_\mu \rangle)(X_\nu - \langle X_\nu \rangle) \rangle = 0,$$
(32)

where, in the last line, we used the identically distributed variables hypothesis to replace the sum over $\mu \neq \nu$ with the factor $(v-1)$.

In addition, with $\nu_1 \neq \mu_1$,

$$\langle p_i(\mu_0|\mu_1)p_i(\mu_0|\nu_1) \rangle = \frac{1}{m^2}\sum_{\psi_1,\psi_2=1}^{m}\mathbb{P}\left\{\mu_1 \xrightarrow{\psi_1,i} \mu_0; \nu_1 \xrightarrow{\psi_2,i} \mu_0\right\}$$

$$= \frac{1}{m^2}\sum_{\psi_1,\psi_2}\mathbb{P}\left\{\mu_1 \xrightarrow{\psi_1,i} \mu_0\right\}\mathbb{P}\left\{\nu_1 \xrightarrow{\psi_1,i} \mu_0|\mu_1 \xrightarrow{\psi_1,i} \mu_0\right\}$$

$$= \frac{1}{m^2}\left[\frac{m^2}{v}\frac{v^{s-1}-1}{v^s-1}\right],$$
(33)

thus

$$\langle (p_i(\mu_0|\mu_1) - \langle p \rangle)(p_i(\mu_0|\nu_1) - \langle p \rangle) \rangle = \langle p_i(\mu_0|\mu_1)p_i(\mu_0|\nu_1) \rangle - \left(\frac{1}{v}\right)^2 = -\frac{1}{v^2}\frac{v-1}{v^s-1}, \quad (34)$$

and

$$\langle p_i(\mu_0|\mu_1)p_i(\nu_0|\nu_1) \rangle = \frac{1}{m^2}\sum_{\psi_1,\psi_2=1}^{m}\mathbb{P}\left\{\mu_1 \xrightarrow{\psi_1,i} \mu_0; \nu_1 \xrightarrow{\psi_2,i} \nu_0\right\}$$

$$= \frac{1}{m^2}\sum_{\psi_1,\psi_2}\mathbb{P}\left\{\mu_1 \xrightarrow{\psi_1,i} \mu_0\right\}\mathbb{P}\left\{\nu_1 \xrightarrow{\psi_1,i} \nu_0|\mu_1 \xrightarrow{\psi_1,i} \mu_0\right\}$$

$$= \frac{1}{m^2}\left[\frac{m^2}{v}\frac{v^{s-1}}{v^s-1}\right],$$
(35)

thus

$$\langle (p_i(\mu_0|\mu_1) - \langle p \rangle)(p_i(\nu_0|\nu_1) - \langle p \rangle) \rangle = \langle p_i(\mu_0|\mu_1)p_i(\nu_0|\mu_1) \rangle - \left(\frac{1}{v}\right)^2 = \frac{1}{v^2}\frac{1}{v^s-1}. \quad (36)$$

## D Analytic computation of spatial correlations

Given a dataset of $d$-dimensional sequences of tokens in $\mathcal{V}$, we measure correlations via the token co-occurrences matrix,

$$C_{i,j}(\mu,\nu) := \mathbb{P}\left\{X_i = \mu, X_j = \nu\right\}_X - \mathbb{P}\left\{X_i = \mu\right\}_X \mathbb{P}\left\{X_j = \nu\right\}_X, \quad (37)$$

where $\mu$ and $\nu$ are arbitrary elements of the vocabulary $\mathcal{V}$ and $\mathbb{P}_X$ refers to the probability of the data distribution (distinct from $\mathbb{P}_{RHM}$, indicating the probability of the rules of the generative process). Joint and single-token probabilities are given by Eq. 17 and Eq. 38, respectively. We now prove Eq. 8 of the main text.

### D.1 Level-1 LCA (i-th and j-th tokens are in the same patch)

When the LCA of the $i$-th and $j$-th tokens is a level-1 hidden symbol, i.e. the two tokens lie in the same $s$-patch,

$$
\mathbb{P}\{X_i = \mu, X_j = \nu\}_X = \sum_{\mu_1=1}^{v} p_{i_1,j_1}^{(1)}(\mu, \nu|\mu_1) p_{i_2}^{(2)}(\mu_1), \quad (i_1 \neq j_1),
$$

$$
\mathbb{P}\{X_i = \mu\}_X = \sum_{\mu_1=1}^{v} p_{i_1}^{(1)}(\mu|\mu_1) p_{i_2}^{(2)}(\mu_1),
$$

$$
\mathbb{P}\{X_j = \nu\}_X = \sum_{\nu_1=1}^{v} p_{j_1}^{(1)}(\nu|\nu_1) p_{j_2}^{(2)}(\nu_1), \quad (j_2 = i_2). \tag{38}
$$

We consider the limit of large $m$, where the univariate probabilities of the hidden variables of any level converge to $1/v$, with relative fluctuations of order $1/\sqrt{m}$ [20] [5]. In this limit, we can substitute the probability of the LCA with $1/v$, thus obtaining,

$$
C^{(1)}(\mu, \nu) = \frac{1}{v} \sum_{\mu_1} p_{i_1,j_1}^{(1)}(\mu, \nu|\mu_1) - \frac{1}{v^2} \sum_{\mu_1,\nu_1} p_{i_1}^{(1)}(\mu|\mu_1) p_{j_1}^{(1)}(\nu|\nu_1). \tag{39}
$$

As we will prove in this and the following sections, the correlations have $0$ average but nonvanishing variance. Including the fluctuations of the LCA probability results in corrections to the variance that vanish in the limit of large $m$. Furthermore, notice that we removed the dependence of $C(\mu, \nu)$ on the positional indices $i$ and $j$. This is because, asymptotically in $v$ and $m$, the aforementioned statistics depend only on the depth of the LCA of $i$-th and $j$-th tokens, justifying our notation.

Since $i_1 \neq j_1$, $p_{i_1}^{(1)}(\mu|\mu_1)$ and $p_{j_1}^{(1)}(\nu|\nu_1)$ are independent. Hence, by Eq. 21 and Eq. 23 with $s' = 2$,

$$
\left\langle C^{(1)}(\mu, \nu) \right\rangle = \frac{1}{v} \sum_{\mu_1} \frac{\langle N_2 \rangle}{m} - \frac{1}{v^2} \sum_{\mu_1,\nu_1} \frac{\langle N \rangle}{m} \frac{\langle N \rangle}{m} = 0. \tag{40}
$$

The variance/2nd moment reads

$$
\left\langle \left( C^{(1)}(\mu, \nu) \right)^2 \right\rangle = \left\langle \left( \frac{1}{v} \sum_{\mu_1} p_{i_1,j_1}^{(1)}(\mu, \nu|\mu_1) \right)^2 \right\rangle + \left\langle \left( \frac{1}{v} \sum_{\mu_1} p_{i_1}^{(1)}(\mu|\mu_1) \right)^2 \right\rangle \left\langle \left( \frac{1}{v} \sum_{\nu_1} p_{j_1}^{(1)}(\nu|\nu_1) \right)^2 \right\rangle
$$

$$
- 2 \frac{1}{v^3} \sum_{\mu_1,\lambda_1,\kappa_1} \left\langle p_{i_1,j_1}^{(1)}(\mu, \nu|\mu_1) p_{i_1}^{(1)}(\mu|\lambda_1) p_{j_1}^{(1)}(\nu|\kappa_1) \right\rangle. \tag{41}
$$

We will compute the three contributions on the right-hand side separately in the following subsections.

### D.1.1 One-point term (marginal probability)

$$
\mathcal{P}_c(\mu) = \left\langle \left( \frac{1}{v} \sum_{\mu_1} p_{i_1}^{(1)}(\mu|\mu_1) \right)^2 \right\rangle
$$

$$
= \frac{1}{v^2} \sum_{\mu_1,\nu_1} \left\langle p_{i_1}^{(1)}(\mu|\mu_1) p_{i_1}^{(1)}(\mu|\nu_1) \right\rangle \tag{42}
$$

We can split the sum into two kinds of terms: those with $\mu_1 = \nu_1$ (mult. $v$) and those with $\mu_1 \neq \nu_1$ (mult. $v(v-1)$). In the following, to simplify the notation, we omit the spatial index $i$.

---

[5]Section 1d of Appendix B.

**(i)**—$\mu_1 = \nu_1$ **(mult. $v$)**

$$
\begin{aligned}
\mathcal{P}_c(\mu)^{(i)} &= \frac{v}{(mv)^2} \sum_{\psi_1,\psi_2} \langle \delta(r_{\psi_1}(\mu_1),(\mu)) \delta(r_{\psi_2}(\mu_1),(\mu)) \rangle \\
&= \frac{v}{(mv)^2} \sum_{\psi_1,\psi_2} \mathbb{P}\left\{ \mu_1 \xrightarrow{\psi_1} \mu; \mu_1 \xrightarrow{\psi_2} \mu \right\} \\
&= \frac{v}{(mv)^2} \left( \sum_{\psi_1} \mathbb{P}\left\{ \mu_1 \xrightarrow{\psi_1} \mu \right\} + \sum_{\psi_1,\psi_2 \neq \psi_1} \mathbb{P}\left\{ \mu_1 \xrightarrow{\psi_1} \mu \right\} \mathbb{P}\left\{ \mu_1 \xrightarrow{\psi_2} \mu | \mu_1 \xrightarrow{\psi_1} \mu \right\} \right) \\
&= \frac{v}{(mv)^2} \left[ m\frac{1}{v} + m(m-1)\frac{1}{v}\frac{v^{s-1}-1}{v^s-1} \right].
\end{aligned}
\tag{43}
$$

**(ii)**—$\mu_1 \neq \nu_1$ **(mult. $v(v-1)$)**

$$
\begin{aligned}
\mathcal{P}_c(\mu,\nu)^{(ii)} &= \frac{v(v-1)}{(mv)^2} \sum_{\psi_1,\psi_2} \mathbb{P}\left\{ \mu_1 \xrightarrow{\psi_1} \mu; \nu_1 \xrightarrow{\psi_2} \mu \right\} \\
&= \frac{v(v-1)}{(mv)^2} \sum_{\psi_1,\psi_2} \mathbb{P}\left\{ \mu_1 \xrightarrow{\psi_1} \mu \right\} \mathbb{P}\left\{ \nu_1 \xrightarrow{\psi_2} \mu | \mu_1 \xrightarrow{\psi_1} \mu \right\} \\
&= \frac{v(v-1)}{(mv)^2} \left[ m^2\frac{1}{v}\frac{v^{s-1}-1}{v^s-1} \right].
\end{aligned}
\tag{44}
$$

**Variance of the marginal probability**

$$
\begin{aligned}
\left\langle \left( \frac{1}{v} \sum_{\mu_1} p_{i_1}^{(1)}(\mu|\mu_1) \right)^2 \right\rangle &- \left\langle \frac{1}{v} \sum_{\mu_1} p_{i_1}^{(1)}(\mu|\mu_1) \right\rangle^2 = \mathcal{P}_c(\mu,\nu) - \left( \frac{1}{v} \right)^2 \\
&= \frac{v-1}{v^3 m}\frac{v^s - mv}{v^s - 1}.
\end{aligned}
\tag{45}
$$

### D.1.2 Two-point term (joint probability)

$$
\begin{aligned}
\mathcal{J}_c(\mu,\nu) &:= \left\langle \left( \frac{1}{v} \sum_{\mu_1} p_{i_1,j_1}^{(1)}(\mu,\nu|\mu_1) \right)^2 \right\rangle \\
&= \frac{1}{v^2} \sum_{\mu_1,\nu_1} \left\langle p_{i,j}^{(1)}(\mu,\nu|\mu_1) p_{i,j}^{(1)}(\mu,\nu|\nu_1) \right\rangle.
\end{aligned}
\tag{46}
$$

We can split the sum into two kinds of terms: those with $\mu_1 = \nu_1$ (mult. $v$) and those with $\mu_1 \neq \nu_1$ (mult. $v(v-1)$). In the following, to simplify the notation, we omit the spatial indices $i$ and $j$.

**(i)**—$\mu_1 = \nu_1$ **(mult. $v$)**

$$
\begin{aligned}
\mathcal{J}_c(\mu,\nu)^{(i)} &= \frac{v}{(mv)^2} \sum_{\psi_1,\psi_2} \langle \delta(r_{\psi_1}(\mu_1),(\mu,\nu)) \delta(r_{\psi_2}(\mu_1),(\mu,\nu)) \rangle \\
&= \frac{v}{(mv)^2} \sum_{\psi_1,\psi_2} \mathbb{P}\left\{ \mu_1 \xrightarrow{\psi_1} \mu\nu; \mu_1 \xrightarrow{\psi_2} \mu\nu \right\} \\
&= \frac{v}{(mv)^2} \left( \sum_{\psi_1} \mathbb{P}\left\{ \mu_1 \xrightarrow{\psi_1} \mu\nu \right\} + \sum_{\psi_1,\psi_2 \neq \psi_1} \mathbb{P}\left\{ \mu_1 \xrightarrow{\psi_1} \mu\nu \right\} \mathbb{P}\left\{ \mu_1 \xrightarrow{\psi_2} \mu\nu | \mu_1 \xrightarrow{\psi_1} \mu\nu \right\} \right) \\
&= \frac{v}{(mv)^2} \left[ m\frac{1}{v^2} + m(m-1)\frac{1}{v^2}\frac{v^{s-2}-1}{v^s-1} \right].
\end{aligned}
\tag{47}
$$

**(ii)—$\mu_1 \neq \nu_1$ (mult. $v(v-1)$)**

$$\mathcal{J}_c(\mu,\nu)^{(ii)} = \frac{v(v-1)}{(mv)^2} \sum_{\psi_1,\psi_2} \mathbb{P}\left\{\mu_1 \xrightarrow{\psi_1} \mu\nu; \nu_1 \xrightarrow{\psi_2} \mu\nu\right\}$$

$$= \frac{v(v-1)}{(mv)^2} \sum_{\psi_1,\psi_2} \mathbb{P}\left\{\mu_1 \xrightarrow{\psi_1} \mu\nu\right\} \mathbb{P}\left\{\nu_1 \xrightarrow{\psi_2} \mu\nu | \mu_1 \xrightarrow{\psi_1} \mu\nu\right\}$$

$$= \frac{v(v-1)}{(mv)^2} \left[m^2 \frac{1}{v^2} \frac{v^{s-2}-1}{v^s-1}\right]. \tag{48}$$

**Variance of the joint probability**

$$\left\langle \left(\frac{1}{v}\sum_{\mu_1} p_{i_1,j_1}^{(1)}(\mu,\nu|\mu_1)\right)^2 \right\rangle - \left\langle \frac{1}{v}\sum_{\mu_1} p_{i_1,j_1}^{(1)}(\mu,\nu|\mu_1) \right\rangle^2 = \mathcal{J}_c(\mu,\nu) - \left(\frac{1}{v^2}\right)^2$$

$$= \frac{v^2-1}{v^5 m}\frac{v^s-mv}{v^s-1}. \tag{49}$$

### D.1.3 Three-point term

$$\mathcal{T}_c(\mu,\nu) := \frac{1}{v^3} \sum_{\mu_1,\lambda_1,\kappa_1} \left\langle p_{i,j}^{(1)}(\mu,\nu|\mu_1) p_i^{(1)}(\mu|\lambda_1) p_j^{(1)}(\nu|\kappa_1) \right\rangle$$

$$= \frac{1}{v^3} \sum_{\mu_1,\lambda_1,\kappa_1} \sum_{\mu',\nu'} \left\langle p_{i,j}^{(1)}(\mu,\nu|\mu_1) p_{i,j}^{(1)}(\mu,\nu'|\lambda_1) p_{i,j}^{(1)}(\mu',\nu|\kappa_1) \right\rangle$$

$$= \frac{1}{(vm)^3} \sum_{\mu_1,\psi_1;\lambda_1,\psi_2;\kappa_1,\psi_3} \sum_{\mu',\nu'} \mathbb{P}\left\{\mu_1 \xrightarrow{\psi_1,ij} \mu\nu; \lambda_1 \xrightarrow{\psi_2,ij} \mu\nu'; \kappa_1 \xrightarrow{\psi_3,ij} \mu'\nu\right\} \tag{50}$$

The sum over $\mu', \nu'$ can be split in 4 terms: one with $\mu' = \mu$ and $\nu' = \nu$ (mult. 1), one with $\mu' = \mu$ and $\nu' \neq \nu$ (mult. $(v-1)$), one with $\mu' \neq \mu$ and $\nu' = \nu$ (mult. $(v-1)$) and one with $\mu' \neq \mu$ and $\nu' \neq \nu$ (mult. $(v-1)^2$). Fixing the right-hand sides, the value of the joint probability of the rules depends only on the rule indices $\tilde{\psi}_1 = (\mu_1,\psi_1)$, $\tilde{\psi}_2 = (\lambda_1,\psi_2)$ and $\tilde{\psi}_3 = (\kappa_1,\psi_3)$.

The sum over $\mu_1, \lambda_1, \kappa_1$ can be split in 5 terms: one with $\mu_1 = \lambda_1 = \kappa_1$ (mult. $v$), one with $\mu_1 = \lambda_1 \neq \kappa_1$ (mult. $v(v-1)$), one with $\mu_1 = \kappa_1 \neq \lambda_1$ (mult. $v(v-1)$), one with $\mu_1 \neq \lambda_1 = \kappa_1$ (mult. $v(v-1)$), one with $\mu_1 \neq \lambda_1 \neq \kappa_1$ (mult. $v(v-1)(v-2)$). In the following, to simplify the notation, we omit the spatial indices $i$ and $j$.

**(i-a)—$\mu_1 = \lambda_1 = \kappa_1$; $\mu' = \mu$ and $\nu' = \nu$ (mult. $v$)**

$$\mathcal{T}_c(\mu,\nu)^{(i-a)} = \frac{v}{(mv)^3} \sum_{\psi_1,\psi_2,\psi_3} \left\langle \delta(r_{\psi_1}(\mu_1),(\mu,\nu))\delta(r_{\psi_2}(\mu_1),(\mu,\nu))\delta(r_{\psi_3}(\mu_1),(\mu,\nu)) \right\rangle$$

$$= \frac{v}{(mv)^3} \sum_{\psi_1,\psi_2,\psi_3} \mathbb{P}\left\{\mu_1 \xrightarrow{\psi_1} \mu\nu; \mu_1 \xrightarrow{\psi_2} \mu\nu; \mu_1 \xrightarrow{\psi_3} \mu\nu\right\}$$

$$= \frac{v}{(mv)^3} \sum_{\psi_1=\psi_2=\psi_3} \mathbb{P}\left\{\mu_1 \xrightarrow{\psi_1} \mu\nu\right\}$$

$$+ \frac{3v}{(mv)^3} \sum_{\psi_1=\psi_2\neq\psi_3} \mathbb{P}\left\{\mu_1 \xrightarrow{\psi_1} \mu\nu\right\} \mathbb{P}\left\{\mu_1 \xrightarrow{\psi_3} \mu\nu | \mu_1 \xrightarrow{\psi_1} \mu\nu\right\}$$

$$+ \frac{v}{(mv)^3} \sum_{\psi_1,\psi_2\neq\psi_1,\psi_3\neq\psi_2,\psi_1} \mathbb{P}\left\{\mu_1 \xrightarrow{\psi_1} \mu\nu\right\} \mathbb{P}\left\{\mu_1 \xrightarrow{\psi_2} \mu\nu | \mu_1 \xrightarrow{\psi_1} \mu\nu\right\} \mathbb{P}\left\{\mu_1 \xrightarrow{\psi_3} \mu\nu | \mu_1 \xrightarrow{\psi_2} \mu\nu; \mu_1 \xrightarrow{\psi_1} \mu\nu\right\}$$

$$= \frac{v}{(mv)^3} \left[m\frac{1}{v^2} + 3m(m-1)\frac{1}{v^2}\frac{v^{s-2}-1}{v^s-1} + m(m-1)(m-2)\frac{1}{v^2}\frac{v^{s-2}-1}{v^s-1}\frac{v^{s-2}-2}{v^s-2}\right]. \tag{51}$$

**(i-b)**—$\mu_1 = \lambda_1 = \kappa_1$; $\mu' = \mu$ **and** $\nu' \neq \nu$ **(mult.** $v(v-1)$**)**

$$
\begin{aligned}
\mathcal{T}_c(\mu,\nu)^{(i-b)} &= \frac{v(v-1)}{(mv)^3} \sum_{\psi_1,\psi_2,\psi_3} \mathbb{P}\left\{\mu_1 \xrightarrow{\psi_1} \mu\nu; \mu_1 \xrightarrow{\psi_2} \mu\nu'; \mu_1 \xrightarrow{\psi_3} \mu\nu\right\} \\
&= \frac{v(v-1)}{(mv)^3} \sum_{\psi_1=\psi_3,\psi_2\neq\psi_1} \mathbb{P}\left\{\mu_1 \xrightarrow{\psi_1} \mu\nu\right\} \mathbb{P}\left\{\mu_1 \xrightarrow{\psi_2} \mu\nu' | \mu_1 \xrightarrow{\psi_1} \mu\nu\right\} \\
&\quad \frac{v(v-1)}{(mv)^3} \sum_{\psi_1,\psi_2\neq\psi_1,\psi_3\neq\psi_2,\psi_1} \mathbb{P}\left\{\mu_1 \xrightarrow{\psi_1} \mu\nu\right\} \mathbb{P}\left\{\mu_1 \xrightarrow{\psi_2} \mu\nu' | \mu_1 \xrightarrow{\psi_1} \mu\nu\right\} \\
&\quad \times \mathbb{P}\left\{\mu_1 \xrightarrow{\psi_3} \mu\nu | \mu_1 \xrightarrow{\psi_2} \mu\nu'; \mu_1 \xrightarrow{\psi_1} \mu\nu\right\} \\
&= \frac{v(v-1)}{(mv)^3} \left[ m(m-1)\frac{1}{v^2}\frac{v^{s-2}}{v^s-1} + m(m-1)(m-2)\frac{1}{v^2}\frac{v^{s-2}}{v^s-1}\frac{v^{s-2}-1}{v^s-2} \right].
\end{aligned}
$$

(52)

**(i-c)**—$\mu_1 = \lambda_1 = \kappa_1$; $\mu' \neq \mu$ **and** $\nu' = \nu$ **(mult.** $v(v-1)$**)**

$$
\mathcal{T}_c(\mu,\nu)^{(i-c)} = \frac{v(v-1)}{(mv)^3} \sum_{\psi_1,\psi_2,\psi_3} \mathbb{P}\left\{\mu_1 \xrightarrow{\psi_1} \mu\nu; \mu_1 \xrightarrow{\psi_2} \mu\nu; \mu_1 \xrightarrow{\psi_3} \mu'\nu\right\} = \mathcal{T}_c(\mu,\nu)^{(i-b)},
$$

(53)

by symmetry for exchanging $\mu'$ and $\nu'$.

**(i-d)**—$\mu_1 = \lambda_1 = \kappa_1$; $\mu' \neq \mu$ **and** $\nu' \neq \nu$ **(mult.** $v(v-1)^2$**)**

$$
\begin{aligned}
\mathcal{T}_c(\mu,\nu)^{(i-d)} &= \frac{v(v-1)^2}{(mv)^3} \sum_{\psi_1,\psi_2,\psi_3} \mathbb{P}\left\{\mu_1 \xrightarrow{\psi_1} \mu\nu; \mu_1 \xrightarrow{\psi_2} \mu\nu'; \mu_1 \xrightarrow{\psi_3} \mu'\nu\right\} \\
&= \frac{v(v-1)^2}{(mv)^3} \sum_{\psi_1,\psi_2\neq\psi_1,\psi_3\neq\psi_2,\psi_1} \mathbb{P}\left\{\mu_1 \xrightarrow{\psi_1} \mu\nu\right\} \mathbb{P}\left\{\mu_1 \xrightarrow{\psi_2} \mu\nu' | \mu_1 \xrightarrow{\psi_1} \mu\nu\right\} \\
&\quad \times \mathbb{P}\left\{\mu_1 \xrightarrow{\psi_3} \mu'\nu | \mu_1 \xrightarrow{\psi_2} \mu\nu'; \mu_1 \xrightarrow{\psi_1} \mu\nu\right\} \\
&= \frac{v(v-1)^2}{(mv)^3} \left[ m(m-1)(m-2)\frac{1}{v^2}\frac{v^{s-2}}{v^s-1}\frac{v^{s-2}}{v^s-2} \right].
\end{aligned}
$$

(54)

**(ii-a)**—$\mu_1 \neq \lambda_1 = \kappa_1$; $\mu' = \mu$ **and** $\nu' = \nu$ **(mult.** $v(v-1)$**)**

$$
\mathcal{T}_c(\mu,\nu)^{(ii-a)} = \frac{v(v-1)}{(mv)^3} \sum_{\psi_1,\psi_2,\psi_3} \mathbb{P}\left\{\mu_1 \xrightarrow{\psi_1} \mu\nu; \lambda_1 \xrightarrow{\psi_2} \mu\nu; \lambda_1 \xrightarrow{\psi_3} \mu\nu\right\}
$$

$$
= \frac{v(v-1)}{(mv)^3} \sum_{\psi_1,\psi_2=\psi_3} \mathbb{P}\left\{\mu_1 \xrightarrow{\psi_1} \mu\nu\right\} \mathbb{P}\left\{\lambda_1 \xrightarrow{\psi_2} \mu\nu | \mu_1 \xrightarrow{\psi_1} \mu\nu\right\}
$$

$$
+ \frac{v(v-1)}{(mv)^3} \sum_{\psi_1\psi_2,\psi_3\neq\psi_2} \mathbb{P}\left\{\mu_1 \xrightarrow{\psi_1} \mu\nu\right\} \mathbb{P}\left\{\lambda_1 \xrightarrow{\psi_2} \mu\nu | \mu_1 \xrightarrow{\psi_1} \mu\nu\right\} \mathbb{P}\left\{\lambda_1 \xrightarrow{\psi_3} \mu\nu | \lambda_1 \xrightarrow{\psi_2} \mu\nu; \mu_1 \xrightarrow{\psi_1} \mu\nu\right\}
$$

$$
\frac{v(v-1)}{(mv)^3} \left[ m^2\frac{1}{v^2}\frac{v^{s-2}-1}{v^s-1} + m^2(m-1)\frac{1}{v^2}\frac{v^{s-2}-1}{v^s-1}\frac{v^{s-2}-2}{v^s-2} \right].
$$

(55)

**(ii-b)**—$\mu_1 \neq \lambda_1 = \kappa_1$; $\mu' = \mu$ **and** $\nu' \neq \nu$ **(mult.** $v(v-1)^2$**)**

$$
\mathcal{T}_c(\mu,\nu)^{(ii-b)} = \frac{v(v-1)^2}{(mv)^3} \sum_{\psi_1,\psi_2,\psi_3} \mathbb{P}\left\{\mu_1 \xrightarrow{\psi_1} \mu\nu; \lambda_1 \xrightarrow{\psi_2} \mu\nu'; \lambda_1 \xrightarrow{\psi_3} \mu\nu\right\}
$$

$$
= \frac{v(v-1)^2}{(mv)^3} \sum_{\psi_1\psi_2,\psi_3\neq\psi_2} \mathbb{P}\left\{\mu_1 \xrightarrow{\psi_1} \mu\nu\right\} \mathbb{P}\left\{\lambda_1 \xrightarrow{\psi_2} \mu\nu' | \mu_1 \xrightarrow{\psi_1} \mu\nu\right\} \mathbb{P}\left\{\lambda_1 \xrightarrow{\psi_3} \mu\nu | \lambda_1 \xrightarrow{\psi_2} \mu\nu'; \mu_1 \xrightarrow{\psi_1} \mu\nu\right\}
$$

$$
\frac{v(v-1)^2}{(mv)^3} \left[ m^2(m-1)\frac{1}{v^2}\frac{v^{s-2}}{v^s-1}\frac{v^{s-2}-1}{v^s-2} \right].
$$

(56)

**(ii-c)**—$\mu_1 \neq \lambda_1 = \kappa_1$; $\mu' \neq \mu$ **and** $\nu' = \nu$ **(mult. $v(v-1)^2$)**

$$\mathcal{T}_c(\mu,\nu)^{(ii-c)} = \frac{v(v-1)^2}{(mv)^3} \sum_{\psi_1,\psi_2,\psi_3} \mathbb{P}\left\{ \mu_1 \xrightarrow{\psi_1} \mu\nu; \lambda_1 \xrightarrow{\psi_2} \mu\nu; \lambda_1 \xrightarrow{\psi_3} \mu'\nu \right\} = \mathcal{T}_c(\mu,\nu)^{(ii-b)},$$

(57)

by symmetry for exchanging $\mu'$ and $\nu'$.

**(ii-d)**—$\mu_1 \neq \lambda_1 = \kappa_1$; $\mu' \neq \mu$ **and** $\nu' \neq \nu$ **(mult. $v(v-1)^3$)**

$$\mathcal{T}_c(\mu,\nu)^{(ii-d)} = \frac{v(v-1)^3}{(mv)^3} \sum_{\psi_1,\psi_2,\psi_3} \mathbb{P}\left\{ \mu_1 \xrightarrow{\psi_1} \mu\nu; \lambda_1 \xrightarrow{\psi_2} \mu\nu'; \lambda_1 \xrightarrow{\psi_3} \mu'\nu \right\}$$

$$= \frac{v(v-1)^3}{(mv)^3} \sum_{\psi_1\psi_2,\psi_3 \neq \psi_2} \mathbb{P}\left\{ \mu_1 \xrightarrow{\psi_1} \mu\nu \right\} \mathbb{P}\left\{ \lambda_1 \xrightarrow{\psi_2} \mu\nu' | \mu_1 \xrightarrow{\psi_1} \mu\nu \right\} \mathbb{P}\left\{ \lambda_1 \xrightarrow{\psi_3} \mu'\nu | \lambda_1 \xrightarrow{\psi_2} \mu\nu'; \mu_1 \xrightarrow{\psi_1} \mu\nu \right\}$$

$$\frac{v(v-1)^3}{(mv)^3} \left[ m^2(m-1)\frac{1}{v^2}\frac{v^{s-2}}{v^s-1}\frac{v^{s-2}}{v^s-2} \right].$$

(58)

**(iii-a)**—$\mu_1 = \lambda_1 \neq \kappa_1$; $\mu' = \mu$ **and** $\nu' = \nu$ **(mult. $v(v-1)$)**

$$\mathcal{T}_c(\mu,\nu)^{(iii-a)} = \frac{v(v-1)}{(mv)^3} \sum_{\psi_1,\psi_2,\psi_3} \mathbb{P}\left\{ \mu_1 \xrightarrow{\psi_1} \mu\nu; \mu_1 \xrightarrow{\psi_2} \mu\nu; \kappa_1 \xrightarrow{\psi_3} \mu\nu \right\}$$

$$= \frac{v(v-1)}{(mv)^3} \sum_{\psi_1=\psi_2,\psi_3} \mathbb{P}\left\{ \mu_1 \xrightarrow{\psi_1} \mu\nu \right\} \mathbb{P}\left\{ \kappa_1 \xrightarrow{\psi_3} \mu\nu | \mu_1 \xrightarrow{\psi_1} \mu\nu \right\}$$

$$+ \frac{v(v-1)}{(mv)^3} \sum_{\psi_1,\psi_2\neq\psi_1,\psi_3} \mathbb{P}\left\{ \mu_1 \xrightarrow{\psi_1} \mu\nu \right\} \mathbb{P}\left\{ \mu_1 \xrightarrow{\psi_2} \mu\nu | \mu_1 \xrightarrow{\psi_1} \mu\nu \right\} \mathbb{P}\left\{ \kappa_1 \xrightarrow{\psi_3} \mu\nu | \mu_1 \xrightarrow{\psi_2} \mu\nu; \mu_1 \xrightarrow{\psi_1} \mu\nu \right\}$$

$$\frac{v(v-1)}{(mv)^3} \left[ m^2\frac{1}{v^2}\frac{v^{s-2}-1}{v^s-1} + m^2(m-1)\frac{1}{v^2}\frac{v^{s-2}-1}{v^s-1}\frac{v^{s-2}-2}{v^s-2} \right] = \mathcal{T}_c(\mu,\nu)^{(ii-a)}.$$

(59)

**(iii-b)**—$\mu_1 = \lambda_1 \neq \kappa_1$; $\mu' = \mu$ **and** $\nu' \neq \nu$ **(mult. $v(v-1)^2$)**

$$\mathcal{T}_c(\mu,\nu)^{(iii-b)} = \frac{v(v-1)^2}{(mv)^3} \sum_{\psi_1,\psi_2,\psi_3} \mathbb{P}\left\{ \mu_1 \xrightarrow{\psi_1} \mu\nu; \mu_1 \xrightarrow{\psi_2} \mu\nu'; \kappa_1 \xrightarrow{\psi_3} \mu\nu \right\}$$

$$= \frac{v(v-1)^2}{(mv)^3} \sum_{\psi_1\psi_2\neq\psi_1,\psi_3} \mathbb{P}\left\{ \mu_1 \xrightarrow{\psi_1} \mu\nu \right\} \mathbb{P}\left\{ \mu_1 \xrightarrow{\psi_2} \mu\nu' | \mu_1 \xrightarrow{\psi_1} \mu\nu \right\} \mathbb{P}\left\{ \kappa_1 \xrightarrow{\psi_3} \mu\nu | \mu_1 \xrightarrow{\psi_2} \mu\nu'; \mu_1 \xrightarrow{\psi_1} \mu\nu \right\}$$

$$\frac{v(v-1)^2}{(mv)^3} \left[ m^2(m-1)\frac{1}{v^2}\frac{v^{s-2}}{v^s-1}\frac{v^{s-2}-1}{v^s-2} \right].$$

(60)

**(iii-c)**—$\mu_1 = \lambda_1 \neq \kappa_1$; $\mu' \neq \mu$ **and** $\nu' = \nu$ **(mult. $v(v-1)^2$)**

$$\mathcal{T}_c(\mu,\nu)^{(iii-c)} = \frac{v(v-1)^2}{(mv)^3} \sum_{\psi_1,\psi_2,\psi_3} \mathbb{P}\left\{ \mu_1 \xrightarrow{\psi_1} \mu\nu; \mu_1 \xrightarrow{\psi_2} \mu\nu; \kappa_1 \xrightarrow{\psi_3} \mu'\nu \right\}$$

$$= \frac{v(v-1)^2}{(mv)^3} \sum_{\psi_1,\psi_2=\psi_1,\psi_3} \mathbb{P}\left\{ \mu_1 \xrightarrow{\psi_1} \mu\nu \right\} \mathbb{P}\left\{ \kappa_1 \xrightarrow{\psi_3} \mu'\nu | \mu_1 \xrightarrow{\psi_1} \mu\nu \right\}$$

$$+ \frac{v(v-1)^2}{(mv)^3} \sum_{\psi_1,\psi_2\neq\psi_1,\psi_3} \mathbb{P}\left\{ \mu_1 \xrightarrow{\psi_1} \mu\nu \right\} \mathbb{P}\left\{ \mu_1 \xrightarrow{\psi_2} \mu\nu | \mu_1 \xrightarrow{\psi_1} \mu\nu \right\} \mathbb{P}\left\{ \kappa_1 \xrightarrow{\psi_3} \mu'\nu | \mu_1 \xrightarrow{\psi_2} \mu\nu; \mu_1 \xrightarrow{\psi_1} \mu\nu \right\}$$

$$= \frac{v(v-1)^2}{(mv)^3} \left[ m^2\frac{1}{v^2}\frac{v^{s-2}}{v^s-1} + m^2(m-1)\frac{1}{v^2}\frac{v^{s-2}-1}{v^s-1}\frac{v^{s-2}}{v^s-2} \right].$$

(61)

**(iii-d)**—$\mu_1 = \lambda_1 \neq \kappa_1$; $\mu' \neq \mu$ **and** $\nu' \neq \nu$ **(mult.** $v(v-1)^3$**)**

$$\mathcal{T}_c(\mu,\nu)^{(iii-d)} = \frac{v(v-1)^3}{(mv)^3} \sum_{\psi_1,\psi_2,\psi_3} \mathbb{P}\left\{\mu_1 \xrightarrow{\psi_1} \mu\nu; \mu_1 \xrightarrow{\psi_2} \mu\nu'; \kappa_1 \xrightarrow{\psi_3} \mu'\nu\right\}$$

$$= \frac{v(v-1)^3}{(mv)^3} \sum_{\psi_1\psi_2 \neq \psi_1,\psi_3} \mathbb{P}\left\{\mu_1 \xrightarrow{\psi_1} \mu\nu\right\} \mathbb{P}\left\{\mu_1 \xrightarrow{\psi_2} \mu\nu'|\mu_1 \xrightarrow{\psi_1} \mu\nu\right\} \mathbb{P}\left\{\kappa_1 \xrightarrow{\psi_3} \mu'\nu|\mu_1 \xrightarrow{\psi_2} \mu\nu'; \mu_1 \xrightarrow{\psi_1} \mu\nu\right\}$$

$$\frac{v(v-1)^3}{(mv)^3}\left[m^2(m-1)\frac{1}{v^2}\frac{v^{s-2}}{v^s-1}\frac{v^{s-2}}{v^s-2}\right]. \tag{62}$$

**(iv-a)**—$\mu_1 = \kappa_1 \neq \lambda_1$; $\mu' = \mu$ **and** $\nu' = \nu$ **(mult.** $v(v-1)$**)**

$$\mathcal{T}_c(\mu,\nu)^{(iv-a)} = \frac{v(v-1)}{(mv)^3} \sum_{\psi_1,\psi_2,\psi_3} \mathbb{P}\left\{\mu_1 \xrightarrow{\psi_1} \mu\nu; \lambda_1 \xrightarrow{\psi_2} \mu\nu; \mu_1 \xrightarrow{\psi_3} \mu\nu\right\} = \mathcal{T}_c(\mu,\nu)^{(iii-a)} = \mathcal{T}_c(\mu,\nu)^{(ii-a)}, \tag{63}$$

by symmetry for exchanging $\kappa_1$ and $\lambda_1$.

**(iv-b)**—$\mu_1 = \kappa_1 \neq \lambda_1$; $\mu' = \mu$ **and** $\nu' \neq \nu$ **(mult.** $v(v-1)^2$**)**

$$\mathcal{T}_c(\mu,\nu)^{(iv-b)} = \frac{v(v-1)^2}{(mv)^3} \sum_{\psi_1,\psi_2,\psi_3} \mathbb{P}\left\{\mu_1 \xrightarrow{\psi_1} \mu\nu; \lambda_1 \xrightarrow{\psi_2} \mu\nu'; \mu_1 \xrightarrow{\psi_3} \mu\nu\right\}$$

$$= \frac{v(v-1)^2}{(mv)^3} \sum_{\psi_1,\psi_2,\psi_3=\psi_1} \mathbb{P}\left\{\mu_1 \xrightarrow{\psi_1} \mu\nu\right\} \mathbb{P}\left\{\lambda_1 \xrightarrow{\psi_2} \mu\nu'|\mu_1 \xrightarrow{\psi_1} \mu\nu\right\}$$

$$+ \frac{v(v-1)^2}{(mv)^3} \sum_{\psi_1,\psi_2,\psi_3 \neq \psi_1} \mathbb{P}\left\{\mu_1 \xrightarrow{\psi_1} \mu\nu\right\} \mathbb{P}\left\{\lambda_1 \xrightarrow{\psi_2} \mu\nu'|\mu_1 \xrightarrow{\psi_1} \mu\nu\right\} \mathbb{P}\left\{\mu_1 \xrightarrow{\psi_3} \mu\nu|\lambda_1 \xrightarrow{\psi_2} \mu\nu'; \mu_1 \xrightarrow{\psi_1} \mu\nu\right\}$$

$$= \frac{v(v-1)^2}{(mv)^3}\left[m^2\frac{1}{v^2}\frac{v^{s-2}}{v^s-1} + m^2(m-1)\frac{1}{v^2}\frac{v^{s-2}}{v^s-1}\frac{v^{s-2}-1}{v^s-2}\right] = \mathcal{I}_c(\mu,\nu)^{(iii-c)}. \tag{64}$$

**(iv-c)**—$\mu_1 = \kappa_1 \neq \lambda_1$; $\mu' \neq \mu$ **and** $\nu' = \nu$ **(mult.** $v(v-1)^2$**)**

$$\mathcal{T}_c(\mu,\nu)^{(iv-c)} = \frac{v(v-1)^2}{(mv)^3} \sum_{\psi_1,\psi_2,\psi_3} \mathbb{P}\left\{\mu_1 \xrightarrow{\psi_1} \mu\nu; \lambda_1 \xrightarrow{\psi_2} \mu\nu; \mu_1 \xrightarrow{\psi_3} \mu'\nu\right\}$$

$$= \frac{v(v-1)^2}{(mv)^3} \sum_{\psi_1\psi_2,\psi_3 \neq \psi_1} \mathbb{P}\left\{\mu_1 \xrightarrow{\psi_1} \mu\nu\right\} \mathbb{P}\left\{\lambda_1 \xrightarrow{\psi_2} \mu\nu|\mu_1 \xrightarrow{\psi_1} \mu\nu\right\} \mathbb{P}\left\{\mu_1 \xrightarrow{\psi_3} \mu'\nu|\lambda_1 \xrightarrow{\psi_2} \mu\nu; \mu_1 \xrightarrow{\psi_1} \mu\nu\right\}$$

$$\frac{v(v-1)^2}{(mv)^3}\left[m(m-1)^2\frac{1}{v^2}\frac{v^{s-2}-1}{v^s-1}\frac{v^{s-2}}{v^s-2}\right] = \mathcal{T}_c(\mu,\nu)^{(iii-b)}. \tag{65}$$

**(iv-d)**—$\mu_1 = \kappa_1 \neq \lambda_1$; $\mu' \neq \mu$ **and** $\nu' \neq \nu$ **(mult.** $v(v-1)^3$**)**

$$\mathcal{T}_c(\mu,\nu)^{(iv-d)} = \frac{v(v-1)^3}{(mv)^3} \sum_{\psi_1,\psi_2,\psi_3} \mathbb{P}\left\{\mu_1 \xrightarrow{\psi_1} \mu\nu; \lambda_1 \xrightarrow{\psi_2} \mu\nu'; \mu_1 \xrightarrow{\psi_3} \mu'\nu\right\}$$

$$= \frac{v(v-1)^3}{(mv)^3} \sum_{\psi_1\psi_2,\psi_3 \neq \psi_1} \mathbb{P}\left\{\mu_1 \xrightarrow{\psi_1} \mu\nu\right\} \mathbb{P}\left\{\lambda_1 \xrightarrow{\psi_2} \mu\nu'|\mu_1 \xrightarrow{\psi_1} \mu\nu\right\} \mathbb{P}\left\{\mu_1 \xrightarrow{\psi_3} \mu'\nu|\lambda_1 \xrightarrow{\psi_2} \mu\nu'; \mu_1 \xrightarrow{\psi_1} \mu\nu\right\}$$

$$\frac{v(v-1)^3}{(mv)^3}\left[m^2(m-1)\frac{1}{v^2}\frac{v^{s-2}}{v^s-1}\frac{v^{s-2}}{v^s-2}\right] = \mathcal{T}_c^{(iii-d)}. \tag{66}$$

**(v-a)**—$\mu_1 \neq \lambda_1 \neq \kappa_1$; $\mu' = \mu$ **and** $\nu' = \nu$ **(mult. $v(v-1)(v-2)$)**

$$\mathcal{T}_c(\mu,\nu)^{(v-a)} = \frac{v(v-1)(v-2)}{(mv)^3} \sum_{\psi_1,\psi_2,\psi_3} \mathbb{P}\left\{\mu_1 \xrightarrow{\psi_1} \mu\nu; \lambda_1 \xrightarrow{\psi_2} \mu\nu; \kappa_1 \xrightarrow{\psi_3} \mu\nu\right\}$$

$$= \frac{v(v-1)(v-2)}{(mv)^3} \sum_{\psi_1,\psi_2,\psi_3} \mathbb{P}\left\{\mu_1 \xrightarrow{\psi_1} \mu\nu\right\} \mathbb{P}\left\{\lambda_1 \xrightarrow{\psi_2} \mu\nu | \mu_1 \xrightarrow{\psi_1} \mu\nu\right\} \mathbb{P}\left\{\kappa_1 \xrightarrow{\psi_3} \mu\nu | \lambda_1 \xrightarrow{\psi_2} \mu\nu; \mu_1 \xrightarrow{\psi_1} \mu\nu\right\}$$

$$= \frac{v(v-1)(v-2)}{(mv)^3} \left[m^3 \frac{1}{v^2} \frac{v^{s-2}-1}{v^s-1} \frac{v^{s-2}-2}{v^s-2}\right]. \tag{67}$$

**(v-b)**—$\mu_1 \neq \lambda_1 \neq \kappa_1$; $\mu' = \mu$ **and** $\nu' \neq \nu$ **(mult. $v(v-1)^2(v-2)$)**

$$\mathcal{T}_c(\mu,\nu)^{(v-b)} = \frac{v(v-1)^2(v-2)}{(mv)^3} \sum_{\psi_1,\psi_2,\psi_3} \mathbb{P}\left\{\mu_1 \xrightarrow{\psi_1} \mu\nu; \lambda_1 \xrightarrow{\psi_2} \mu\nu'; \kappa_1 \xrightarrow{\psi_3} \mu\nu\right\}$$

$$= \frac{v(v-1)^2(v-2)}{(mv)^3} \left[m^3 \frac{1}{v^2} \frac{v^{s-2}}{v^s-1} \frac{v^{s-2}-1}{v^s-2}\right]. \tag{68}$$

**(v-c)**—$\mu_1 \neq \lambda_1 \neq \kappa_1$; $\mu' \neq \mu$ **and** $\nu' = \nu$ **(mult. $v(v-1)^2(v-2)$)**

$$\mathcal{T}_c(\mu,\nu)^{(v-c)} = \frac{v(v-1)^2(v-2)}{(mv)^3} \sum_{\psi_1,\psi_2,\psi_3} \mathbb{P}\left\{\mu_1 \xrightarrow{\psi_1} \mu\nu; \lambda_1 \xrightarrow{\psi_2} \mu\nu; \kappa_1 \xrightarrow{\psi_3} \mu'\nu\right\}$$

$$= \frac{v(v-1)^2(v-2)}{(mv)^3} \left[m^3 \frac{1}{v^2} \frac{v^{s-2}-1}{v^s-1} \frac{v^{s-2}}{v^s-2}\right]. \tag{69}$$

**(v-d)**—$\mu_1 \neq \lambda_1 \neq \kappa_1$; $\mu' \neq \mu$ **and** $\nu' \neq \nu$ **(mult. $v(v-1)^3(v-2)$)**

$$\mathcal{T}_c(\mu,\nu)^{(v-d)} = \frac{v(v-1)^3(v-2)}{(mv)^3} \sum_{\psi_1,\psi_2,\psi_3} \mathbb{P}\left\{\mu_1 \xrightarrow{\psi_1} \mu\nu; \lambda_1 \xrightarrow{\psi_2} \mu\nu'; \kappa_1 \xrightarrow{\psi_3} \mu'\nu\right\}$$

$$= \frac{v(v-1)^3(v-2)}{(mv)^3} \left[m^3 \frac{1}{v^2} \frac{v^{s-2}}{v^s-1} \frac{v^{s-2}}{v^s-2}\right]. \tag{70}$$

### D.1.4 Variance of the correlations

By adding together all the terms,

$$\left\langle \left(C^{(1)}(\mu,\nu)\right)^2\right\rangle = \left\langle \left(\frac{1}{v}\sum_{\mu_1} p^{(1)}_{i_1,j_1}(\mu,\nu|\mu_1)\right)^2\right\rangle + \left\langle \left(\frac{1}{v}\sum_{\mu_1} p^{(1)}_{i_1}(\mu|\mu_1)\right)^2\right\rangle \left\langle \left(\frac{1}{v}\sum_{\nu_1} p^{(1)}_{j_1}(\nu|\nu_1)\right)^2\right\rangle$$

$$- 2\frac{1}{v^3} \sum_{\mu_1,\lambda_1,\kappa_1} \left\langle p^{(1)}_{i_1,j_1}(\mu,\nu|\mu_1) p^{(1)}_{i_1}(\mu|\lambda_1) p^{(1)}_{j_1}(\nu|\kappa_1)\right\rangle$$

$$= \frac{v^{3s}}{(v^s-1)^2(v^s-2)} \frac{v^s-mv}{v^s} \frac{(mv-1)(v-1)^2}{v^6 m^2} \xrightarrow{v\gg 1} \frac{\left(1-m/v^{s-1}\right)}{v^3 m}. \tag{71}$$

### D.1.5 Covariance of the correlations

For all $\lambda \neq \mu$,

$$\sum_{\mu=1}^{v} C(\mu,\nu) = \sum_{\nu=1}^{v} C(\mu,\nu) = 0. \tag{72}$$

Therefore,

$$C(\mu,\nu)\sum_{\lambda=1}^{v} C(\lambda,\nu) = C(\mu,\nu)^2 + \sum_{\lambda\neq\mu} C(\mu,\nu)C(\lambda,\nu) = 0 \Rightarrow$$

$$\sum_{\lambda\neq\mu} \langle C(\mu,\nu)C(\lambda,\nu)\rangle = -\left\langle C(\mu,\nu)^2\right\rangle. \tag{73}$$

Analogously,

$$\sum_{\kappa \neq \nu} \langle C(\mu, \nu) C(\mu, \kappa) \rangle = -\langle C(\mu, \nu)^2 \rangle. \tag{74}$$

In addition,

$$C(\mu, \nu) \sum_{\lambda=1}^{v} C(\lambda, \kappa) = C(\mu, \nu) C(\mu, \kappa) + \sum_{\lambda \neq \mu} C(\mu, \nu) C(\lambda, \kappa) = 0 \Rightarrow$$

$$\sum_{\lambda \neq \mu, \kappa \neq \nu} \langle C(\mu, \nu) C(\lambda, \kappa) \rangle = -\sum_{\kappa \neq \nu} \langle C(\mu, \nu) C(\mu, \kappa) \rangle = \langle C(\mu, \nu)^2 \rangle. \tag{75}$$

### D.2 Level-2 LCA

When the parents of the $i$-th and $j$-th tokens are in the same level-1 patch ,

$$\mathbb{P}\{X_i = \mu, X_j = \nu\}_X = \sum_{\substack{\mu_1=1 \\ \nu_1=1}}^{v} \sum_{\mu_2=1}^{v} p_{i_1}^{(1)}(\mu|\mu_1) p_{j_1}^{(1)}(\nu|\nu_1) p_{i_2,j_2}^{(2)}(\mu_1, \nu_1|\mu_2) p_{i_3}^{(3)}(\mu_2), \quad (i_2 \neq j_2),$$

$$\mathbb{P}\{X_i = \mu\}_X = \sum_{\mu_1,\mu_2=1}^{v} p_{i_1}^{(1)}(\mu|\mu_1) p_{i_2}^{(2)}(\mu_1|\mu_2) p_{i_3}^{(3)}(\mu_2),$$

$$\mathbb{P}\{X_j = \nu\}_X = \sum_{\nu_1,\nu_2=1}^{v} p_{j_1}^{(1)}(\nu|\nu_1) p_{j_2}^{(2)}(\nu_1|\nu_2) p_{j_3}^{(3)}(\nu_2), \quad (j_3 = i_3). \tag{76}$$

Therefore, in the limit of large $m$, where the univariate probabilities of the hidden variables of any level converge to $1/v$,

$$C^{(2)}(\mu, \nu) = \mathbb{P}\{X_i = \mu, X_j = \nu\}_X - \mathbb{P}\{X_i = \mu\}_X \mathbb{P}\{X_j = \nu\}_X$$

$$= \sum_{\mu_1,\nu_1} p_{i_1}^{(1)}(\mu|\mu_1) p_{j_1}^{(1)}(\nu|\nu_1) \times$$

$$\left[ \sum_{\mu_2} p_{i_2,j_2}^{(2)}(\mu_1, \nu_1|\mu_2) p_{i_3}^{(3)}(\mu_2) - \frac{1}{v^2} \sum_{\mu_2,\nu_2} p_{i_2}^{(2)}(\mu_1|\mu_2) p_{i_3}^{(3)}(\mu_2) p_{j_2}^{(2)}(\nu_1|\nu_2) p_{i_3}^{(3)}(\nu_2) \right]$$

$$= \sum_{\mu_1,\nu_1} p_{i_1}^{(1)}(\mu|\mu_1) p_{j_1}^{(1)}(\nu|\nu_1) C^{(1)}(\mu_1, \nu_1). \tag{77}$$

Since the rules of different levels are independent, and $\langle C^{(1)}(\mu_1, \nu_1) \rangle = 0$,

$$\langle C^{(2)}(\mu, \nu) \rangle = \sum_{\mu_1,\nu_1} \langle p_{i_1}^{(1)}(\mu|\mu_1) p_{j_1}^{(1)}(\nu|\nu_1) \rangle \langle C^{(1)}(\mu_1, \nu_1) \rangle = 0. \tag{78}$$

The variance/2nd moment reads

$$\left\langle \left( C^{(2)}(\mu, \nu) \right)^2 \right\rangle = \sum_{\substack{\mu_1,\nu_1 \\ \lambda_1,\kappa_1}} \left\langle p_{i_1}^{(1)}(\mu|\mu_1) p_{i_1}^{(1)}(\mu|\lambda_1) p_{j_1}^{(1)}(\nu|\nu_1) p_{j_1}^{(1)}(\nu|\kappa_1) \right\rangle \left\langle C^{(1)}(\mu_1, \nu_1) C^{(1)}(\lambda_1, \kappa_1) \right\rangle$$

$$= \sum_{\mu_1,\nu_1} \left( \left\langle p_{i_1}^{(1)}(\mu|\mu_1)^2 p_{j_1}^{(1)}(\nu|\nu_1)^2 \right\rangle \left\langle C^{(1)}(\mu_1, \nu_1)^2 \right\rangle \right.$$

$$+ \sum_{\kappa_1 \neq \nu_1} \left\langle p_{i_1}^{(1)}(\mu|\mu_1)^2 p_{j_1}^{(1)}(\nu|\nu_1) p_{j_1}^{(1)}(\nu|\kappa_1) \right\rangle \left\langle C^{(1)}(\mu_1, \nu_1) C^{(1)}(\mu_1, \kappa_1) \right\rangle$$

$$+ \sum_{\lambda_1 \neq \mu_1} \left\langle p_{i_1}^{(1)}(\mu|\mu_1) p_{i_1}^{(1)}(\mu|\lambda_1) p_{j_1}^{(1)}(\nu|\nu_1)^2 \right\rangle \left\langle C^{(1)}(\mu_1, \nu_1) C^{(1)}(\lambda_1, \nu_1) \right\rangle$$

$$+ \sum_{\lambda_1 \neq \mu_1, \kappa_1 \neq \nu_1} \left\langle p_{i_1}^{(1)}(\mu|\mu_1) p_{i_1}^{(1)}(\mu|\lambda_1) p_{j_1}^{(1)}(\nu|\nu_1) p_{j_1}^{(1)}(\nu|\kappa_1) \right\rangle \left\langle C^{(1)}(\mu_1, \nu_1) C^{(1)}(\lambda_1, \kappa_1) \right\rangle \right). \tag{79}$$

### D.2.1 $i_1 \neq j_1$ case.

This is the easiest case because the production rule probabilities $p_i^{(1)}$ relative to different positions $i$ are independent and identically distributed (with respect to realisations of the RHM). Therefore, using Eq. 34,

$$\left\langle p_{i_1}^{(1)}(\mu|\mu_1)^2 p_{j_1}^{(1)}(\nu|\nu_1)^2 \right\rangle = \left\langle p_{i_1}^{(1)}(\mu|\mu_1)^2 \right\rangle \left\langle p_{j_1}^{(1)}(\nu|\nu_1)^2 \right\rangle = \left( \frac{1}{v^2} + \sigma_p^2 \right)^2,$$

$$\left\langle p_{i_1}^{(1)}(\mu|\mu_1)^2 p_{j_1}^{(1)}(\nu|\nu_1) p_{j_1}^{(1)}(\nu|\kappa_1) \right\rangle = \left\langle p_{i_1}^{(1)}(\mu|\mu_1)^2 \right\rangle \left\langle p_{j_1}^{(1)}(\nu|\nu_1) p_{j_1}^{(1)}(\nu|\kappa_1) \right\rangle =$$
$$= \left( \frac{1}{v^2} + \sigma_p^2 \right) \left( \frac{1}{v^2} - \frac{1}{v^2} \frac{v-1}{v^s - 1} \right),$$

$$\left\langle p_{i_1}^{(1)}(\mu|\mu_1) p_{i_1}^{(1)}(\mu|\lambda_1) p_{j_1}^{(1)}(\nu|\nu_1)^2 \right\rangle = \left\langle p_{i_1}^{(1)}(\mu|\mu_1) p_{i_1}^{(1)}(\mu|\lambda_1) \right\rangle \left\langle p_{j_1}^{(1)}(\nu|\nu_1)^2 \right\rangle =$$
$$= \left( \frac{1}{v^2} - \frac{1}{v^2} \frac{v-1}{v^s - 1} \right) \left( \frac{1}{v^2} + \sigma_p^2 \right),$$

$$\left\langle p_{i_1}^{(1)}(\mu|\mu_1) p_{i_1}^{(1)}(\mu|\lambda_1) p_{j_1}^{(1)}(\nu|\nu_1) p_{j_1}^{(1)}(\nu|\kappa_1) \right\rangle = \left( \frac{1}{v^2} - \frac{1}{v^2} \frac{v-1}{v^s - 1} \right)^2. \tag{80}$$

By bringing these factors outside of the $\lambda_1$ and $\kappa_1$ sums in the right-hand side of Eq. 79,

$$\left\langle \left( C^{(2)}(\mu,\nu) \right)^2 \right\rangle = \sum_{\mu_1,\nu_1} \left( \left( \frac{1}{v^2} + \sigma_p^2 \right)^2 \left\langle C^{(1)}(\mu_1,\nu_1)^2 \right\rangle \right.$$
$$+ \left( \frac{1}{v^2} + \sigma_p^2 \right) \left( \frac{1}{v^2} - \frac{1}{v^2} \frac{v-1}{v^s - 1} \right) \sum_{\kappa_1 \neq \nu_1} \left\langle C^{(1)}(\mu_1,\nu_1) C^{(1)}(\mu_1,\kappa_1) \right\rangle$$
$$+ \left( \frac{1}{v^2} + \sigma_p^2 \right) \left( \frac{1}{v^2} - \frac{1}{v^2} \frac{v-1}{v^s - 1} \right) \sum_{\lambda_1 \neq \mu_1} \left\langle C^{(1)}(\mu_1,\nu_1) C^{(1)}(\lambda_1,\nu_1) \right\rangle$$
$$+ \left( \frac{1}{v^2} - \frac{1}{v^2} \frac{v-1}{v^s - 1} \right)^2 \sum_{\lambda_1 \neq \mu_1, \kappa_1 \neq \nu_1} \left\langle C^{(1)}(\mu_1,\nu_1) C^{(1)}(\lambda_1,\kappa_1) \right\rangle$$
$$= \sum_{\mu_1,\nu_1} \left\langle C^{(1)}(\mu_1,\nu_1)^2 \right\rangle \left[ \left( \frac{1}{v^2} + \sigma_p^2 \right) - \left( \frac{1}{v^2} - \frac{1}{v^2} \frac{v-1}{v^s - 1} \right) \right]^2, \tag{81}$$

where, in the last line, we used Eq. 73, Eq. 74 and Eq. 75 to express the covariances of the $C^{(1)}(\mu,\nu)$'s in terms of $\left\langle C^{(1)}(\mu,\nu)^2 \right\rangle$. After simple algebraic steps, recalling the definition of $\sigma_p^2$ in Eq. 29,

$$\left\langle \left( C^{(2)}(\mu,\nu) \right)^2 \right\rangle = v^2 \left\langle C^{(1)}(\mu_1,\nu_1)^2 \right\rangle \left( \frac{v-1}{v} \frac{v^s}{v^s - 1} \frac{1}{vm} \right)^2 \xrightarrow{v \gg 1} \frac{\left\langle C^{(1)}(\mu_1,\nu_1)^2 \right\rangle}{m^2}. \tag{82}$$

### D.2.2 $i_1 = j_1$ case.

In this case, we need to evaluate some four-point functions. Since the spatial index of the $p$'s is the same, we will drop it to ease the notation. For the same reason, we will drop the level index too. First, it is convenient to use Eq. 73, Eq. 74 and Eq. 75 to rearrange the right-hand side of Eq. 79 as follows,

$$\left\langle \left( C^{(2)}(\mu,\nu) \right)^2 \right\rangle = \sum_{\mu_1,\nu_1} \left\langle C^{(1)}(\mu_1,\nu_1)^2 \right\rangle \left( \left\langle p(\mu|\mu_1)^2 p(\nu|\nu_1)^2 \right\rangle \right.$$
$$- \frac{1}{v-1} \sum_{\kappa_1 \neq \nu_1} \left\langle p(\mu|\mu_1)^2 p(\nu|\nu_1) p(\nu|\kappa_1) \right\rangle$$
$$- \frac{1}{v-1} \sum_{\lambda_1 \neq \mu_1} \left\langle p(\mu|\mu_1) p(\mu|\lambda_1) p(\nu|\nu_1)^2 \right\rangle$$
$$\left. + \frac{1}{(v-1)^2} \sum_{\lambda_1 \neq \mu_1, \kappa_1 \neq \nu_1} \left\langle p(\mu|\mu_1) p(\mu|\lambda_1) p(\nu|\nu_1) p(\nu|\kappa_1) \right\rangle \right). \tag{83}$$

The value of $\langle C^{(1)}(\mu_1,\nu_1)^2 \rangle$ is actually independent of $\mu_1$ and $\nu_1$, thus

$$\left\langle \left(C^{(2)}(\mu,\nu)\right)^2 \right\rangle = \left\langle (C^{(1)})^2 \right\rangle \sum_{\mu_1,\nu_1} \left( \langle p(\mu|\mu_1)^2 p(\nu|\nu_1)^2 \rangle \right.$$

$$- \frac{1}{v-1} \sum_{\kappa_1 \neq \nu_1} \langle p(\mu|\mu_1)^2 p(\nu|\nu_1) p(\nu|\kappa_1) \rangle - \frac{1}{v-1} \sum_{\lambda_1 \neq \mu_1} \langle p(\mu|\mu_1) p(\mu|\lambda_1) p(\nu|\nu_1)^2 \rangle$$

$$\left. + \frac{1}{(v-1)^2} \sum_{\lambda_1 \neq \mu_1, \kappa_1 \neq \nu_1} \langle p(\mu|\mu_1) p(\mu|\lambda_1) p(\nu|\nu_1) p(\nu|\kappa_1) \rangle \right). \tag{84}$$

The first term to deal with is (2-2),

$$\langle p(\mu|\mu_1)^2 p(\nu|\nu_1)^2 \rangle = \frac{1}{m^4} \sum_{\psi_1,\psi_2,\psi_3,\psi_4} \mathbb{P}\left\{ \mu_1 \xrightarrow{\psi_1} \mu; \mu_1 \xrightarrow{\psi_2} \mu; \nu_1 \xrightarrow{\psi_3} \nu; \nu_1 \xrightarrow{\psi_4} \nu \right\}; \tag{85}$$

then (2-1-1) and (1-1-2),

$$\langle p(\mu|\mu_1)^2 p(\nu|\nu_1) p(\nu|\kappa_1) \rangle = \frac{1}{m^4} \sum_{\psi_1,\psi_2,\psi_3,\psi_4} \mathbb{P}\left\{ \mu_1 \xrightarrow{\psi_1} \mu; \mu_1 \xrightarrow{\psi_2} \mu; \nu_1 \xrightarrow{\psi_3} \nu; \kappa_1 \xrightarrow{\psi_4} \nu \right\}; \tag{86}$$

$$\langle p(\mu|\mu_1) p(\mu|\lambda_1) p(\nu|\nu_1)^2 \rangle = \frac{1}{m^4} \sum_{\psi_1,\psi_2,\psi_3,\psi_4} \mathbb{P}\left\{ \mu_1 \xrightarrow{\psi_1} \mu; \lambda_1 \xrightarrow{\psi_2} \mu; \nu_1 \xrightarrow{\psi_3} \nu; \nu_1 \xrightarrow{\psi_4} \nu \right\}; \tag{87}$$

and, finally, (1-1-1-1),

$$\langle p(\mu|\mu_1) p(\mu|\lambda_1) p(\nu|\nu_1) p(\nu|\kappa_1) \rangle = \frac{1}{m^4} \sum_{\psi_1,\psi_2,\psi_3,\psi_4} \mathbb{P}\left\{ \mu_1 \xrightarrow{\psi_1} \mu; \lambda_1 \xrightarrow{\psi_2} \mu; \nu_1 \xrightarrow{\psi_3} \nu; \kappa_1 \xrightarrow{\psi_4} \nu \right\}. \tag{88}$$

We will further separate the case where $\mu = \nu$ (i) from the case $\mu \neq \nu$ (ii).

**2-2, i-a)** ($\mu = \nu$, $\mu_1 = \nu_1$)

$$\langle p(\mu|\mu_1)^2 p(\nu|\nu_1)^2 \rangle = \frac{1}{m^4} \sum_{\psi_1,\psi_2,\psi_3,\psi_4} \mathbb{P}\left\{ \mu_1 \xrightarrow{\psi_1} \mu; \mu_1 \xrightarrow{\psi_2} \mu; \mu_1 \xrightarrow{\psi_3} \mu; \mu_1 \xrightarrow{\psi_4} \mu \right\}$$

$$= \frac{1}{m^4} \sum_{\psi_1,\psi_2=\psi_3=\psi_4=\psi_1} \mathbb{P}\left\{ \mu_1 \xrightarrow{\psi_1} \mu \right\}$$

$$+ \frac{4}{m^4} \sum_{\psi_1,\psi_2 \neq \psi_1,\psi_3=\psi_4=\psi_1} \mathbb{P}\left\{ \mu_1 \xrightarrow{\psi_1} \mu; \mu_1 \xrightarrow{\psi_2} \mu \right\}$$

$$+ \frac{3}{m^4} \sum_{\psi_1,\psi_2=\psi_1,\psi_3 \neq \psi_1,\psi_4=\psi_3} \mathbb{P}\left\{ \mu_1 \xrightarrow{\psi_1} \mu; \mu_1 \xrightarrow{\psi_3} \mu \right\}$$

$$+ \frac{6}{m^4} \sum_{\psi_1,\psi_2 \neq \psi_1,\psi_3 \neq (\psi_1,\psi_2),\psi_4=\psi_1} \mathbb{P}\left\{ \mu_1 \xrightarrow{\psi_1} \mu; \mu_1 \xrightarrow{\psi_2} \mu; \mu_1 \xrightarrow{\psi_3} \mu \right\}$$

$$+ \frac{1}{m^4} \sum_{\psi_1,\psi_2 \neq \psi_1,\psi_3 \neq (\psi_1,\psi_2),\psi_4 \neq (\psi_1,\psi_2,\psi_3)} \mathbb{P}\left\{ \mu_1 \xrightarrow{\psi_1} \mu; \mu_1 \xrightarrow{\psi_2} \mu; \mu_1 \xrightarrow{\psi_3} \mu, \mu_1 \xrightarrow{\psi_4} \mu \right\}$$

$$= \frac{1}{m^4}\left[ \frac{m}{v} + 7\frac{m(m-1)}{v}\frac{v^{s-1}-1}{v^s-1} + 6\frac{m(m-1)(m-2)}{v}\frac{v^{s-1}-1}{v^s-1}\frac{v^{s-1}-2}{v^s-2} \right.$$

$$\left. + \frac{m(m-1)(m-2)(m-3)}{v}\frac{v^{s-1}-1}{v^s-1}\frac{v^{s-1}-2}{v^s-2}\frac{v^{s-1}-3}{v^s-3} \right] \tag{89}$$

**2-2, i-b)** ($\mu = \nu$; $\mu_1 \neq \nu_1$)

$$
\begin{aligned}
\left\langle p(\mu|\mu_1)^2 p(\nu|\nu_1)^2 \right\rangle &= \frac{1}{m^4} \sum_{\psi_1,\psi_2,\psi_3,\psi_4} \mathbb{P}\left\{ \mu_1 \xrightarrow{\psi_1} \mu; \mu_1 \xrightarrow{\psi_2} \mu; \nu_1 \xrightarrow{\psi_3} \mu; \nu_1 \xrightarrow{\psi_4} \mu \right\} \\
&= \frac{1}{m^4} \sum_{\psi_1,\psi_2=\psi_1,\psi_3,\psi_4=\psi_3} \mathbb{P}\left\{ \mu_1 \xrightarrow{\psi_1} \mu; \nu_1 \xrightarrow{\psi_3} \mu \right\} \\
&+ \frac{1}{m^4} \sum_{\psi_1,\psi_2=\psi_1,\psi_3,\psi_4\neq\psi_3} \mathbb{P}\left\{ \mu_1 \xrightarrow{\psi_1} \mu; \nu_1 \xrightarrow{\psi_3} \mu; \nu_1 \xrightarrow{\psi_4} \mu \right\} \\
&+ \frac{1}{m^4} \sum_{\psi_1,\psi_2\neq\psi_1,\psi_3,\psi_4=\psi_3} \mathbb{P}\left\{ \mu_1 \xrightarrow{\psi_1} \mu; \mu_1 \xrightarrow{\psi_2} \mu; \nu_1 \xrightarrow{\psi_3} \mu \right\} \\
&+ \frac{1}{m^4} \sum_{\psi_1,\psi_2\neq\psi_1,\psi_3,\psi_4\neq\psi_3} \mathbb{P}\left\{ \mu_1 \xrightarrow{\psi_1} \mu; \mu_1 \xrightarrow{\psi_2} \mu; \nu_1 \xrightarrow{\psi_3} \mu; \nu_1 \xrightarrow{\psi_4} \mu \right\} \\
&= \frac{1}{m^4} \left[ \frac{m^2}{v} \frac{v^{s-1}-1}{v^s-1} + 2\frac{m^2(m-1)}{v} \frac{v^{s-1}-1}{v^s-1} \frac{v^{s-1}-2}{v^s-2} \right. \\
&\left. + \frac{m^2(m-1)^2}{v} \frac{v^{s-1}-1}{v^s-1} \frac{v^{s-1}-2}{v^s-2} \frac{v^{s-1}-3}{v^s-3} \right]
\end{aligned} \tag{90}
$$

**2-2, ii-a)** ($\mu \neq \nu$, $\mu_1 = \nu_1$)

$$
\begin{aligned}
\left\langle p(\mu|\mu_1)^2 p(\nu|\nu_1)^2 \right\rangle &= \frac{1}{m^4} \sum_{\psi_1,\psi_2,\psi_3,\psi_4} \mathbb{P}\left\{ \mu_1 \xrightarrow{\psi_1} \mu; \mu_1 \xrightarrow{\psi_2} \mu; \mu_1 \xrightarrow{\psi_3} \nu; \mu_1 \xrightarrow{\psi_4} \nu \right\} \\
&+ \frac{1}{m^4} \sum_{\psi_1,\psi_2=\psi_1,\psi_3\neq\psi_1,\psi_4=\psi_3} \mathbb{P}\left\{ \mu_1 \xrightarrow{\psi_1} \mu; \mu_1 \xrightarrow{\psi_3} \nu \right\} \\
&+ \frac{1}{m^4} \sum_{\psi_1,\psi_2\neq\psi_1,\psi_3\neq(\psi_1,\psi_2),\psi_4=\psi_3} \mathbb{P}\left\{ \mu_1 \xrightarrow{\psi_1} \mu; \mu_1 \xrightarrow{\psi_2} \mu; \mu_1 \xrightarrow{\psi_3} \nu \right\} \\
&+ \frac{1}{m^4} \sum_{\psi_1,\psi_2=\psi_1,\psi_3\neq\psi_1,\psi_4\neq(\psi_3,\psi_1)} \mathbb{P}\left\{ \mu_1 \xrightarrow{\psi_1} \mu; \mu_1 \xrightarrow{\psi_3} \nu; \mu_1 \xrightarrow{\psi_4} \nu \right\} \\
&+ \frac{1}{m^4} \sum_{\psi_1,\psi_2\neq\psi_1,\psi_3\neq(\psi_1,\psi_2),\psi_4\neq(\psi_1,\psi_2,\psi_3)} \mathbb{P}\left\{ \mu_1 \xrightarrow{\psi_1} \mu; \mu_1 \xrightarrow{\psi_2} \mu; \mu_1 \xrightarrow{\psi_3} \nu, \mu_1 \xrightarrow{\psi_4} \nu \right\} \\
&= \frac{1}{m^4} \left[ \frac{m(m-1)}{v} \frac{v^{s-1}}{v^s-1} + 2\frac{m(m-1)(m-2)}{v} \frac{v^{s-1}}{v^s-1} \frac{v^{s-1}-1}{v^s-2} \right. \\
&\left. + \frac{m(m-1)(m-2)(m-3)}{v} \frac{v^{s-1}}{v^s-1} \frac{v^{s-1}-1}{v^s-2} \frac{v^{s-1}-1}{v^s-3} \right]
\end{aligned} \tag{91}
$$

**2-2, ii-b)** ($\mu \neq \nu$; $\mu_1 \neq \nu_1$)

$$\langle p(\mu|\mu_1)^2 p(\nu|\nu_1)^2 \rangle = \frac{1}{m^4} \sum_{\psi_1,\psi_2,\psi_3,\psi_4} \mathbb{P}\left\{\mu_1 \xrightarrow{\psi_1} \mu; \mu_1 \xrightarrow{\psi_2} \mu; \nu_1 \xrightarrow{\psi_3} \nu; \nu_1 \xrightarrow{\psi_4} \nu\right\}$$

$$= \frac{1}{m^4} \sum_{\psi_1,\psi_2=\psi_1,\psi_3,\psi_4=\psi_3} \mathbb{P}\left\{\mu_1 \xrightarrow{\psi_1} \mu; \nu_1 \xrightarrow{\psi_3} \nu\right\}$$

$$+ \frac{1}{m^4} \sum_{\psi_1,\psi_2=\psi_1,\psi_3,\psi_4\neq\psi_3} \mathbb{P}\left\{\mu_1 \xrightarrow{\psi_1} \mu; \nu_1 \xrightarrow{\psi_3} \nu; \nu_1 \xrightarrow{\psi_4} \nu\right\}$$

$$+ \frac{1}{m^4} \sum_{\psi_1,\psi_2\neq\psi_1,\psi_3,\psi_4=\psi_3} \mathbb{P}\left\{\mu_1 \xrightarrow{\psi_1} \mu; \mu_1 \xrightarrow{\psi_2} \mu; \nu_1 \xrightarrow{\psi_3} \nu\right\}$$

$$+ \frac{1}{m^4} \sum_{\psi_1,\psi_2\neq\psi_1,\psi_3,\psi_4\neq\psi_3} \mathbb{P}\left\{\mu_1 \xrightarrow{\psi_1} \mu; \mu_1 \xrightarrow{\psi_2} \mu; \nu_1 \xrightarrow{\psi_3} \nu; \nu_1 \xrightarrow{\psi_4} \nu\right\}$$

$$= \frac{1}{m^4}\left[\frac{m^2}{v}\frac{v^{s-1}}{v^s-1} + 2\frac{m^2(m-1)}{v}\frac{v^{s-1}}{v^s-1}\frac{v^{s-1}-1}{v^s-2}\right.$$

$$\left.+ \frac{m^2(m-1)^2}{v}\frac{v^{s-1}-1}{v^s-1}\frac{v^{s-1}-1}{v^s-2}\frac{v^{s-1}-1}{v^s-3}\right]. \tag{92}$$

**2-1-1, i-a)** ($\mu = \nu$, $\mu_1 = \nu_1$)

$$\langle p(\mu|\mu_1)^2 p(\nu|\nu_1) p(\nu|\kappa_1) \rangle = \frac{1}{m^4} \sum_{\psi_1,\psi_2,\psi_3,\psi_4} \mathbb{P}\left\{\mu_1 \xrightarrow{\psi_1} \mu; \mu_1 \xrightarrow{\psi_2} \mu; \mu_1 \xrightarrow{\psi_3} \mu; \kappa_1 \xrightarrow{\psi_4} \mu\right\}$$

$$= \frac{1}{m^4} \sum_{\psi_1,\psi_2=\psi_1,\psi_3=\psi_1,\psi_4} \mathbb{P}\left\{\mu_1 \xrightarrow{\psi_1} \mu; \kappa_1 \xrightarrow{\psi_4} \mu\right\}$$

$$+ \frac{3}{m^4} \sum_{\psi_1,\psi_2=\psi_1,\psi_3\neq\psi_1,\psi_4} \mathbb{P}\left\{\mu_1 \xrightarrow{\psi_1} \mu; \mu_1 \xrightarrow{\psi_3} \mu; \kappa_1 \xrightarrow{\psi_4} \mu\right\}$$

$$+ \frac{1}{m^4} \sum_{\psi_1,\psi_2\neq\psi_1,\psi_3\neq(\psi_1,\psi_2),\psi_4} \mathbb{P}\left\{\mu_1 \xrightarrow{\psi_1} \mu; \mu_1 \xrightarrow{\psi_2} \mu; \mu_1 \xrightarrow{\psi_3} \mu; \kappa_1 \xrightarrow{\psi_4} \mu\right\}$$

$$= \frac{1}{m^4}\left[\frac{m^2}{v}\frac{v^{s-1}-1}{v^s-1} + 3\frac{m^2(m-1)}{v}\frac{v^{s-1}-1}{v^s-1}\frac{v^{s-1}-2}{v^s-2}\right.$$

$$\left.+ \frac{m^2(m-1)(m-2)}{v}\frac{v^{s-1}-1}{v^s-1}\frac{v^{s-1}-2}{v^s-2}\frac{v^{s-1}-3}{v^s-3}\right]. \tag{93}$$

**2-1-1, i-b)** ($\mu = \nu$, $\nu_1 \neq \mu_1$, $\kappa_1 = \mu_1$)

$$\langle p(\mu|\mu_1)^2 p(\nu|\nu_1) p(\nu|\kappa_1) \rangle = \frac{1}{m^4} \sum_{\psi_1,\psi_2,\psi_3,\psi_4} \mathbb{P}\left\{\mu_1 \xrightarrow{\psi_1} \mu; \mu_1 \xrightarrow{\psi_2} \mu; \nu_1 \xrightarrow{\psi_3} \mu; \mu_1 \xrightarrow{\psi_4} \mu\right\}$$

$$= \frac{1}{m^4} \sum_{\psi_1,\psi_2=\psi_1,\psi_3,\psi_4=\psi_1} \mathbb{P}\left\{\mu_1 \xrightarrow{\psi_1} \mu; \nu_1 \xrightarrow{\psi_3} \mu\right\}$$

$$+ \frac{3}{m^4} \sum_{\psi_1,\psi_2=\psi_1,\psi_3,\psi_4\neq\psi_1} \mathbb{P}\left\{\mu_1 \xrightarrow{\psi_1} \mu; \mu_1 \xrightarrow{\psi_4} \mu; \nu_1 \xrightarrow{\psi_3} \mu\right\}$$

$$+ \frac{1}{m^4} \sum_{\psi_1,\psi_2\neq\psi_1,\psi_3,\psi_4\neq(\psi_1,\psi_2)} \mathbb{P}\left\{\mu_1 \xrightarrow{\psi_1} \mu; \mu_1 \xrightarrow{\psi_2} \mu; \mu_1 \xrightarrow{\psi_4} \mu; \nu_1 \xrightarrow{\psi_3} \mu\right\}$$

$$= \frac{1}{m^4}\left[\frac{m^2}{v}\frac{v^{s-1}-1}{v^s-1} + 3\frac{m^2(m-1)}{v}\frac{v^{s-1}-1}{v^s-1}\frac{v^{s-1}-2}{v^s-2}\right.$$

$$\left.+ \frac{m^2(m-1)(m-2)}{v}\frac{v^{s-1}-1}{v^s-1}\frac{v^{s-1}-2}{v^s-2}\frac{v^{s-1}-3}{v^s-3}\right], \tag{94}$$

equal to the value of 2-1-1, *i-a)* as it should be.

**2-1-1, i-c)** $(\mu = \nu,\, \nu_1 \neq \mu_1,\, \kappa_1 \neq (\mu_1, \nu_1))$

$$\left\langle p(\mu|\mu_1)^2 p(\nu|\nu_1) p(\nu|\kappa_1) \right\rangle = \frac{1}{m^4} \sum_{\psi_1, \psi_2, \psi_3, \psi_4} \mathbb{P}\left\{ \mu_1 \xrightarrow{\psi_1} \mu; \mu_1 \xrightarrow{\psi_2} \mu; \nu_1 \xrightarrow{\psi_3} \mu; \kappa_1 \xrightarrow{\psi_4} \mu \right\}$$

$$= \frac{1}{m^4} \sum_{\psi_1, \psi_2 = \psi_1, \psi_3, \psi_4} \mathbb{P}\left\{ \mu_1 \xrightarrow{\psi_1} \mu; \nu_1 \xrightarrow{\psi_3} \mu; \kappa_1 \xrightarrow{\psi_4} \mu \right\}$$

$$+ \frac{1}{m^4} \sum_{\psi_1, \psi_2 \neq \psi_1, \psi_3, \psi_4} \mathbb{P}\left\{ \mu_1 \xrightarrow{\psi_1} \mu; \mu_1 \xrightarrow{\psi_2} \mu; \nu_1 \xrightarrow{\psi_3} \mu; \kappa_1 \xrightarrow{\psi_4} \mu \right\}$$

$$= \frac{1}{m^4} \left[ \frac{m^3}{v} \frac{v^{s-1}-1}{v^s-1} \frac{v^{s-1}-2}{v^s-2} + \frac{m^3(m-1)}{v} \frac{v^{s-1}-1}{v^s-1} \frac{v^{s-1}-2}{v^s-2} \frac{v^{s-1}-3}{v^s-3} \right]$$

$$\tag{95}$$

**2-1-1, ii-a)** $(\mu \neq \nu,\, \mu_1 = \nu_1)$

$$\left\langle p(\mu|\mu_1)^2 p(\nu|\nu_1) p(\nu|\kappa_1) \right\rangle = \frac{1}{m^4} \sum_{\psi_1, \psi_2, \psi_3, \psi_4} \mathbb{P}\left\{ \mu_1 \xrightarrow{\psi_1} \mu; \mu_1 \xrightarrow{\psi_2} \mu; \mu_1 \xrightarrow{\psi_3} \nu; \kappa_1 \xrightarrow{\psi_4} \nu \right\}$$

$$= \frac{1}{m^4} \sum_{\psi_1, \psi_2 = \psi_1, \psi_3 \neq \psi_1, \psi_4} \mathbb{P}\left\{ \mu_1 \xrightarrow{\psi_1} \mu; \mu_1 \xrightarrow{\psi_3} \nu; \kappa_1 \xrightarrow{\psi_4} \nu \right\}$$

$$+ \frac{1}{m^4} \sum_{\psi_1, \psi_2 \neq \psi_1, \psi_3 \neq (\psi_1, \psi_2), \psi_4} \mathbb{P}\left\{ \mu_1 \xrightarrow{\psi_1} \mu; \mu_1 \xrightarrow{\psi_2} \mu; \mu_1 \xrightarrow{\psi_3} \nu; \kappa_1 \xrightarrow{\psi_4} \nu \right\}$$

$$= \frac{1}{m^4} \left[ \frac{m^2(m-1)}{v} \frac{v^{s-1}}{v^s-1} \frac{v^{s-1}-1}{v^s-2} \right.$$

$$\left. + \frac{m^2(m-1)(m-2)}{v} \frac{v^{s-1}-1}{v^s-1} \frac{v^{s-1}}{v^s-2} \frac{v^{s-1}-1}{v^s-3} \right]. \tag{96}$$

**2-1-1, ii-b)** $(\mu \neq \nu,\, \nu_1 \neq \mu_1,\, \kappa_1 = \mu_1)$

$$\left\langle p(\mu|\mu_1)^2 p(\nu|\nu_1) p(\nu|\kappa_1) \right\rangle = \frac{1}{m^4} \sum_{\psi_1, \psi_2, \psi_3, \psi_4} \mathbb{P}\left\{ \mu_1 \xrightarrow{\psi_1} \mu; \mu_1 \xrightarrow{\psi_2} \mu; \nu_1 \xrightarrow{\psi_3} \nu; \mu_1 \xrightarrow{\psi_4} \nu \right\}$$

$$= \frac{1}{m^4} \sum_{\psi_1, \psi_2 = \psi_1, \psi_3, \psi_4 \neq \psi_1} \mathbb{P}\left\{ \mu_1 \xrightarrow{\psi_1} \mu; \nu_1 \xrightarrow{\psi_3} \nu; \mu_1 \xrightarrow{\psi_4} \nu \right\}$$

$$+ \frac{1}{m^4} \sum_{\psi_1, \psi_2 \neq \psi_1, \psi_3, \psi_4 \neq (\psi_1, \psi_2)} \mathbb{P}\left\{ \mu_1 \xrightarrow{\psi_1} \mu; \mu_1 \xrightarrow{\psi_2} \mu; \nu_1 \xrightarrow{\psi_3} \nu; \mu_1 \xrightarrow{\psi_4} \nu \right\}$$

$$= \frac{1}{m^4} \left[ \frac{m^2(m-1)}{v} \frac{v^{s-1}}{v^s-1} \frac{v^{s-1}-1}{v^s-2} \right.$$

$$\left. + \frac{m^2(m-1)(m-2)}{v} \frac{v^{s-1}-1}{v^s-1} \frac{v^{s-1}}{v^s-2} \frac{v^{s-1}-1}{v^s-3} \right]. \tag{97}$$

**2-1-1, ii-c)** $(\mu \neq \nu,\, \nu_1 \neq \mu_1,\, \kappa_1 \neq (\mu_1, \nu_1))$

$$\left\langle p(\mu|\mu_1)^2 p(\nu|\nu_1) p(\nu|\kappa_1) \right\rangle = \frac{1}{m^4} \sum_{\psi_1, \psi_2, \psi_3, \psi_4} \mathbb{P}\left\{ \mu_1 \xrightarrow{\psi_1} \mu; \mu_1 \xrightarrow{\psi_2} \mu; \nu_1 \xrightarrow{\psi_3} \nu; \kappa_1 \xrightarrow{\psi_4} \nu \right\}$$

$$= \frac{1}{m^4} \sum_{\psi_1, \psi_2 = \psi_1, \psi_3, \psi_4} \mathbb{P}\left\{ \mu_1 \xrightarrow{\psi_1} \mu; \nu_1 \xrightarrow{\psi_3} \nu; \kappa_1 \xrightarrow{\psi_4} \nu \right\}$$

$$+ \frac{1}{m^4} \sum_{\psi_1, \psi_2 \neq \psi_1, \psi_3, \psi_4} \mathbb{P}\left\{ \mu_1 \xrightarrow{\psi_1} \mu; \mu_1 \xrightarrow{\psi_2} \mu; \nu_1 \xrightarrow{\psi_3} \nu; \kappa_1 \xrightarrow{\psi_4} \nu \right\}$$

$$= \frac{1}{m^4} \left[ \frac{m^3}{v} \frac{v^{s-1}}{v^s-1} \frac{v^{s-1}-1}{v^s-2} + \frac{m^3(m-1)}{v} \frac{v^{s-1}-1}{v^s-1} \frac{v^{s-1}}{v^s-2} \frac{v^{s-1}-1}{v^s-3} \right]. \tag{98}$$

**1-1-2,** overall contribution equal to that of 2-1-1.

**1-1-1-1, i-a)** ($\mu = \nu$, $\mu_1 = \nu_1$, $\lambda_1 = \kappa_1$, $(v-1)$ **of the** $(v-1)^2$ **choices of** $\lambda_1, \kappa_1$**)**

$$
\begin{aligned}
\langle p(\mu|\mu_1)p(\mu|\lambda_1)p(\nu|\nu_1)p(\nu|\kappa_1)\rangle &= \frac{1}{m^4} \sum_{\psi_1,\psi_2,\psi_3,\psi_4} \mathbb{P}\left\{\mu_1 \xrightarrow{\psi_1} \mu; \lambda_1 \xrightarrow{\psi_2} \mu; \mu_1 \xrightarrow{\psi_3} \mu; \lambda_1 \xrightarrow{\psi_4} \mu\right\} \\
&= \langle p(\mu|\mu_1)^2 p(\mu|\lambda_1)^2 \rangle \\
&= \frac{1}{m^4}\left[\frac{m^2}{v}\frac{v^{s-1}-1}{v^s-1} + 2\frac{m^2(m-1)}{v}\frac{v^{s-1}-1}{v^s-1}\frac{v^{s-1}-2}{v^s-2}\right. \\
&\quad \left. + \frac{m^2(m-1)^2}{v}\frac{v^{s-1}-1}{v^s-1}\frac{v^{s-1}-2}{v^s-2}\frac{v^{s-1}-3}{v^s-3}\right]
\end{aligned}
\tag{99}
$$

from the value of 2-2, case *i-b)*.

**1-1-1-1, i-b)** ($\mu = \nu$, $\mu_1 = \nu_1$, $\lambda_1 \neq \kappa_1$, $(v-1)(v-2)$ **of the** $(v-1)^2$ **choices of** $\lambda_1, \kappa_1$**)**

$$
\begin{aligned}
\langle p(\mu|\mu_1)p(\mu|\lambda_1)p(\nu|\nu_1)p(\nu|\kappa_1)\rangle &= \frac{1}{m^4} \sum_{\psi_1,\psi_2,\psi_3,\psi_4} \mathbb{P}\left\{\mu_1 \xrightarrow{\psi_1} \mu; \lambda_1 \xrightarrow{\psi_2} \mu; \mu_1 \xrightarrow{\psi_3} \mu; \kappa_1 \xrightarrow{\psi_4} \mu\right\} \\
&= \frac{1}{m^4}\left[\frac{m^3}{v}\frac{v^{s-1}-1}{v^s-1}\frac{v^{s-1}-2}{v^s-2} + \frac{m^3(m-1)}{v}\frac{v^{s-1}-1}{v^s-1}\frac{v^{s-1}-2}{v^s-2}\frac{v^{s-1}-3}{v^s-3}\right]
\end{aligned}
\tag{100}
$$

from the value of 2-1-1, case *i-c)*.

**1-1-1-1, i-c)** ($\mu = \nu$, $\mu_1 \neq \nu_1$, $\lambda_1 = \kappa_1$, $v-2$ **of the** $(v-1)^2$ **choices of** $\lambda_1, \kappa_1$**)**

$$
\begin{aligned}
\langle p(\mu|\mu_1)p(\mu|\lambda_1)p(\nu|\nu_1)p(\nu|\kappa_1)\rangle &= \frac{1}{m^4} \sum_{\psi_1,\psi_2,\psi_3,\psi_4} \mathbb{P}\left\{\mu_1 \xrightarrow{\psi_1} \mu; \lambda_1 \xrightarrow{\psi_2} \mu; \nu_1 \xrightarrow{\psi_3} \mu; \lambda_1 \xrightarrow{\psi_4} \mu\right\} \\
&= \frac{1}{m^4}\left[\frac{m^3}{v}\frac{v^{s-1}-1}{v^s-1}\frac{v^{s-1}-2}{v^s-2} + \frac{m^3(m-1)}{v}\frac{v^{s-1}-1}{v^s-1}\frac{v^{s-1}-2}{v^s-2}\frac{v^{s-1}-3}{v^s-3}\right]
\end{aligned}
\tag{101}
$$

from the value of 2-1-1, case *i-c)*.

**1-1-1-1, i-d)** ($\mu = \nu$, $\mu_1 \neq \nu_1$, $\lambda_1 = \nu_1$, $\kappa_1 = \mu_1$, $1$ **of the** $(v-1)^2$ **choices of** $\lambda_1, \kappa_1$**)**

$$
\begin{aligned}
\langle p(\mu|\mu_1)p(\mu|\lambda_1)p(\nu|\nu_1)p(\nu|\kappa_1)\rangle &= \frac{1}{m^4} \sum_{\psi_1,\psi_2,\psi_3,\psi_4} \mathbb{P}\left\{\mu_1 \xrightarrow{\psi_1} \mu; \nu_1 \xrightarrow{\psi_2} \mu; \nu_1 \xrightarrow{\psi_3} \mu; \mu_1 \xrightarrow{\psi_4} \mu\right\} \\
&= \frac{1}{m^4}\left[\frac{m^2}{v}\frac{v^{s-1}-1}{v^s-1} + 2\frac{m^2(m-1)}{v}\frac{v^{s-1}-1}{v^s-1}\frac{v^{s-1}-2}{v^s-2}\right. \\
&\quad \left. + \frac{m^2(m-1)^2}{v}\frac{v^{s-1}-1}{v^s-1}\frac{v^{s-1}-2}{v^s-2}\frac{v^{s-1}-3}{v^s-3}\right]
\end{aligned}
\tag{102}
$$

from the value of 2-2, case *i-b)*

**1-1-1-1, i-e)** ($\mu = \nu$, $\mu_1 \neq \nu_1$, $\lambda_1 = \nu_1$, $\kappa_1 \neq (\mu_1, \nu_1)$, $v-2$ **of the** $(v-1)^2$ **choices of** $\lambda_1, \kappa_1$**)**

$$
\begin{aligned}
\langle p(\mu|\mu_1)p(\mu|\lambda_1)p(\nu|\nu_1)p(\nu|\kappa_1)\rangle &= \frac{1}{m^4} \sum_{\psi_1,\psi_2,\psi_3,\psi_4} \mathbb{P}\left\{\mu_1 \xrightarrow{\psi_1} \mu; \nu_1 \xrightarrow{\psi_2} \mu; \nu_1 \xrightarrow{\psi_3} \mu; \kappa_1 \xrightarrow{\psi_4} \mu\right\} \\
&= \frac{1}{m^4}\left[\frac{m^3}{v}\frac{v^{s-1}-1}{v^s-1}\frac{v^{s-1}-2}{v^s-2} + \frac{m^3(m-1)}{v}\frac{v^{s-1}-1}{v^s-1}\frac{v^{s-1}-2}{v^s-2}\frac{v^{s-1}-3}{v^s-3}\right]
\end{aligned}
\tag{103}
$$

from the value of 2-1-1, case *i-c)*.

**1-1-1-1, i-f)** ($\mu = \nu$, $\mu_1 \neq \nu_1$, $\lambda_1 \neq (\mu_1, \nu_1)$, $\kappa_1 = \mu_1$, $v - 2$ **of the** $(v-1)^2$ **choices of** $\lambda_1, \kappa_1$)

$$\langle p(\mu|\mu_1)p(\mu|\lambda_1)p(\nu|\nu_1)p(\nu|\kappa_1)\rangle = \frac{1}{m^4} \sum_{\psi_1,\psi_2,\psi_3,\psi_4} \mathbb{P}\left\{\mu_1 \xrightarrow{\psi_1} \mu; \lambda_1 \xrightarrow{\psi_2} \mu; \nu_1 \xrightarrow{\psi_3} \mu; \mu_1 \xrightarrow{\psi_4} \mu\right\}$$

$$= \frac{1}{m^4}\left[\frac{m^3}{v}\frac{v^{s-1}-1}{v^s-1}\frac{v^{s-1}-2}{v^s-2} + \frac{m^3(m-1)}{v}\frac{v^{s-1}-1}{v^s-1}\frac{v^{s-1}-2}{v^s-2}\frac{v^{s-1}-3}{v^s-3}\right]$$

(104)

from the value of 2-1-1, case *i-c)*.

**1-1-1-1, i-g)** ($\mu = \nu$, $\mu_1 \neq \nu_1$, $\lambda_1 \neq (\mu_1, \nu_1)$, $\kappa_1 = (\mu_1, \nu_1, \lambda_1)$, $(v-2)(v-3)$ **of the** $(v-1)^2$ **choices of** $\lambda_1, \kappa_1$)

$$\langle p(\mu|\mu_1)p(\mu|\lambda_1)p(\nu|\nu_1)p(\nu|\kappa_1)\rangle = \frac{1}{m^4} \sum_{\psi_1,\psi_2,\psi_3,\psi_4} \mathbb{P}\left\{\mu_1 \xrightarrow{\psi_1} \mu; \lambda_1 \xrightarrow{\psi_2} \mu; \nu_1 \xrightarrow{\psi_3} \mu; \kappa_1 \xrightarrow{\psi_4} \mu\right\}$$

$$= \frac{1}{m^4}\left[\frac{m^4}{v}\frac{v^{s-1}-1}{v^s-1}\frac{v^{s-1}-2}{v^s-2}\frac{v^{s-1}-3}{v^s-3}\right].$$

(105)

**1-1-1-1, ii-a)** ($\mu \neq \nu$, $\mu_1 = \nu_1$, $\lambda_1 = \kappa_1$, $(v-1)$ **of the** $(v-1)^2$ **choices of** $\lambda_1, \kappa_1$)

$$\langle p(\mu|\mu_1)p(\mu|\lambda_1)p(\nu|\nu_1)p(\nu|\kappa_1)\rangle = \frac{1}{m^4} \sum_{\psi_1,\psi_2,\psi_3,\psi_4} \mathbb{P}\left\{\mu_1 \xrightarrow{\psi_1} \mu; \lambda_1 \xrightarrow{\psi_2} \mu; \mu_1 \xrightarrow{\psi_3} \nu; \lambda_1 \xrightarrow{\psi_4} \nu\right\}$$

$$= \frac{1}{m^4} \sum_{\psi_1,\psi_2,\psi_3 \neq \psi_1,\psi_4 \neq \psi_2} \mathbb{P}\left\{\mu_1 \xrightarrow{\psi_1} \mu; \mu_1 \xrightarrow{\psi_3} \nu; \lambda_1 \xrightarrow{\psi_2} \mu; \lambda_1 \xrightarrow{\psi_4} \nu\right\}$$

$$= \frac{1}{m^4}\left[\frac{m^2(m-1)^2}{v}\frac{v^{s-1}-1}{v^s-1}\frac{v^{s-1}}{v^s-2}\frac{v^{s-1}-1}{v^s-3}\right].$$

(106)

**1-1-1-1, ii-b)** ($\mu \neq \nu$, $\mu_1 = \nu_1$, $\lambda_1 \neq \kappa_1$, $(v-1)(v-2)$ **of the** $(v-1)^2$ **choices of** $\lambda_1, \kappa_1$)

$$\langle p(\mu|\mu_1)p(\mu|\lambda_1)p(\nu|\nu_1)p(\nu|\kappa_1)\rangle = \frac{1}{m^4} \sum_{\psi_1,\psi_2,\psi_3,\psi_4} \mathbb{P}\left\{\mu_1 \xrightarrow{\psi_1} \mu; \lambda_1 \xrightarrow{\psi_2} \mu; \mu_1 \xrightarrow{\psi_3} \nu; \kappa_1 \xrightarrow{\psi_4} \nu\right\}$$

$$= \frac{1}{m^4} \sum_{\psi_1,\psi_2,\psi_3 \neq \psi_1,\psi_4} \mathbb{P}\left\{\mu_1 \xrightarrow{\psi_1} \mu; \lambda_1 \xrightarrow{\psi_2} \mu; \mu_1 \xrightarrow{\psi_3} \nu; \kappa_1 \xrightarrow{\psi_4} \nu\right\}$$

$$= \frac{1}{m^4}\left[\frac{m^3(m-1)}{v}\frac{v^{s-1}-1}{v^s-1}\frac{v^{s-1}}{v^s-2}\frac{v^{s-1}-1}{v^s-3}\right].$$

(107)

**1-1-1-1, ii-c)** ($\mu \neq \nu$, $\mu_1 \neq \nu_1$, $\lambda_1 = \kappa_1$, $v - 2$ **of the** $(v-1)^2$ **choices of** $\lambda_1, \kappa_1$)

$$\langle p(\mu|\mu_1)p(\mu|\lambda_1)p(\nu|\nu_1)p(\nu|\kappa_1)\rangle = \frac{1}{m^4} \sum_{\psi_1,\psi_2,\psi_3,\psi_4} \mathbb{P}\left\{\mu_1 \xrightarrow{\psi_1} \mu; \lambda_1 \xrightarrow{\psi_2} \mu; \nu_1 \xrightarrow{\psi_3} \nu; \lambda_1 \xrightarrow{\psi_4} \nu\right\}$$

$$= \frac{1}{m^4}\left[\frac{m^3(m-1)}{v}\frac{v^{s-1}-1}{v^s-1}\frac{v^{s-1}}{v^s-2}\frac{v^{s-1}-1}{v^s-3}\right],$$

(108)

from the value of 1-1-1-1, case *ii-b)*.

**1-1-1-1, ii-d)** ($\mu \neq \nu$, $\mu_1 \neq \nu_1$, $\lambda_1 = \nu_1$, $\kappa_1 = \mu_1$, $1$ **of the** $(v-1)^2$ **choices of** $\lambda_1, \kappa_1$)

$$\langle p(\mu|\mu_1)p(\mu|\lambda_1)p(\nu|\nu_1)p(\nu|\kappa_1)\rangle = \frac{1}{m^4} \sum_{\psi_1,\psi_2,\psi_3,\psi_4} \mathbb{P}\left\{\mu_1 \xrightarrow{\psi_1} \mu; \nu_1 \xrightarrow{\psi_2} \mu; \nu_1 \xrightarrow{\psi_3} \nu; \mu_1 \xrightarrow{\psi_4} \nu\right\}$$

$$= \frac{1}{m^4}\left[\frac{m^2(m-1)^2}{v}\frac{v^{s-1}-1}{v^s-1}\frac{v^{s-1}}{v^s-2}\frac{v^{s-1}-1}{v^s-3}\right],$$

(109)

from the value of 1-1-1-1, case *ii-a)*.

**1-1-1-1, ii-e)** ($\mu \neq \nu$, $\mu_1 \neq \nu_1$, $\lambda_1 = \nu_1$, $\kappa_1 \neq (\mu_1, \nu_1)$, $v - 2$ **of the** $(v-1)^2$ **choices of** $\lambda_1, \kappa_1$)

$$
\begin{aligned}
\langle p(\mu|\mu_1)p(\mu|\lambda_1)p(\nu|\nu_1)p(\nu|\kappa_1)\rangle &= \frac{1}{m^4} \sum_{\psi_1,\psi_2,\psi_3,\psi_4} \mathbb{P}\left\{\mu_1 \xrightarrow{\psi_1} \mu; \nu_1 \xrightarrow{\psi_2} \mu; \nu_1 \xrightarrow{\psi_3} \nu; \kappa_1 \xrightarrow{\psi_4} \nu\right\} \\
&= \frac{1}{m^4}\left[\frac{m^3(m-1)}{v}\frac{v^{s-1}-1}{v^s-1}\frac{v^{s-1}}{v^s-2}\frac{v^{s-1}-1}{v^s-3}\right],
\end{aligned}
\tag{110}
$$

from the value of 1-1-1-1, case *ii-b)*.

**1-1-1-1, ii-f)** ($\mu \neq \nu$, $\mu_1 \neq \nu_1$, $\lambda_1 \neq (\mu_1, \nu_1)$, $\kappa_1 = \mu_1$, $v - 2$ **of the** $(v-1)^2$ **choices of** $\lambda_1, \kappa_1$)

$$
\begin{aligned}
\langle p(\mu|\mu_1)p(\mu|\lambda_1)p(\nu|\nu_1)p(\nu|\kappa_1)\rangle &= \frac{1}{m^4} \sum_{\psi_1,\psi_2,\psi_3,\psi_4} \mathbb{P}\left\{\mu_1 \xrightarrow{\psi_1} \mu; \lambda_1 \xrightarrow{\psi_2} \mu; \nu_1 \xrightarrow{\psi_3} \nu; \mu_1 \xrightarrow{\psi_4} \nu\right\} \\
&= \frac{1}{m^4}\left[\frac{m^3(m-1)}{v}\frac{v^{s-1}-1}{v^s-1}\frac{v^{s-1}}{v^s-2}\frac{v^{s-1}-1}{v^s-3}\right],
\end{aligned}
\tag{111}
$$

from the value of 1-1-1-1, case *ii-b)*.

**1-1-1-1, ii-g)** ($\mu \neq \nu$, $\mu_1 \neq \nu_1$, $\lambda_1 \neq (\mu_1, \nu_1)$, $\kappa_1 = (\mu_1, \nu_1, \lambda_1)$, $(v-2)(v-3)$ **of the** $(v-1)^2$ **choices of** $\lambda_1, \kappa_1$)

$$
\begin{aligned}
\langle p(\mu|\mu_1)p(\mu|\lambda_1)p(\nu|\nu_1)p(\nu|\kappa_1)\rangle &= \frac{1}{m^4} \sum_{\psi_1,\psi_2,\psi_3,\psi_4} \mathbb{P}\left\{\mu_1 \xrightarrow{\psi_1} \mu; \lambda_1 \xrightarrow{\psi_2} \mu; \nu_1 \xrightarrow{\psi_3} \nu; \kappa_1 \xrightarrow{\psi_4} \nu\right\} \\
&= \frac{1}{m^4}\left[\frac{m^4}{v}\frac{v^{s-1}-1}{v^s-1}\frac{v^{s-1}}{v^s-2}\frac{v^{s-1}-1}{v^s-3}\right].
\end{aligned}
\tag{112}
$$

**Total.** Consider the factor multiplying $\langle (C^{(1)})^2\rangle$ in the right-hand side of Eq. 84,

$$
\begin{aligned}
\mathcal{F}(\mu,\nu) :=& \sum_{\mu_1,\nu_1}\left(\langle p(\mu|\mu_1)^2 p(\nu|\nu_1)^2\rangle - \frac{1}{v-1}\sum_{\kappa_1 \neq \nu_1}\langle p(\mu|\mu_1)^2 p(\nu|\nu_1)p(\nu|\kappa_1)\rangle \right. \\
& - \frac{1}{v-1}\sum_{\lambda_1 \neq \mu_1}\langle p(\mu|\mu_1)p(\mu|\lambda_1)p(\nu|\nu_1)^2\rangle \\
& \left. + \frac{1}{(v-1)^2}\sum_{\lambda_1 \neq \mu_1, \kappa_1 \neq \nu_1}\langle p(\mu|\mu_1)p(\mu|\lambda_1)p(\nu|\nu_1)p(\nu|\kappa_1)\rangle\right).
\end{aligned}
\tag{113}
$$

By organising the terms in the sum according to the classification of the previous paragraphs,

$$
\begin{aligned}
\mathcal{F}(\mu,\nu) =& \; v\left(\mathcal{F}_a^{(2\text{-}2)}(\mu,\nu) + (v-1)\mathcal{F}_b^{(2\text{-}2)}(\mu,\nu)\right) \\
& - \frac{2v(v-1)}{v-1}\left(\mathcal{F}_a^{(2\text{-}1\text{-}1)} + \mathcal{F}_b^{(2\text{-}1\text{-}1)} + (v-2)\mathcal{F}_c^{(2\text{-}1\text{-}1)}\right) \\
& + \frac{v(v-1)}{(v-1)^2}\left[\left(\mathcal{F}_a^{(1\text{-}1\text{-}1\text{-}1)} + (v-2)\mathcal{F}_b^{(1\text{-}1\text{-}1\text{-}1)}\right)\right. \\
& \left. + (v-2)\mathcal{F}_c^{(1\text{-}1\text{-}1\text{-}1)} + \mathcal{F}_d^{(1\text{-}1\text{-}1\text{-}1)} + (v-2)\mathcal{F}_e^{(1\text{-}1\text{-}1\text{-}1)} + (v-2)\mathcal{F}_f^{(1\text{-}1\text{-}1\text{-}1)} + (v-2)(v-3)\mathcal{F}_g^{(1\text{-}1\text{-}1\text{-}1)}\right].
\end{aligned}
\tag{114}
$$

For $\nu = \mu$, by summing all the case *i)* terms, we get

$$
\begin{aligned}
\mathcal{F}(\mu,\mu) &= \frac{v^s\left(mv^3(v+1) - v^{2+s}(1-v+mv+mv^2) + (v+1)v^{2s}(6-6v+mv+v^2+mv^2)\right)}{(vm)^3(v^s-1)(v^s-2)(v^s-3)} \\
&\xrightarrow{v\gg 1}\frac{(m+1)}{m^3}\xrightarrow{m\gg 1}\frac{1}{m^2}.
\end{aligned}
\tag{115}
$$

Summing, instead, all the case *ii)* terms, we get, for $\nu \neq \mu$,

$$
\begin{aligned}
\mathcal{F}(\mu,\nu) &= \frac{v^s\left(mv^3(v+1)+v^{2+s}(v-1-8m+7mv-3mv^2)\right)}{(v-1)(vm)^3(v^s-1)(v^s-2)(v^s-3)} \\
&= \frac{v^{3s}(6-10v+mv+5v^2+3mv^2-v^3-3mv^3+mv^4)}{(v-1)(vm)^3(v^s-1)(v^s-2)(v^s-3)} \xrightarrow{v\gg 1} \frac{1}{m^2}.
\end{aligned}
\tag{116}
$$

To sum up, as in the $i_1 \neq j_1$ case (Eq. 82), for large vocabulary size $v \gg 1$ and large $m$ (e.g. $m = fv^{s-1}$, with $f \in (0,1]$),

$$
\left\langle\left(C^{(2)}(\mu,\nu)\right)^2\right\rangle \xrightarrow{v,m\gg 1} \frac{\left\langle C^{(1)}(\mu_1,\nu_1)^2\right\rangle}{m^2}.
\tag{117}
$$

### D.3 Level-l LCA

By replacing $C^{(1)}$ with $C^{(\ell-1)}$ and $C^{(2)}$ with $C^{(\ell)}$, the recurrence relation Eq. 79 extends to the case where the LCA of the $i$-th and $j$-th tokens is a level-$\ell$ symbol. Asymptotically in $m$ and $v$, and independently of $\mu$ and $\nu$,

$$
\left\langle\left(C^{(1)}(\mu,\nu)\right)^2\right\rangle = \frac{(1-m/v^{s-1})}{v^3m}, \quad \left\langle\left(C^{(\ell)}(\mu,\nu)\right)^2\right\rangle = \frac{\left\langle\left(C^{(\ell-1)}(\mu,\nu)\right)^2\right\rangle}{m^2} \Rightarrow
$$

$$
\left\langle\left(C^{(\ell)}(\mu,\nu)\right)^2\right\rangle = \frac{(1-m/v^{s-1})}{v^3m^{2\ell-1}}.
\tag{118}
$$

## E  Sampling noise in the empirical correlation function

In this appendix, we prove that the sampling noise on empirical correlation functions of RHM data has a characteristic size $(v^2P)^{-1/2}$.

Let us denote, to ease notation, $\mathbb{P}\{X_{d-t}=\mu, X_d=\nu\}$ with $p(\mu,\nu)$, $\mathbb{P}\{X_{d-t}=\mu\}$ with $p(\mu)$ and $\mathbb{P}\{X_{d-t}=\mu\}$ with $p(\nu)$. When measuring probabilities from the frequency of observations over $P$ i.i.d. samples,

$$
\hat{p}(\mu,\nu) = \frac{1}{P}\sum_{k=1}^{P}\delta(X_{k,d-t}=\mu, X_{k,d}=\nu),
\tag{119}
$$

where $\hat{\ }$ denotes the empirical estimate and the indicator variable $\delta$ is 1 with probability $p(\mu,\nu)$ and 0 otherwise. With $\delta$ having finite mean and variance, by the central limit theorem,

$$
\hat{p}(\mu,\nu) \xrightarrow{P\to\infty} p(\mu,\nu) + \sqrt{\frac{p(\mu,\nu)(1-p(\mu,\nu))}{P}}\xi,
\tag{120}
$$

where $\xi$ is a Gaussian random variable with zero mean and unitary variance. Analogously,

$$
\hat{p}(\mu) \xrightarrow{P\to\infty} p(\mu) + \sqrt{\frac{p(\mu)(1-p(\mu))}{P}}\zeta_1,
$$

$$
\hat{p}(\nu) \xrightarrow{P\to\infty} p(\nu) + \sqrt{\frac{p(\nu)(1-p(\nu))}{P}}\zeta_2,
\tag{121}
$$

where $\zeta_1$ and $\zeta_2$ are also Gaussian random variables with zero mean and unitary variance, correlated with each other and with $\xi$.

As a result, the empirical estimation of $C_t(\mu,\nu)$ reads

$$
\hat{C}_t(\mu,\nu) \xrightarrow{P\to\infty} p(\mu,\nu)-p(\mu)p(\nu) + \sqrt{\frac{p(\mu,\nu)(1-p(\mu,\nu))}{P}}\xi
$$

$$
-p(\mu)\sqrt{\frac{p(\nu)(1-p(\nu))}{P}}\zeta_2 - p(\nu)\sqrt{\frac{p(\mu)(1-p(\mu))}{P}}\zeta_1.
\tag{122}
$$

In the limit of large $v$ and $m$, where $p(\mu, \nu)$ converges to $1/v^2$ plus vanishingly small fluctuations and $p(\mu)$, $p(\nu)$ converge to $1/v$ plus vanishingly small fluctuations, the dominant noise contribution is the one of $\xi$, with standard deviation

$$\sqrt{\frac{p(\mu, \nu)(1 - p(\mu, \nu))}{P}} \xrightarrow{v, m \gg 1} \sqrt{\frac{1}{v^2 P}}. \tag{123}$$

The correlation function $\tilde{C}(t)$ is the standard deviation of $C_t(\mu, \nu)$ over vocabulary entries. Hence, the sampling noise on $C_t(\mu, \nu)$ results in an additive factor of $(v^2 P)^{-1/2}$.

## F    Correlations between mask and tuples of observable tokens

In this section, we generalise the results of App. D and App. E to the correlations between the last token and a $s$-tuple of observable tokens.

Let us then replace $\mu$ and $i$ with the $s$-tuple of input features $\boldsymbol{\mu}$ and the multi-index $\boldsymbol{i}$. This change only affects the level-1 rules probability $p^{(1)}$ in Eq. 17 and Eq. 38. Therefore, we can write the tuple-token correlation with LCA at level $\ell$ as follows,

$$
\begin{aligned}
C^{(\ell)}(\boldsymbol{\mu}, \nu) &= \sum_{\mu_1, \nu_1} p_{\boldsymbol{i}_1}^{(1)}(\boldsymbol{\mu}|\mu_1) p_{j_1}^{(1)}(\nu|\nu_1) C^{(\ell-1)}(\mu_1, \nu_1) \\
&= \frac{1}{m} \sum_{\nu_1} p_{j_1}^{(1)}(\nu|\nu_1) C^{(\ell-1)}(\mu_1(\boldsymbol{\mu}), \nu_1),
\end{aligned} \tag{124}
$$

where the second line is obtained by recalling that, for each available $s$-tuple of input features $\boldsymbol{\mu}$, there is a unique level-1 variable $\mu_1(\boldsymbol{\mu})$ that can generate it, with probability $1/m$. The mean of $C^{(\ell)}(\boldsymbol{\mu}, \nu)$ vanishes together with that of $C^{(\ell-1)}(\mu_1(\boldsymbol{\mu}), \nu_1)$. Te variance reads

$$
\begin{aligned}
\left\langle \left( C^{(\ell)}(\boldsymbol{\mu}, \nu) \right)^2 \right\rangle &= \frac{1}{m^2} \sum_{\nu_1, \kappa_1} \left\langle p_{j_1}^{(1)}(\nu|\nu_1) p_{j_1}^{(1)}(\nu|\kappa_1) \right\rangle \left\langle C^{(\ell-1)}(\mu_1(\boldsymbol{\mu}), \nu_1) C^{(\ell-1)}(\mu_1(\boldsymbol{\mu}), \kappa_1) \right\rangle \\
&= \frac{1}{m^2} \sum_{\nu_1 = \kappa_1} \left( \frac{1}{v^2} + \sigma_p^2 \right) \left\langle \left( C^{(\ell-1)}(\mu_1, \nu_1) \right)^2 \right\rangle \\
&\quad + \frac{1}{m^2} \left( \frac{1}{v^2} - \frac{1}{v^2} \frac{v-1}{v^s - 1} \right) \sum_{\nu_1, \kappa_1 \neq \nu_1} \left\langle C^{(\ell-1)}(\mu_1(\boldsymbol{\mu}), \nu_1) C^{(\ell-1)}(\mu_1(\boldsymbol{\mu}), \kappa_1) \right\rangle \\
&= \frac{v}{m^2} \left[ \left( \frac{1}{v^2} + \sigma_p^2 \right) - \left( \frac{1}{v^2} - \frac{1}{v^2} \frac{v-1}{v^s - 1} \right) \right] \left\langle \left( C^{(\ell-1)}(\mu_1, \nu_1) \right)^2 \right\rangle, \tag{125}
\end{aligned}
$$

where we used Eq. 74. Using Eq. 29 and Eq. 34,

$$
\left\langle \left( C^{(\ell)}(\boldsymbol{\mu}, \nu) \right)^2 \right\rangle = \frac{v}{m^2} \left( \frac{v-1}{v} \frac{v^s}{v^s - 1} \frac{1}{vm} \right) \left\langle \left( C^{(\ell-1)}(\mu_1, \nu_1) \right)^2 \right\rangle \xrightarrow{v \gg 1} \frac{\left\langle \left( C^{(\ell-1)}(\mu_1, \nu_1) \right)^2 \right\rangle}{m^3}, \tag{126}
$$

which equals the token-token correlation divided by a factor of $m$.

Correspondingly, $\tilde{C}^\ell$ is reduced by a factor of $\sqrt{m}$. Crucially, since the average joint tuple-token probability $p(\boldsymbol{\mu}, \nu)$ is $1/(v^2 m)$, the sampling noise size, obtained via the calculations of App. E, is also reduced by a factor of $\sqrt{m}$, leaving the condition of Eq. 9 unaltered.

## G    Experiments on deep CNNs and scaling of the loss steps

In this section, we present empirical learning curves of Deep CNNs trained for last-token prediction (details in subsection A.1). In particular, we discuss discrepancies between these curves and those of Transformers (Fig. 2) in subsection G.1, verify the scaling with $m$ of the first two steps of Eq. 12 in subsection G.2, then discuss the role of the context window size $t$ in subsection G.3.

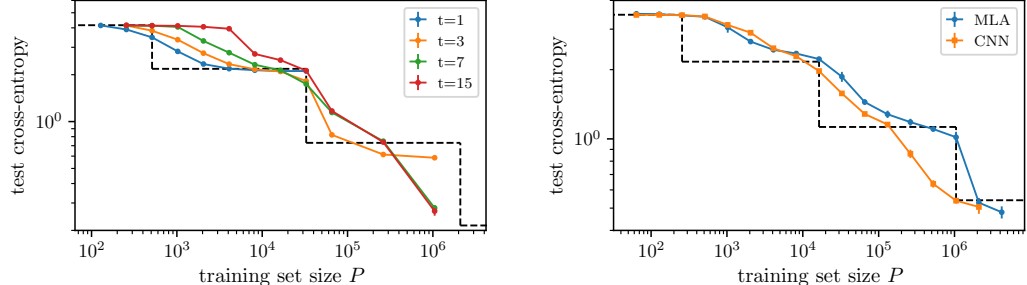

Figure 6: **Left:** Learning curves of deep CNNs trained on RHM data with $L=4$, $s=2$, $v=64$ and $m=8$ for different sizes $t$ of the context window. The network's depth is fixed to $\log s^L / \log(t+1)$ and the blacked dashed line represents predictions from Eq. 12 and Eq. 11. The finite context window causes saturation of the loss as predicted by our analysis. However, the third step occurs with less training data than $P_3$. **Right:** This discrepancy is highlighted by the comparison of Transformer and deep CNN learning curves, here for $L=4$, $s=2$, $v=64$ and $m=8$.

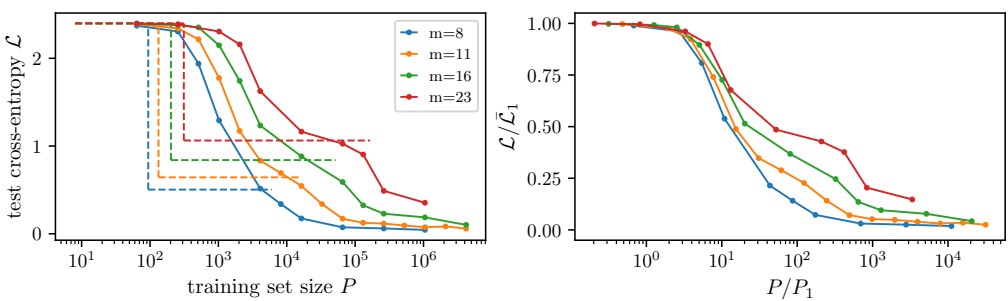

Figure 7: Learning curves of depth-3 CNNs trained on RHM data with $L=3$, $s=3$, $v=11$ and $m$ as in the key. Dashed curves highlight our prediction for the first step. In the right panel, the first step is made to collapse by rescaling $P$ with $P_1$ and $\mathcal{L}$ with $\mathcal{L}_0 = \log v$. The collapse confirms our prediction on the behaviour of $P_1$ with $m$.

## G.1 Differences between Transformers and deep CNNs

The learning curves of deep CNNs are qualitatively similar to those of transformers, but also present apparent quantitative differences, as shown in Fig.6. Specifically, a noticeable difference is the sample complexity of the third step $P_3$. This difference is possibly due to favourable implicit biases of CNNs, such as weight sharing. Indeed after learning the penultimate level-1 features in the second step, weight sharing would facilitate learning the other level-1 features along the entire data. As a result, the model can directly access the correlations between the last token and tuples of level-1 symbols. According to the discussion of App. F, these correlations are stronger than those with tuples of level-0 symbols by a factor of $\sqrt{m}$. Correspondingly, the sample complexity of the third step $P_3$ is reduced by a factor $m$ with respect to Eq. 12. In general, we can assume that, in the presence of weight sharing, after the $\ell$-th step all level-$(\ell-1)$ features have been learnt, so that the $(\ell+1)$-th step requires resolving correlations between the last token and tuples of level-$(\ell-1)$ features. The corresponding sample complexity scales like $m^{\ell+1}$ instead of $m^{2\ell-1}$. However, the steps with $\ell \geq 3$ occur for large values of the training set size, and we cannot investigate this issue systematically with our current numerical experiments.

## G.2 Scaling with the number of production rules $m$

Fig. 7 and Fig. 8 show a scaling analysis of the behaviour of $P_1$ and $P_2$ from Eq. 12 in Deep CNNs. The collapse achieved when rescaling the number of data $P$ by $P_\ell$ and the test loss by the value before the jump $\mathcal{L}_{\ell-1}$ confirms this prediction.

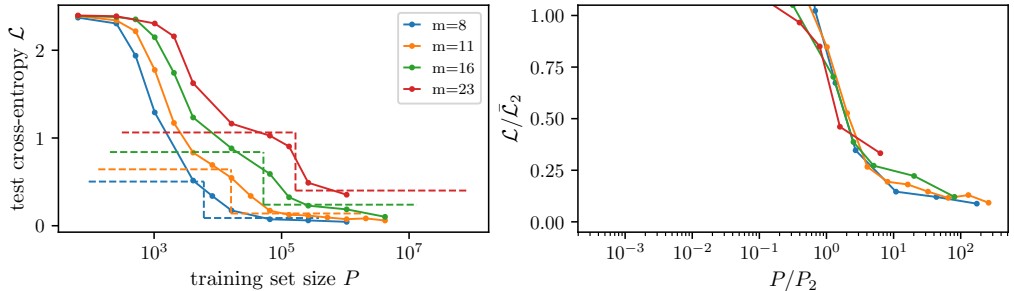

Figure 8: Same as Fig. 7 but focused on the second step, highlighted on the left by dashed curves. For the second step, collapse is achieved by rescaling $P$ with $P_2 = vm^3$ and $\mathcal{L}$ with $\mathcal{L}_1$ from Eq. 11.

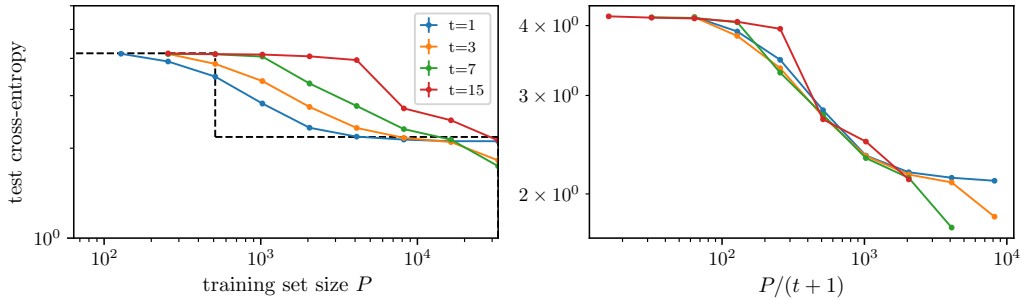

Figure 9: Zoom of the learning curves in Fig. 6, left, on the first step. The zoom highlights the dependence of the sample complexity on the context size $t$. The collapse of the curves on the right panel, achieved after dividing $P$ by $(t+1)$, reveals that $P_1 \propto (t+1)$. This dependence is analogous to the sample complexity of regression of a target function depending on a low-dimensional linear projection of a large-dimensional input [55].

### G.3 Scaling with the size of the context window $t$

Similarly, Fig. 9 shows a scaling analysis (for CNNs) of the behaviour of $P_1$ with the number of $s$-tuples in the input, proportional to $(t+1)$ with $t$ the size of the context window. The figure highlights a linear behaviour $P_1 \propto (t+1)$ that our analysis does not capture. Nevertheless, this behaviour is expected from the theory of regression with one-hidden-layer neural networks [55]: when the target function depends on a small number of variables among $d$, the sample complexity is generically proportional to $d$. Proving this result by considering a single or a few steps of gradient descent, as often done in this literature, is an interesting work for the future.

