# OpenReview forum: "Towards a theory of how the structure of language is acquired by deep neural networks"
_NeurIPS.cc/2024/Conference — NeurIPS 2024 poster_

### Official Review · Reviewer_Y3P4 · 2024-07-07

**Soundness:** 3
**Presentation:** 2
**Contribution:** 3
**Rating:** 6
**Confidence:** 3

**Summary:**

This paper proposes a conceptual characterization of how much data is required to learn the latent hierarchical structure of input languages. The theory is based on token correlations, which themselves are qualified based on token distances; these correlations are further traced back to the hierarchical structure that explains the distribution of sets of tokens. It is argued that language models can leverage these correlations to compose more complex hierarchical representations of the relationship between these sets of tokens.

A series of empirical analyses are performed to verify predictions from the theory. These include analyses from artificial PCFGs/RHMs, and verifications on natural language using Shakespeare plays. Key findings include that (1) more data is helpful because it allows LMs to resolve longer-distance dependencies and correlations; (2) that these correlations allow the LM to compose deeper representations of the relationship between sets of tokens.

**Strengths:**

* Thorough characterization of how latent hierarchical structures can be induced by neural language models from token correlations.
* The notion of “token correlation” is also deeply defined in a way that goes well beyond a naive analysis (e.g., bigram frequencies). Specifically, the relationship between token distance and token correlations, and how these can be traced back to the hierarchical dependencies between them, is explored in a way that I have not seen before.
* Two architectures, Transformers and CNNs, are analyzed. Comparisons are appropriately qualified and thorough.
* The analysis quantifies the importance of variables whose contributions to learning were not previously quantified or theoretically analyzed—most notably, (effective) context window size.

**Weaknesses:**

1. The linguistic foundations seem shaky. For example: the poverty of the stimulus is mischaracterized in L19-20 (see Questions field under “Suggestions” for more detail), and PCFGs are used as models of the structure of languages, when in fact language is context-*sensitive*, rather than context-*free* [1]. Nonetheless, I think the experiments do still give a thorough characterization of how hierarchical phenomena in general will be learned in models with self-supervision objectives.
2. Relatedly, the proposed RHM has many non-language-like features: the production rules are required to be unambiguous, whereas natural language is full of ambiguity that humans leverage intentionally to convey particular points; additionally, sampling production rules uniformly is suspect, considering that natural language leverages certain structures far more than others. This feels more like an issue with framing rather than methods, as I think this is still a valid analysis of how hierarchical structure is learned by models in general—just not necessarily how naturalistic text is learned. The Shakespeare text is meant to overcome this limitation, but this brings me to my next point:
3. Shakespearean English (early modern English) is very different in structure, word choice, and typological features than Modern English. It is also written in a poetic style that does not accurately capture the distribution of linguistic phenomena that would appear in more naturalistic domains or casual registers in any language. Why use this data, rather than something more naturalistic and contemporary like subsamples of The Pile or FineWeb?
4. The analyses seem very similar to past empirical work on how language models learn to compose atomic token units into hierarchical structures—most notably, the work in [2] on LSTMs. This work should be cited.
5. Unclear whether this will have practical impact on LM research or interpretability methods. I do not find this a significant flaw, but it may limit the impact and reach of this work. This is the main reason I have set my contribution score to 3.

All of these points are reasons why I have set my presentation score to 2 and soundness score to 3.

See Questions for references.

**Questions:**

Questions:
1. What was the motivation behind using Shakespearean text, rather than more naturalistic text? If you believe that the conjecture will also extend to any natural language dataset, would it be possible to run this analysis on something like The Pile?
2. What is the empirical impact of this work? This isn’t meant as a detraction of this work, as I believe analyses like these are valuable for our understanding. I’m more curious as to whether the authors believe there are specific impacts that this could have on how we train, interpret, or adapt LMs.

Suggestions/Typos:
* L19-20: The poverty of the stimulus is more in reference to the *ambiguity* of language’s underlying structure. It is not saying that the data is insufficient for hierarchical structure to be learned, but rather that the data can be explained by many differing hypotheses—including linear or “flat” explanations of the structure of the data. Nonetheless, children always settle on the exact same explanation: hierarchical structure, whereas language models in the past have been shown to be predisposed to linear explanations [3].
* “CGF” and “PCGF” are often used (when “CFG” or “PCFG” are meant). For example: L67-74, L97, L98
* L87: “wthe” -> “the”
* It could be good to cite

References:
[1] Shieber (1985). Evidence against the context-freeness of natural language. https://www.eecs.harvard.edu/shieber/Biblio/Papers/shieber85.pdf
[2] Saphra & Lopez (2020). LSTMs compose (and learn) bottom-up. https://aclanthology.org/2020.findings-emnlp.252/
[3] McCoy et al. (2023). Does syntax need to grow on trees? Sources of hierarchical inductive bias in sequence-to-sequence networks. https://aclanthology.org/2020.tacl-1.9/

**Limitations:**

The authors describe the practical limitations of their methods w.r.t. how neural models learn, but I believe the authors should also more thoroughly discuss the discrepancies between the data in their experimental setting and how natural language is actually structured, and whether/how these might affect findings.

---

> ### Author Rebuttal · Authors · 2024-08-06
>
> **Weakness 1,a (context-freeness).** We agree with the reviewer's remark on the context-freeness of natural languages and we will modify the text accordingly, stating specifically in both the abstract and the limitations section that the context-free description is approximate and captures many (but not all) syntactical forms observed in natural language.
>
> **Weakness 1,b (poverty of the stimulus.** We agree with the reviewer's definition and we will clarify the text to make it explicit. We will replace the sentence in line 20 of the introduction with *stating that the data acquired by children is not sufficient to uniquely determine the grammatical structure of their language*.
>
> **Weakness 2 (non-ambiguity of RHM).** We agree that the non-ambiguity of the rules and the uniformity of the production rules are not  language-like features, and we will emphasize this point further both in section 2.1 and in the limitations section.
>
> **Weakness 3 (additional dataset, contemporary text).** Please see comment above (author rebuttal).
>
> **Weakness 4 (ref. on LSTM).** Ref. 2 is indeed relevant to our work and we thank the reviewer for pointing it to us. We will include the following sentence at the beginning of the 'additional related works' section to acknowledge this result: *[ref. 2] provides evidence that LSTM language models learn short-range dependencies first and use them as a basis to build longer-range dependencies. Our results provide a theoretical framework for understanding this phenomenon.*
>
> **Weakness 5 (practical impact on LM research).**  This paper provides a novel explanation for the scaling laws characterising the performance of LLMs, and extends them to include the context window length(see also reply to **Weakness 2 of reviewer h6xP**). These laws are having a huge practical impact, both in showing that success could be achieved simply by scaling up LLMs and in guiding the choice of hyper-parameters depending on the available data and compute. In addition, our work suggests a criterion for optimising the size of the context window depending on the available data and the behaviour of token-token correlations. We will emphasise this point further in the conclusion section of the revised manuscript.
>
> **Question 1.** See comment above (author rebuttal).
>
> **Question 2.** See reply to weakness 5 and **Weakness 2 of reviewer h6xP**.
>
> In addition, we will implement the reviewer's suggestions and cite the mentioned papers in the revised manuscript.

---

> > ### Comment · Reviewer_Y3P4 · 2024-08-12
> >
> > Thanks for the response. I think the addition of the more naturalistic text data helps. It also sounds like much of the terminology surrounding the structure of natural language will be changed (W1-3).
> >
> > I think that the takeaway that scaling is helpful will not be surprising to readers (W5). Optimizing the size of the context window can be helpful, as shown in some prior work, but it still does not seem like it will change how we think about or train LMs.
> >
> > That said, I think there is value in this contribution. I am therefore keeping my current positive score.

---

### Official Review · Reviewer_qWaK · 2024-07-12

**Soundness:** 3
**Presentation:** 3
**Contribution:** 3
**Rating:** 7
**Confidence:** 3

**Summary:**

This paper investigates the sample complexity of learning languages generated by a kind of PCFG. The paper uses a model called the Random Hierarchy Model (RHM; a PCFG with a fixed tree structure) and identifies a relationship between the size of the training set and the "effective context window" that can be learned, which comes from the fact that more data is needed to resolve correlations between tokens that are farther away. The theoretical predictions are supported by experiments on synthetic data (generated by an RHM) and real data (lines from Shakespeare).

**Strengths:**

- I think this paper addresses an interesting and useful question about the relationship between sample complexity, language learning, and effective context size. Prior work has investigated how neural networks could recognize CFGs, and I think this work represents significant extension of this direction by trying to also characterize the learning dynamics and sample complexity. More generally, better understanding the relationship between sample complexity and effective context window could be significant given the growing interest in long-context language models.
- The key results (about the relationship between sample size and the ability to resolve long-distance correlations) are interesting, and I think these methods could be useful for future work on more naturalistic models (which the authors discuss in the conclusion).
- The experiments seem to support the theoretical results. I appreciate the analysis of hidden representations and the experiments with natural language.
- I find the writing and notation to be clear and the experiments easy to follow.

**Weaknesses:**

- I think biggest weaknesses are the assumptions of the RHM (a fixed tree geometry, all sequences of the same length). The fact that the correlation resolutions decay with distance is a consequence of the RHM, and it is not obvious how well these assumptions will hold for more naturalistic language data. However, I think this kind of limitation is necessary for getting analytical results, and the authors provide some empirical results on natural language and discuss how the assumptions can be relaxed in the future.
- I would have been interested to see some experiments on a language dataset other than the Shakespeare dataset. In particular, there might be other datasets that are more likely to exhibit long-distance dependencies (for example, predicting the next character in computer code), which could help to understand when the RHM might not be a good model for natural language data. However, I acknowledge that these experiments might be out of scope for this submission.

**Questions:**

No additional questions.

**Limitations:**

I think the authors adequately addressed the limitations of their work.

---

> ### Author Rebuttal · Authors · 2024-08-06
>
> **Weakness 1.** While the assumption of fixed tree geometry is unlikely to be satisfied in natural language data, we do not believe this assumption to be necessary for the power-law decay of correlations. [This paper](https://arxiv.org/abs/1606.06737), for instance, shows other examples of text data where correlations decay as a power of the distance and argues that this is possible due to the context-free structure. This is indeed what we find in our model, where correlations decay exponentially with the distance along the tree, while the actual distance between tokens grows exponentially with the tree distance, resulting in power-law behaviour. We expect that this result will be quite generic.
>
> **Weakness 2.** While we will add experiments on natural text data to the present submission, the research of effects that require other assumptions to be described is indeed out of the scope of the present work.

---

> > ### Comment · Reviewer_qWaK · 2024-08-12
> >
> > Thank you for the response and for the new results with WikiText. I still think the paper should be accepted and I will leave my score as is.

---

### Official Review · Reviewer_h6xP · 2024-07-14

**Soundness:** 3
**Presentation:** 3
**Contribution:** 3
**Rating:** 6
**Confidence:** 4

**Summary:**

This work studies the relation between the amount of training data required to learn the structure of language via the next token prediction objective & neural nets (CNNs, Transformers). The underlying training data is systematically varied using PCFGs and the authors find that that the size of the training dataset limits the resolution of token-token correlations to an effective context window (proportionally) and that a larger training set allows the representation of deeper hidden hidden variables. The authors also find that the sample complexities of learning the deeper representations is polynomial wrt effective context window. Besides synthetic data, the authors test their conjecture on a collection of Shakespeare's line and find the findings consistent with the conjecture.

**Strengths:**

1. This is an interesting paper which tries to understand a very fundamental problem -- what is the relation between the training set size and the resolution of the learnt token-token correlations in neural nets learnt via next token prediction.

**Weaknesses:**

1. The main weakness of the paper is the lack of comprehensive empirical experiments on "natural" text datasets -- the authors test their conjecture on Shakespeare's lines, but I find this limited experiment quite ad-hoc -- why not test it with more natural datasets across domains.
2. The authors didn't characterize their findings in the light of emergent properties or scaling laws of real large language models -- as such the importance of the findings is only weakly characterized in the paper.

**Questions:**

1. Why is the conjecture only tested on the shakespeare dataset? How about testing it on news corpora or other domains?
2. What does the conjecture predict as to the max context length for training real LLMs on natural text data? Can such predictions we tested empirically on LLMs?

**Limitations:**

Limitations were adequately addressed.

---

> ### Author Rebuttal · Authors · 2024-08-06
>
> **Weakness 1/Question 1.** Please see the comment above (author rebuttal).
>
> **Weakness 2.** We agree with the reviewer that this key point needs to be emphasised much more. In particular for deep generative models of data (large $L$), we indeed find that for large $P$, the learning curve can be described with a power-law $P^{−\alpha}$ as in~[17] (Kaplan *et al.*,  Scaling laws for neural language models), with exponent $\alpha =\log{ (m/v^{s−1})/(2 \log m)}$. It is interesting to note that this power-law results from a series of `emergent' phenomena where sub-portions of the data tree of different depths are learned. We will add a full paragraph on this point in section 4.1 and restate the importance of the result in the conclusions.
>
> **Question 2.** The tests performed in sections 4.3 ( behaviour of hidden representations with increasing $P$) and 5 (saturation of the scaling law due to the size of the context window) can also be performed on state-of-the-art LLMs trained on larger datasets.  Such a study would require a paper of its own---we are currently seeking to develop collaborations to make it possible. We also hope that the present paper can trigger concurrent works in that direction.

---

### Official Review · Reviewer_Fszw · 2024-07-30

**Soundness:** 3
**Presentation:** 3
**Contribution:** 3
**Rating:** 6
**Confidence:** 4

**Summary:**

1) This paper looks at the relationship between training dataset size in language model settings and the token x token correlations learned by the model.

2) The authors first use a synthetic setting to study this relationship and derive the results followed by testing it on a real dataset of lines from a Shakespeare’s play to demonstrate that the relationship discovered holds in real settings.

3) The  synthetic setting uses a probabilistic context free grammar model to generate the training data at different sizes, more specifically a random heirarchy model (RHM from https://arxiv.org/pdf/2307.02129) is used as it gives fine grained control over the correlations between generated tokens. Tokens far away in the heirarchy tree are less correlated.

4) The key contributions from this paper are :

a) Authors observe that both in RHM and in real settings the size of the training dataset size (P) caps the token correlations that can be learned by the model i.e correlations between tokens beyond a distance (t* - effective context window) in the heirarchy tree cannot be learned by the model. As training dataset size increases this cap is lifted enabling the model to learn these correlations well.

b) "Key finding is that the test loss decay levels off at a characteristic training set size that depends on the length of the context window and can be measured from correlations." (copied verbatim from paper contributions as it's a clear statement for other reviewers to look at )

c) A general framework based on synthetic data generation model (RHM) to further study the relationships between training dataset size, effective context window etc

**Strengths:**

1) The paper is well written with clear notations to motivate problem setup followed by clear sections solidifying each contribution.

2) The order of claims made is clear and authors did a good job at showing results in synthetic setting and then applying same methodology in real datasets and models.

**Weaknesses:**

1) The experiments on real-world datasets are limited and I'd encourage the authors to try this out on at least one other dataset of larger size and with a larger model.

2) Since the synthetic data generation process is cheaper, authors could have used higher P (training dataset size) in experiments.

**Questions:**

1) Have the authors tried replicating the results from Shakespeare experiment on a larger dataset ?

2) Have the authors tried varying model size in addition to dataset size to see if it recovers chinchilla curves under fixed compute ?

**Limitations:**

1) This work has no negative societal impact
2) Authors have done a good job at discussing limitations of their approach that comes due to their use of fixed geometry of data tree.

---

> ### Author Rebuttal · Authors · 2024-08-06
>
> **Weakness 1/Question 1.** Please see the comment above (author rebuttal).
>
> **Weakness 2**. Although the generation process is cheaper, the costs of training limit the available range of $P$ to that considered in the paper (up to a few million).
>
> **Question 2.** Since this paper focuses on sample complexity (and not on optimising performance at fixed compute), we varied the model size only to guarantee overparametrisation, in the sense that the training loss approaches zero as the training time increases. Increasing, depth, number of heads or embedding dimension of the architectures beyond the values reported in the paper does not affect performance (measured by the test loss at early stopping) beyond the fluctuations due to the randomness of the initialisation. We will clarify this point further in the text and mention the possibility of studying compute-optimal scaling laws as an interesting question for the future.

---

### Author Rebuttal · Authors · 2024-08-06

We thank all the referees for the detailed comments, and for finding our work interesting and supporting publication. They all pointed out that the paper will be improved by adding an additional set of experiments, perhaps involving a larger dataset made of contemporary text. We agree with the reviewers and, to answer their comments, we will repeat the Shakespeare dataset analysis presented in the first submission to the WikiText dataset, introduced [here](https://openreview.net/forum?id=Byj72udxe). As shown in the attached pdf, the token-token correlations of this dataset display the same behaviour as that of Figure 4, top right and bottom left---decay followed by training set size-dependent saturation. We are currently training a deep Bert-like transformer on this dataset to complete the analysis.

We address all the other comments and questions below the reviewers' comments.

---

> ### Author Response · Authors · 2024-08-12
> **Experiments on WikiText**
>
> Dear Reviewers,
>
> we have completed the analysis of the Wikitext dataset and we confirm that the results are similar to those obtained for the Shakespeare dataset (as displayed in Figure 4 of the original manuscript). We will include this analysis in the revised manuscript.
>
> We would be happy to share the relevant figure here if allowed.
>
> Best regards,
> The authors

---

### Comment · Area_Chair_S9xV · 2024-08-08
**Please respond to authors' rebuttals**

Dear Reviewers,

Thanks for writing your reviewers of the paper. Now the authors' rebuttals are in. Please go through them and see if they have addressed your questions. Please start discussions with the authors if you have further comments.

Regards,
AC

---

### Decision · Program_Chairs · 2024-09-25

**Decision:**

Accept (poster)

**Comment:**

This paper studies the relationship between training dataset size in language model settings and the token x token correlations learned by the model. All the reviewers agree that this is an interesting topic with solid theoretical analysis and thorough supporting experiments. The addition of results on WikiText during rebuttal makes the results more convincing.